

**A review on the global soil property maps for Earth system models**

Yongjiu Dai[1]*, Wei Shangguan[1]*, Dagang Wang[2], Nan Wei[1], Qinchuan Xin[2], Hua Yuan[1], Shupeng Zhang[1], Shaofeng Liu[1], Xingji Lu[1], Fapeng Yan[3]

[1] Guangdong Province Key Laboratory for Climate Change and Natural Disaster Studies, School of Atmospheric Sciences, Sun Yat-sen University, Guangzhou, China.
[2]School of Geography and Planning, Sun Yat-sen University, Guangzhou, China.
[3]College of Global Change and Earth System Science, Beijing Normal University, Beijing, China
Correspondence to: Yongjiu Dai (daiyj6@mail.sysu.edu.cn) and Wei Shangguan (shgwei@mail.sysu.edu.cn)

**Abstract.** Soil is an important regulator of Earth system processes, but remains one of the least well-described data layers in Earth System Models (ESMs). We reviewed global soil property maps from the perspective of ESMs, including soil physical and, chemical and biological properties, which can also offer insights to soil data developers. These soil datasets provide model inputs, initial variables and benchmark datasets. For modelling use, the dataset should be geographically continuous, scalable and have uncertainty estimates. The popular soil datasets used in ESMs are often based on limited soil profiles and coarse resolution soil type maps with various uncertainty sources. Updated and comprehensive soil information needs to be incorporated in ESMs. New generation soil datasets derived through digital soil mapping with abundant, harmonized and quality controlled soil observations and environmental covariates are preferred to those derived through the linkage method (i.e., taxotransfer rule-based method) for ESMs. SoilGrids has the highest accuracy and resolution among the global soil datasets, while other recently developed datasets offer useful compensation. Because there is no universal pedotransfer function, an ensemble of them may be more suitable to provide derived soil properties to ESMs. Aggregation and upscaling of soil data are needed for model use but can be avoided by using a subgrid method in ESMs at the expense of increases in model complexity. Producing soil property maps in a time series remains still challenging. The uncertainties in soil data needs to be estimated and incorporated into ESMs.

## 1 Introduction

Soil or the pedosphere is a key component of the Earth system, and plays an important role in water, energy and carbon balances and other biogeochemical processes. An accurate description of soil properties is essential in modelling capability of Earth System Models (ESMs) to predict land surface processes at the global and regional scales (Luo et al., 2016). Soil information is required by land surface models (LSMs), which are a component of ESMs. With the aid of computer-based geographic systems, many researchers have produced geographical databases to organize and harmonize large amounts of soil information generated from soil surveys during recent decades (Batjes, 2017; Hengl et al., 2017). However, soil datasets used in ESMs are not yet well updated or well utilized (Sanchez et al., 2009; FAO/IIASA/ISRIC/ISS-CAS/JRC, 2012). The popular soil datasets used in ESMs are outdated and have limited accuracies. Some soil properties, such as gravel (or coarse fragment) and depth to bedrock, are not utilized in most ESMs. The ESMs' schemes and structures must be changed to represent soil processes in a more realistic manner when utilizing new soil information (Brunke et al., 2016; Luo et al., 2016; Oleson et al., 2010). For example, Brunke et al. (2016) incorporated the depth to bedrock data in a land surface model using variable soil layers instead of the previous constant depth. Better soil information with a high resolution and better representation of soil in models has improved and will improve the performance of simulating the Earth system (eg., Livneh et al., 2015; Dy and Fung, 2016; Kearney and Maino, 2018).

ESMs require detailed information on the physical, chemical and biological properties of the soil. Site observations (called soil profiles) from soil surveys include soil properties such as soil depth, soil texture (sand, silt and clay fractions), organic matter, coarse fragments, bulk density, soil colour, soil nutrients (carbon (C), nitrogen (N), phosphorus (P), potassium (K) and sulphur (S)), amount of roots and so on. The range of soil data collected during a soil survey varies with scale, country or regional specifications, and projected applications of the data (i.e., type of soil surveys, routine versus specifically designed surveys). As a result, the availability of soil properties differs in different soil databases. However, soil hydraulic and thermal parameters as well as biogeochemical parameters are usually not observed in soil surveys, which need to be estimated by pedotransfer functions (PTFs) (Looy et al., 2017). This review focuses on soil data (usually single point observations at a given moment in time) from soil surveys, while variables such as soil temperature and soil moisture are beyond the scope of this paper.

Soil properties function in three aspects in ESMs:

1) Model inputs to estimate parameters. The soil thermal (soil heat capacity and thermal conductivity) and hydraulic characteristics (empirical parameters of the soil water retention curve and hydraulic conductivity) are usually obtained by fitting equations (PTFs) to easily measured and widely available soil properties, such as sand, silt and clay fractions, organic matter content, rock fragments and bulk density (Clapp and Hornberger, 1978; Farouki, 1981; Vereecken et al., 2010; Dai et al., 2013). Soil albedos are significantly correlated with the Munsell soil colour value (Post et al., 2000). For some ESMs, the parameters derived by PTFs are used as direct input

instead of being calculated in the models.
2) Initial variables. The nutrient (C, N, P, K, S and so on.) amounts and the
nutrients associated parameters (pH, cation-exchange capacity, etc.) in soils can be
used to initialize the simulations. Generally, their initial values are assumed to be at
steady state by running the model over thousands of model years (i.e., spin-up) until
there is no change trend in pool sizes (McGuire et al., 1997; Thornton and
Rosenbloom, 2005; Doney et al., 2006; Luo et al., 2016). To initialize nutrient
amounts using soil data derived from observations as background fields could largely
reduce the times of model spin-up, and could avoid the possibility of a non-linear
singularity evolution of the model, which means that the models may have multiple
equilibria and then provide a better estimate of the true terrestrial nutrient state. The
initial nutrient stocks settings are major factors leading to model-to-model variation in
simulation (Todd-Brown et al., 2014).
3) Benchmark data. Soil data, as measurements, could serve as a reference for
model calibration, validation and comparison. Soil carbon stock is one of the sol
properties that is most frequently used as benchmark data (Todd-Brown et al., 2013).
Other nutrient stocks, such as nitrogen stock, can also be used as benchmark data if an
ESM simulated these properties.
Soil properties have great spatial heterogeneity both horizontally and vertically.
As a result, ESMs usually incorporate soil property maps (i.e., horizontal spatial
distribution) for multiply layers rather than a global constant or a single layer. ESMs,
especially LSMs, are evolving towards hyper-resolutions of 1 km or finer with more
detailed parameterization schemes to accommodate the land surface heterogeneity
(Singh et al., 2015; Ji et al., 2017). Therefore, spatially explicit soil data at high
resolutions are necessary to improve land surface representations and simulations.
Because soil properties are observed at individual locations, soil mapping or spatial
prediction models are needed to derive a 3D representation of the soil distribution.
The traditional method (i.e., the linkage method, also called the taxotransfer rule-
based method) involves linking soil profiles and soil mapping units on soil type maps,
sometimes with ancillary maps such as topography and land use (Batjes, 2003;
FAO/IIASA/ISRIC/ISS-CAS/JRC, 2012). In recent decades, various digital soil
mapping technologies have been proposed by finding the relationships between soil
and environmental covariates (usually remote sensing data), such as climate,
topography, land use, geology and so on (McBratney et al., 2003).
There are many challenges related to the application of soil datasets in ESMs.
First, soil datasets are usually not appropriately scaled or formatted for the use of
ESMs and some upscaling issues, which are the most frequently encountered, need to
be addressed. The soil datasets produced by the linkage methods are polygon-based
and need to be converted to fit the grid-based ESMs. This conversion can be
performed by either the subgrid method or spatial aggregation. The up-to-date soil
data are provided at a resolution of 1 km or finer, while the LSMs are mostly ran at a
coarser resolution. Therefore, soil data upscaling is necessary before it can be used by
ESMs. Proper upscaling methods need to be chosen carefully to minimize the
uncertainty introduced by these methods in the modelling results (Hoffmann and
Christian Biernath, 2016; Kuhnert et al., 2017). Second, all the current global soil
datasets represent the average state of the last decades, and the production of soil
property maps in a time series is still challenging. Soil landscape and pedogenic
models are developed to simulate soil formation processes and soil property changes,
which can be incorporated into ESMs. The prediction of changing soil properties can
also be performed by digital soil mapping using the changing climate and land use as
covariates. Third, the uncertainty in the soil properties can be estimated, and adaptive
surrogate modelling based on statistical regression and machine learning may be used
to assess the uncertainty effects of soil properties on ESMs (Gong et al., 2015; Li et
al. 2018). Finally, the layer schemes of soil data sets need to be converted for model
use, and missing values for deeper soil layers need to be filled.
This paper is organized into the following sections. In section 2, we first
introduce soil datasets produced by the linkage method and digital soil mapping
technology at global and national scales, and then, we introduce the soil datasets that
have already been incorporated into ESMs, and we also present PTFs that are used in
ESMs to estimate soil hydraulic and thermal parameters. In section 3, several global
soil datasets are compared and evaluated with a global soil profile database. In section
4, two issues regarding the model use of soil data are described and existing
challenges related to the application of soil datasets in ESMs are discussed.    In
Section 5, a summary and the outlook of further improvements are provided.
**2 General methodology of deriving soil datasets for ESMs**
**2.1 Global and national soil datasets**
Two kinds of soil data are generated from soil surveys: maps (usually in the form
of polygon maps) representing the main soil types in landscape units and soil profiles
with soil property measurements which are considered to be representative of the
main component soils of the respective mapping units. ESMs usually require the
spatial distribution of soil properties (i.e., soil property maps) rather than information
about soil types. Two kinds of methods, i.e., the linkage method and the digital soil
mapping method, are used to derive the soil property maps.
Soil maps (the term soil map refers to soil type map in this paper) show the
geographical distribution of soil types, which are compiled under a certain soil
classification system. There are many soil mapping units (SMUs) in a soil map and an
SMU is composed of more than one component (i.e. soil type) in most cases. At the
global level, there is only one generally accepted global soil map, i.e., the FAO-
UNESCO Soil Map of the World (SMW) (FAO, 1971-1981). The SMW was made
based on soil surveys conducted between the 1930s and 1970s and technology that
was available in the 1960s. Several versions exist in the digital format (FAO, 1995,
2003b; Zöbler, 1986) and these products are known to be outdated. The information
on the initial SMW and DSMW has since been updated for large sections of the world
in the Harmonized World Soil Database (HWSD) product (FAO/IIASA/ISRIC/ISS-
CAS/JRC, 2012), which has recently been revised in WISE30sec (Batjes, 2016).
At the regional and national levels, there are many soil maps based on either
national or international soil classifications.    Some examples of major soil maps
available in digital formats are as follows: the Soil and Terrain Database (SOTER)
databases (Van Engelen and Dijkshoorn, 2012) for different regions, the European
Soil Database (ESB, 2004), the 1: 1 million Soil Map of China (National Soil Survey
Office, 1995), the U.S. General Soil Map (GSM), the 1:1 million Soil Map of Canada
(Soil Landscapes of Canada Working Group, 2010) and the Australian Soil Resource
Information System (ASRIS) (Johnston et al., 2003).
Soil profiles are composed of multiple layers called soil horizons. For each
horizon, soil properties are observed (e.g., site data) or measured (e.g., pH, sand, silt,
and clay content). At the global level, several soil profile databases exist. Here, we
discuss only the two most comprehensive databases. The World Inventory of Soil
Emission Potentials (WISE) database was developed as a homogenized set of soil
profiles (Batjes, 2008). The newest version (WISE 3.1) contains 10,253 soil profiles
and 26 physical and chemical properties. The soil profile database of the World Soil
Information Service (WoSIS) contains the most abundant profiles (about 118,400)
from national and global databases including most of the databases mentioned below
(Batjes et al., 2017), although only a selection of important soil properties (12) are
included (Ribeiro et al., 2018). Data from WoSIS have been standardized, with special
attention to the description and comparability of soil analytical methods worldwide.
However, many countries, although having a large collection of soil profile data, are
not yet sharing such data (Arrouays et al., 2017).
At the regional and national levels, there are many soil profile databases, usually
with soil classifications corresponding to the local soil maps, and here are some
examples: the USA National Cooperative Soil Survey Soil Characterization database
(http://ncsslabdatamart.sc.egov.usda.gov/), profiles from the USA National Soil
Information System (http://soils.usda.gov/technical/nasis/), Africa Soil Profiles
database (Leenaars, 2012), the ASRIS (Karssies, 2011), the Chinese National Soil
Profile database (Shangguan et al., 2013), soil profile archive from the Canadian Soil
Information System (MacDonald and Valentine, 1992), soil profiles from SOTER
(Van Engelen and Dijkshoorn, 2012), the soil profile analytical database for Europe
(Hannam et al., 2009), the Mexico soil profile database ( Instituto Nacional de
Estadística y Geografía, 2016), and the Brazilian national soil profile database
(Cooper et al., 2005).
The linkage method (called the taxotransfer rule-based method) involves linking
soil maps (with SMUs or soil polygons) and soil profiles (with soil properties)
according to taxonomy-based pedotransfer (taxotransfer in short, note that here,
pedotransfer here does not mean PTFs, which are a different thing) rules (Batjes,
2003). The criteria used in the linkage could be one or many factors, such as
following: soil class, soil texture class, depth zone, topographic class, distance
between soil polygons and soil profiles (Shangguan et al., 2012). Each soil type is
represented by one or a group of soil profiles that meet the criteria, and usually, the
median or mean value of a soil property is assigned to the soil type. Because the
linkage method assigned only one value or a statistical distribution to a soil type in the
soil polygons (usually a polygon contains multiple soil types with their fractions), the
intrapolygonal spatial variation is not considered. At the global level, many databases

were derived by the linkage method: the FAO SMW with derived soil properties (FAO, 2003a), the Data and Information System of International Geosphere-Biosphere Programme (IGBP-DIS) database (Global Soil DataTask, 2000), the Soil and Terrain Database (Van Engelen and Dijkshoorn, 2012) for multiply regions and countries, the ISRIC-WISE derived soil property maps (Batjes, 2006), the HWSD (FAO/IIASA/ISRIC/ISS-CAS/JRC, 2012), the Global Soil Dataset for Earth System Model (GSDE) (Shangguan et al., 2014) and WISE30sec (Batjes, 2016). The three most recent databases are HWSD, GSDE and WISE30sec. HWSD was built by combining the existing regional and national soil information updates. GSDE, as an improvement of HWSD, incorporated more soil maps and more soil profiles related to the soil maps, with more soil properties. GSDE accomplished the linkage based on the local soil classification, which required no correlation between classification systems and avoided the error brought by the taxonomy reference. In addition, GSDE provides an estimation of eight layers to a depth of 2.3 m, while HWSD provides an estimation of two layers to the depth of 1 m. WISE30sec is another improvement of HWSD that incorporates more soil profiles with seven layers up to 200 cm depth and with uncertainty estimated by the mean ± standard deviation. WISE30sec used the soil map from HWSD with minor corrections and climate zone maps as categorical covariates. Many national and regional agencies around the world have organized their soil surveys by linking soil maps and soil profiles, including the USA State Soil Geographic Database (STATSGO2) (Soil Survey Staff, 2017), Soil Landscapes of Canada (Soil Landscapes of Canada Working Group, 2010), the ASRIS (Johnston et al., 2003), the Soil-Geographic Database of Russia (Shoba et al., 2008), the European Soil Database (ESB, 2004), and the China dataset of soil properties (Shangguan et al., 2013).

Digital soil mapping (McBratney et al., 2003) is the creation and population of a geographically referenced soil database, generated at a given resolution by using field and laboratory observation methods coupled with environmental data through quantitative relationships (http://digitalsoilmapping.org/). Usually, the soil datasets derived by digital soil mapping provide grid-based spatially continuous estimation while the soil datasets derived by the linkage method provide estimations with abrupt changes at the boundaries of soil polygons. GlobalSoilMap is a global consortium that aims to create global digital maps for key soil properties (Sanchez et al., 2009). This global effort takes a bottom-up framework and produces the best available soil map at a resolution of 3 arc sec (about 100 m) with 90% confidence in the predictions. Soil properties will be provided for six soil layers (i.e., 0-5, 5-15, 15-30, 30-60, 60-100, and 100-200 cm). Many countries have produced soil maps following the GlobalSoilMap specifications (Odgers et al., 2012; Viscarra Rossel et al., 2015; Mulder et al., 2016; Ballabio et al., 2016; Ramcharan et al., 2018; Arrouays, 2018). The SoilGrids system (https://www.soilgrids.org) is another global soil mapping project (Hengl et al., 2014; Hengl et al., 2015; Hengl et al., 2017). The newest version (Hengl et al., 2017) at a resolution of 250 m was produced by fitting an ensemble of machine learning methods based on about 150,000 soil profiles and 158 soil covariates, which is currently the most detailed estimation of global soil distribution.

A third global soil mapping project is the Global SOC (soil organic carbon) Map of
the Global Soil Partnership, which focuses on country-specific soil organic carbon
estimates (Guevara et al., 2018).

Because soil property maps are products that are derived based on soil

measurements of soil profiles and spatial continuous covariates (including soil maps),
it is necessary to discuss the sources of uncertainty, spatial uncertainty estimation and
accuracy assessment of these derived data (the last two are different aspects of
uncertainty estimation). More attention should be given to this issue in ESM
applications instead of taking soil property maps as observations without error. There
are various uncertainty sources in the derivation of soil property maps, including
uncertainty from soil maps, soil measurements, soil-related covariates and the linkage
method itself (Shangguan et al., 2012; Batjes, 2016; Stoorvogel et al., 2017). The
following uncertainties are not a complete list of uncertainties, but the major
uncertainties are listed. Uncertainties in soil maps are major sources of global datasets
derived by the linkage methods. For these datasets, large sections of the world are
incorporated into the coarse FAO SMW map, and the purity of soil maps (referring to
the following website for the definition:
https://esdac.jrc.ec.europa.eu/ESDB_Archive/ESDBv2/esdb/sgdbe/metadata/purity_m
aps/purity.htm) is likely to be around 50 to 65% (Landon, 1991). Another important
source of uncertainty is the limited comparability of different analytical methods for a
given soil property when using soil profiles from various sources. A weak correlation
or even a negative correlation was found between different analytical methods,
although a strong positive correlation was revealed in most cases (McLellan et al.
2013). Both datasets of the linkage method and those by digital soil mapping are
subject to this uncertainty. Although there are no straightforward mechanisms to
harmonize the data, efforts have been undertaken to address this issue and provide
quality assessment (Batjes, 2017; Pillar 5 Working Group, 2017). Another source of
uncertainty comes from the geographic and taxonomic distribution of soil profiles,
especially for the under-represented areas and soils (Batjes, 2016). The fourth source
of uncertainty is from the linkage method itself. The linkage method does not
represent the intra-polygon spatial variation and usually does not explicitly consider
soil-related covariates like digital soil mapping, although there are cases where
climate and topography are considered; and Stoorvogel et al. (2017) proposed a
methodology to incorporate landscape properties in the linkage method. Finally,
uncertainty from the covariates is minor because spatial prediction models such as
machine learning in digital soil mapping can reduce its influences (Hengl et al., 2014),
although a more comprehensive list of covariates with higher resolution and accuracy
will improve the predicted soil property maps. Spatial uncertainty is estimated by
different methods for the linkage method and digital soil mapping methods. For the
linkage method, statistics such as standard derivation and percentiles can be used for
the spatial uncertainty estimation, and these statistics are calculated for the population
of soil profiles linked to a soil type or a land unit (Batjes, 2016). This estimation has
some limitations because soil profiles are not taken probabilistically but based on their
availability, especially for the global soil datasets. Uncertainty will be underestimated

when the sample size is not large enough to represent a soil type. For digital soil mapping, spatial uncertainty could be estimated by methods such as geostatistical methods and quantile regression forest (Vaysse and Lagacherie, 2017), which make sense of the statistics. The accuracy of the soil datasets derived by digital soil mapping is estimated by independent validation or cross-validation. However, this estimation is not trivial for those data derived by the linkage method due to the global scale, the support of the data and independent data (Stoorvogel et al., 2017), and most of these maps are validated by statistics such as the mean error and coefficient of determination. Instead, some datasets, including WISE and GSDE, use indictors such as the linkage level of soil class and sample size to offer quality control information (Shangguan et al. 2014; Batjes, 2016). A simple way to compare the accuracy of using datasets with both methods may be to use a global soil profile database as a validation dataset, though quite a number of these profiles were used when deriving these datasets and questions will be raised. We evaluated several global soil property maps in section 3.

**2.2 Soil dataset incorporated in ESMs**

Table 1 shows ESMs (specifically, their LSMs) and their input soil datasets. The ESMs in Table 1 cover the CMIP5 (Coupled Model Intercomparison Project) list except those without information about the soil dataset inputs. LSMs are key tools to predict the dynamics of land surfaces under climate change and land use. Five datasets are widely used, i.e., the datasets by Wilson and Henderson-Sellers (1985), Zöbler (1986), Webb et al. (1993), Reynolds et al. (2000), Global Soil Data Task (2000), and Miller and White (1998). Except for GSDE, HWSD and STATSGO (Miller and White, 1998) for the USA in Table 1, these datasets were derived from the SMW (note that large sections of GSDE and HWSD still used this map as a base map because there are no available regional or national maps) (FAO, 1971-1981) and limited soil profile data (no more than 5,800 profiles), which gained popularity because of its simplicity and ease of use. However, these datasets are outdated and should no longer be used because much better soil information, as introduced in Section 2.1, can be incorporated (Sanchez et al., 2009; FAO/IIASA/ISRIC/ISS-CAS/JRC, 2012).

In recent years, efforts have been made to improve the soil data condition in ESMs. The Land-Atmosphere Interaction Research Group at Beijing Normal University (BNU, now at Sun Yat-sen University) has put much effort into this topic. Shangguan et al. (2012, 2013) developed a China soil property dataset for land surface modelling based on 8,979 soil profiles and the Soil Map of China using the linkage method. Dai et al. (2013) derived soil hydraulic parameters using PTFs based on the soil properties by Shangguan et al. (2013). Shangguan et al. (2014) further developed a comprehensive global dataset for ESMs. The above soil datasets were widely used in the ESMs. Soil properties from these soil datasets, including soil texture fraction, organic carbon, bulk density and derived soil hydraulic parameters, were implemented in the Common Land Model Version 2014 (CoLM2014, http://land.sysu.edu.cn/). Li et al. (2017) showed that CoLM2014 was more stable

than the previous version and had comparable performance to that of CLM4.5, which may be partially attributed to the new soil parameters being used as input. Wu et al. (2014) showed that soil moisture values are closer to the observations when simulated by CLM3.5 with the China dataset than those simulated with FAO. Zheng and Yang (2016) estimated the effects of soil texture datasets from FAO and BNU based on regional terrestrial water cycle simulations with the Noah-MP land surface model. Tian et al. (2012) used the China soil texture data in a land surface model (GWSiB) coupled with a groundwater model. Lei et al. (2014) used the China soil texture data in CLM to estimate the impacts of climate change and vegetation dynamics on runoff in the mountainous region of the Haihe River basin. Zhou et al. (2015) estimated age-dependent forest carbon sinks with a terrestrial ecosystem model utilizing China soil carbon data. Dy and Fung (2016) updated the soil data for the Weather Research and Forecasting model (WRF).

Researchers have also put efforts into updating ESMs with other soil data. Lawrence and Chase (2007) used MODIS data to derive soil reflectance, which was used as a soil colour parameter in the Community Land Model 3.0 (CLM). De Lannoy et al. (2014) updated the NASA Catchment land surface model with soil texture and organic matter data from HWSD and STATSGO2. Livneh et al. (2015) evaluated the influence of soil textural properties on hydrologic fluxes by comparing the FAO data and STATSGO2. Folberth et al. (2016) evaluated the impact of soil input data on yield estimates in a globally gridded crop model. Slevin et al. (2017) utilized the HWSD to simulate global gross primary productivity in the JULES land surface model. Trinh et al. (2018) proposed an approach that can assimilate coarse global soil data by finer land use and coverage datasets, which improved the performance of hydrologic modelling at the watershed scale. Kearney and Maino (2018) incorporated the new generation of soil data produced by the digital soil mapping method into a climate model and found that compared to the old soil information, the soil moisture simulation was improved at a fine spatial and temporal resolution over Australia. A dataset of globally gridded hydrologic soil groups (HYSOGs250m) were developed based on soil texture and depth to bedrock of SoilGrids (Hengl et al., 2017) and groundwater table depth (Fan et al., 2013) for curve-number based runoff modelling of the U.S. Department of Agriculture (Ross et al., 2018).

Except for soil properties, the estimation of underground boundaries, including the groundwater table depth, the depth to bedrock (DTB) and depth to regolith and its implementation in ESMs is also a new focus. Fan et al. (2013) compiled global observations of water table depth and inferred the global patterns using a groundwater model. Pelletier et al. (2016) developed a global DTB dataset using process-based models for upland and an empirical model for lowland. This dataset was implemented in CLM4.5, and there were significant influences on the water and energy simulations compared to the default constant depth (Brunke et al., 2015). Shangguan et al. (2017) developed a global DTB by digital soil mapping based on about 1.7 million observations from soil profiles and water wells, which has a much higher accuracy than the dataset by Pelletier et al. (2016). Vrettas and Fung (2016) showed that weathered bedrock stores a significant fraction (more than 30%) of the total water despite its low

porosity. Jordan et al. (2018) estimated the global permeability of the unconsolidated and consolidated earth for groundwater modelling. However, due to the lack of data, an accurate global estimation of depth to regolith is not feasible. Caution should be used when employing the so-called soil depth products in ESMs. Soil depth maps are usually estimated based on observations from soil surveys, and soil depth (or depth to the R horizon) is assumed to be equal to DTB. However, these observations are usually less than 2 metres and usually do not reach the DTB (Shangguan et al., 2017). Thus, soil depth maps based on only soil profiles are significantly underestimated (one order of magnitude lower) compared to the actual DTB and should not be taken as the lower boundary of ESMs.

**2.3 Estimating secondary parameters using PTFs**

Earth system modellers have employed different PTFs to estimate soil hydraulic parameters (SHP), soil thermal parameters (STP), and biogeochemical parameters (Looy et al., 2017; Dai et al., 2013) or used these parameters as model inputs. Nearly all ESMs incorporated SHPs and STPs estimated by PTFs but not biogeochemical parameters. PTFs are the empirical, predictive functions that account for the relationships between certain soil properties (e.g., hydraulic conductivity) and more easily obtainable soil properties (e.g. sand, silt, clay and organic carbon content). Direct measurement of these parameters is difficult, expensive and in most cases impractical for obtaining sufficient samples to reflect spatial variation. Thus, most soil databases do not contain these parameters. PTFs provide an alternative means of estimating these parameters. In ESMs, SHPs and STPs are usually derived using simple PTFs, using only soil texture data as the input. As more soil properties become globally available, including gravel, soil organic matter and bulk density, more sophisticated PTFs that use additional soil properties can be employed in ESMs.

PTFs can be expressed as either numerical equations or by machine learning methodology which is more flexible for simulating the highly nonlinear relationship in analysed data. PTFs can also be developed based on soil processes. Most researches have not indicated where the PTFs can potentially be used, and the accuracy of a PTF outside of its development dataset is essentially unknown (McBratney et al., 2011). PTFs are generally not portable from one region to another (i.e. locally or regionally validated). Therefore, PTFs should never be considered as an ultimate source of parameters in soil modelling. Looy et al. (2017) reviewed PTFs extensively in earth system science and emphasized that PTF development must go hand in hand with suitable extrapolation and upscaling techniques such that the PTFs correctly represent the spatial heterogeneity of soils in ESMs. Although the PTFs were evaluated, it is unclear which set of PTFs are the best for global applications. Due to these limitations, a better way to estimate these parameters may be to use an ensemble of PTFs, which can provide the parameter variability. Dai et al. (2013) derived a global soil hydraulic parameter database using the ensemble method. Selection of PTFs was carried out based on the following rules, including a consistent physical definition, adequately large training sample and positive evaluations that are comparable with other PTFs. The selected PTFs not only included those in equations

but also machine learning PTFs. As a result, the modellers could use these parameters as inputs instead of calculating them in ESMs every time the model was run.

New generation soil information has already been utilized to derive SHPs and STPs in some studies. Montzka et al. (2017) produced a global map of SHPs at a 0.25° resolution based on the SoilGrids 1 km dataset. Tóth et al. (2017) calculated SHPs for Europe with EU-HYDI PTFs (Tóth et al., 2015) based on the SoilGrids 250 m. Wu et al. (2018) used an integrated approach that ensembles PTFs to map the field capacity of China based on multi-source soil datasets.

The PTF performance in ESMs has been evaluated in many studies, although PTFs have not been fully exploited and integrated into ESMs (Looy et al., 2017). Some examples are as follows. Chen et al. (2012) incorporated soil organic matter to estimate soil porosity and thermal parameters for use in LSMs. Zhao et al. (2018a) evaluated PTFs performance to estimate SHPs and STPs for land surface modelling over the Tibetan Plateau. Zheng et al. (2018) developed PTFs to estimate the soil optical parameters to derive soil albedo for the Tibetan Plateau, and the PTFs that were incorporated into an eco-hydrological model improved the model simulation of a surface energy budget. Looy et al. (2017) envisaged two possible approaches to improve parameterization of ESMs by PTFs. One approach is to replace constant coefficients in current ESMs that have spatially distributed values with PTFs. The other approach is to develop spatially exploitable PTFs to parameterize specific processes using knowledge of environmental controls and variations in soil properties.

**3 Comparison of available global soil datasets**

For the convenience of ESMs' application, we compared several available soil datasets and evaluated them with soil profiles from WoSIS for some of the key variables (sand, clay content, organic carbon, coarse fragment and bulk density) used in ESMs. In addition to the most recent developed soil datasets, we also included one old data set (i.e., IGBP) used in ESMs for the evaluation. It is not necessary to compare all the old data sets because they are based on similar, limited and outdated source data as described in section 2.2. These datasets have coarser resolutions (Table 1) than the newly developed soil datasets (Table 2).

We present basic descriptions of the new soil datasets in Table 2 and 3. As described in section 2.1, four available global soil datasets, i.e., HWSD, GSDE, WISE30sec and SoilGrids, have been developed in the last several years (Table 2). These soil datasets are selected to be shown here because they have global coverage with key variables used by ESMs and were developed with relatively good data sources in recent years; these data are also freely available. Old versions of these datasets are not shown here. Table 3 shows the available soil properties of these soil datasets. Except for WISE30sec, none of these databases contain spatial uncertainty estimations. The explained soil property variance in SoilGrids is between 56% and 83%, while the other datasets do not offer quantitative accuracy assessments. GSDE has the largest number of soil properties, while SoilGrids currently contains ten primary soil properties defined by the GlobalSoilMap consortium.

The accuracy of the newly developed soil datasets (SoilGrids, GSDE and HWSD)

and an old dataset (IGBP) are evaluated for five key variables using 94,441 soil profiles
from WoSIS (Table 4), though quite a number of the WoSIS soil profiles were
considered in the complication of these datasets which means that this evaluation is not
independent validation. We used four statistics in the evaluation, including mean error
(ME), root mean squared error (RMSE), coefficient of variation (CV) and coefficient
of determination ($R^2$). All soil datasets are evaluated for topsoil (0-30 cm) and subsoil
(30-100 cm). The layer schemes of soil datasets are different (Table 1) and were
converted to the two layers. Soil datasets are high in resolution and were converted to
a resolution of 10 km by averaging. All datasets have relatively small ME. In general,
SoilGrids have much better accuracy than the other three due to RMSE, CV and $R^2$,
and GSDE ranks the second, followed by IGBP and HWSD. However, IGBP is slightly
better than GSDE for bulk density and organic carbon content of topsoil. Notably, only
the IGBP does not contain coarse fragments, which is needed when calculating soil
carbon stocks. We did not evaluate the WISE30sec here to save time in data processing,
because previous evaluation using WoSIS showed that WISE30sec had slightly better
accuracy                    than                    HWSD
(https://github.com/thengl/SoilGrids250m/tree/master/grids/HWSD). This evaluation
has some limitations. First, the datasets developed by the linkage method, which give
the mean value of a SMU, resulted in an abrupt change between the boundaries of soil
polygons whereas the datasets developed by digital soil mapping simulated the soil as
a continuum with a spatial continuous change in soil properties; thus, these datasets
may not be comparable. Second, the original resolutions of soil datasets are different,
which means that maps with higher resolutions provide more spatial details, and we
should judge the map quality by not only the accuracy assessment but also by the
resolution. As a result, datasets with higher resolutions (i.e. HWSD, WISE30sec and
GSDE) are preferred to those with lower resolutions (i.e., IGBP) because the higher
resolution datasets have similar accuracy, especially when the LSMs are run at a high
resolution, such as 1 km. Third, the vertical variation is better represented by SoilGrids,
GSDE and WISE30sec with more than 2 layers and a depth of over 2m (Table 2), which
will provide more useful information for ESMs, especially when they model deeper
soils with multiple layers.
The new generation soil dataset produced by the digital soil mapping method gave
a very different distribution of soil properties from those produced by the linkage
method. Figure 1 shows the soil sand and clay fractions at the surface 0-30 cm layer
from SoilGrids, IGBP and GSDE. Figure 2 shows the SOC and bulk density at the
surface 0-30 cm layer from SoilGrids, IGBP and GSDE. Significant differences are
visible in these datasets. This difference will lead to different modelling results in ESMs.
Tifafi et al. (2018) found that the global SOC stocks down to a depth of 1 m
is 3,400 Pg when estimated by SoilGrids and 2500 according to HWSD, and the
estimates by SoilGrids are closer to the actual observations, although all datasets
underestimated the soil carbon stocks. Figure 1 of Tifafi et al. (2018) shows the global
distribution of soil carbon stocks by SoilGrids and HWSD.
In general, SoilGrids is preferred for ESMs' application because it currently has
the highest accuracy and resolution. When soil properties are not available in SoilGrids,
WISE30sec and GSDE offer alternative options. However, model sensitivity
simulations need to be performed to investigate the effects of different soil datasets on
ESMs in future studies.

**4 Soil data usage in ESMs and existing challenges**
**4.1 Model use of soil data derived by the linkage method**
Soil data by the linkage method are derived for each SMU or land unit and thus
are polygon-based, while ESMs are usually grid-based. However, soil data derived by
digital soil mapping are grid-based. Therefore, the compatibility between soil data
derived by the linkage method and ESMs must be addressed. In the soil map, a SMU
is composed of more than one component soil unit in most cases, and thus, a one-to-
many relationship exists between the SMU and profile attributes of the respective soil
units. This condition makes representing the attributes characterizing an SMU a
nontrivial task. To keep the whole soil variation of in an SMU, it is best to use the
subgrid method in ESMs (Oleson et al., 2010), i.e. aggregate values of soil properties,
and provide the area percentage of each value. This will bring about the problem of
mapping the soil subgrids with land cover (or plant function type) subgrids. A
possible solution is to classify the soil according to the soil properties and obtain a
number of defined soil classes (n classes) such as land cover types (m classes),
overlay the defined soil classes with land cover types and obtain n by m combinations
assuming the soil classes and land cover types are independent. However, this will
increase the computing time and complexity of the ESMs' structures, which requires
implementation the soil processes over each subgrid soil column within a grid instead
of the entire model grid.
Usually, the compatibility issue is addressed by converting the SMU-based soil
data to grid data using spatial aggregation. The ESMs uses grid data as input, and
each grid cell has one unique value of a soil property. Three spatial aggregation
methods were proposed to aggregate compositional attributes in an SMU to a
representative value (Batjes, 2006; Shangguan et al., 2014). The area-weighting
method (method A) obtains the area-weighting of soil attributes. The dominant type
method (method D) obtains the soil attribute of the dominant type. The dominant
binned method (method B) classifies the soil attributes into several preselected classes
and obtains the dominant class. All three methods can be applied to quantitative data,
while method D and method B can be applied to categorical data. The advantages and
disadvantages of these methods have been discussed (Batjes, 2006; Shangguan et al.,
2014). The choice should be made according to the specific applications (Hoffmann
and Christian Biernath, 2016). Method B provides binned classes, which are not
convenient for modelling, although method B is considered more appropriate to
represent a grid cell. Method A maintains mass conservation, which can meet most
model application demands. However, method A may be misleading in cases where
extreme values appeared in an SMU. For the linkage method, the uncertainty is
usually estimated by obtaining the 5 and 95 percentile soil properties (or other
statistics) of the soil profiles that are linked to an SMU. Because the frequency
distribution of the soil properties within a SMU is usually not a normal distribution or

any other typical statistical distribution, the application of statistics such as standard deviation to model use is not proper. This means that the uncertainty in the soil dataset derived by the linkage method cannot be incorporated into ESMs in a straightforward way, and technology such as bootstrap may be more suitable than methods that make assumptions on regarding the distribution.

The basic soil properties are often used to derive the secondary parameters, including SHPs and STPs by PTFs and soil carbon stock or other nutrient stocks by certain equations (Shangguan et al., 2014). This procedure could be performed either before or after the aggregation (referred to here as ''aggregating after'' and ''aggregating first''). Because the relationship between the soil basic properties and the derived soil parameters is usually nonlinear, the ''aggregating first'' method should be used. This was also proven by case studies (Romanowicz et al., 2005; Shangguan et al., 2014). However, some researchers have used the ''aggregating after'' method to produce misleading results (Hiederer and Köchy, 2012).

The aggregation smooths the variation in the soil properties between soil components within a given SMU (Odgers et al., 2012). To avoid aggregation, the spatial disaggregation of soil type maps can be used to determine the location of the SMU components, although the location error may be high in some cases (Thompson et al., 2010; Stoorvogel et al., 2017). This method depends on the high density of soil profiles to establish soil and landscape relationships. Folberth et al. (2016) showed that the correct spatial allocation of the soil type to the present cropland was very important in global crop yield simulations. Currently, aggregation is still the practical method to use at the global scale due to lack of data.

**4.2 Upscaling detailed soil data for model use**

The updated soil datasets derived by both the linkage method and digital soil mapping are usually at a resolution from 1 km to 100 m, and upscaling or aggregation is required to derive lower resolution datasets for model use. The aggregation methods mentioned above can be used. Moreover, there are many upscaling methods such as the window median, variability-weighted methods (Wang et al., 2004), variogram method (Oz et al., 2002), fractal theory (Quattrochi et al., 2001) and the Miller-Miller scaling approach (Montzka et al., 2017). However, few studies have been devoted to determining the upscaling methods that are suitable for soil data. A preliminary effort was made by Shangguan (2014). Five upscaling methods were compared, including the window average, window median, window modal, arithmetic average variability-weighted method and bilinear interpolation method. Differences between aggregation methods varied from 10% to 100% for different parameters. The upscaling methods affected the data derived by the linkage method more than the data derived by digital soil mapping. The window average, window median and arithmetic average variability-weighted method performed similar in upscaling. The RMSE increased rapidly when the window size was less than 40 pixels. Similar to the aggregation of SMUs, the ''aggregating first'' method is recommended when secondary soil parameters are derived. Again, an alternative to avoid the aggregation into one single value for a grid cell is to use the subgrid method in ESMs.

The upscaling effect of soil data on the model simulation has been investigated in
previous studies with controversial conclusions. For example, Melton et al. (2017) used
two linked algorithms to provide tiles of representative soil textures for subgrids in a
terrestrial ecosystem model and found that the model is relatively insensitive to subgrid
soil textures compared to a simple grid-mean soil texture at a global scale. However,
the treatment without soil subgrid structure in JULES resulted in soil moisture
dependent anomalies in simulated carbon flux (Park et al., 2018). Further researches
are necessary to investigate the upscaling effect on models.
**4.3 The changing soil properties**
There are no global soil property maps in the time-series because we do not have
enough available data. In all global soil property maps, all available soil observations
in recent decades have been used in the development of soil property maps without
considering the changing environment. Therefore. these datasets should be considered
as an average state. The critical issue for mapping global soil properties in a time series
is to establish a soil profile database with time stamps and then divide them into two or
more groups of different periods such as the 1950s-1970s. This is still quite challenging
at the global scale because the spatial coverage of soil profiles is quite uneven for
different periods and the sample size may not be adequately large to derive maps with
satisfactory accuracy.
Soil properties are changing, but we are now usually considering them to be static
in ESMs. As some ESMs already simulate the soil carbon, this may be considered in
PTFs used to estimate soil hydraulic and thermal parameters. Other soil properties
affecting soil hydraulic and thermal parameters include soil texture, bulk density, and
soil structure, but the change is relatively slow. The effect of environmental change on
soil properties is the topic of the quantitative modelling of soil forming processes, i.e.,
soil landscape and pedogenic models (Gessler et al., 1995; Minasny et al., 2008). If we
need to simulate the change in soil properties, a coupling of ESMs and soil landscape
and pedogenic models will be needed. Otherwise, we need to predict the soil properties
in the future using soil landscape and pedogenic models, which are small scale with
high uncertainty. The prediction of changing soil properties may also be performed by
digital soil mapping taken the changing (especially for the future) climate and land use
as covariates, which may be easier and more feasible than dynamic models.
**4.4 Incorporating the uncertainty of soil data in ESMs**
Incorporating the uncertainty of soil data in ESMs is increasing challenging.
Except for WISE30sec, all the current global soil datasets do not have a corresponding
uncertainty map for a soil property. However, the spatial uncertainty can be estimated
by the methods mentioned in section 2.1, and soil datasets with uncertainty maps will
be made available sooner or later. It is too expensive to run multiply ESM simulations
that combine the upper and lower bounds in all possible combinations to quantify the
effect of soil data uncertainty on ESMs. Instead, adaptive surrogate modelling based on
statistical regression and machine learning can be used to emulate the responses of
ESMs to the variation of soil properties at each location, which uses much less
computing time and proves to be effective and efficient (Gong et al., 2015; Li et al.
2018).
**4.5 Layer schemes and lack of deep layer soil data**
The layer scheme of a soil data set needs to be converted to that of ESMs for model
use. A simple method for this conversion is the depth weighting method. When a more
accurate conversion is needed, the equal-area quadratic smoothing spline functions can
be used, which is advantageous in predicting the depth function of soil properties
(Bishop et al., 1999). Mass conservation for a soil property of a layer is guaranteed by
this method under the assumption of a continuous vertical variation in soil properties.
This method may produce some negative values that should be set to zero.
The depth of soil observations in the soil survey is usually less than 2 m and thus
results in missing values for the deep layers of ESMs. For the lack of deep soil data,
there is no good solution other than extrapolating the values based on the observations
of shallower layers, which will lead to higher uncertainty of soil properties for deep
layers. The extrapolation can be performed by the abovementioned spline method or
simply by assigning the soil properties of the last layer to the rest of the deeper soil
layers. The DTB map (Shangguan et al., 2017) can be utilized to define the low
boundary of soil layers, and a default set of thermal and hydraulic characteristics can
be assigned for bedrocks.
**5 Summary and outlook**
In this paper, the status of soil datasets and their usage in ESMs is reviewed. Soil
physical and chemical properties serve as model parameters, initial variables or
benchmark datasets in ESMs. Soil profiles, soil maps and soil datasets derived by the
linkage method and digital soil mapping are reviewed at national, regional and global
levels. The soil datasets derived by digital soil mapping are considered to provide a
more realistic estimation of soils than those derived by the linkage method, because
digital soil mapping provides spatially continuous estimations of soil properties using
spatial prediction models with various soil-related covariates. Due to the evaluation of
soil datasets by WoSIS, SoilGrids have the most accurate estimation of soil properties.
However, other soil datasets, including GSDE and WISE30sec, can be considered as
compensation and they provide more soil properties.
The popular soil datasets used in ESMs are outdated and there are updated soil
datasets available. In recent years, efforts have been made to update the soil data in
ESMs. The effects of updated soil properties which are used to estimate soil hydraulic
and thermal parameters, were evaluated. Other major updates include soil reflectance,
ground water tables and DTB.
PTFs are employed to estimate secondary soil parameters, including soil hydraulic
and thermal parameters, and biogeochemical parameters. PTFs can take more soil
properties (i.e., SOC, bulk density and so on.) as input in addition to soil texture data.
An ensemble of PTFs may be more suitable to provide secondary soil parameters as
direct input to ESMs, because the ensemble method has a number of benefits and
potential over a single PTF (Looy et al., 2017).
Soil data derived by the linkage methods and high-resolution data can be
aggregated by different methods to be use in ESMs. The aggregation should be
performed after the secondary parameters are estimated. However, the aggregation will
omit the soil property variation. To avoid aggregation, the subgrid method in ESMs is
an alternative that increases the model complexity. The effect of different upscaling
methods on the performance of ESMs needs to be further investigated.
Because digital soil mapping has many advantages compared to the traditional
linkage method, especially in representing spatial heterogeneity and quantifying
uncertainty in the predictions, the new generation soil datasets derived by digital soil
mapping need to be tested in ESMs, and some regional studies have shown that these
datasets provide better modelling results than products by the linkage method (Kearney
and Maino, 2018; Trinh et al., 2018). Moreover, many studies from digital soil mapping
have identified that soil maps are not very important for predicting soil properties and
are usually not used as a covariate in most studies (e.g., Hengl et al., 2014; Viscarra
Rossel et al., 2015; Arrouays et al., 2018). However, the linkage method usually
considers the soil map to be a base map, which essentially affects the accuracy of the
derived soil property maps, especially for areas without detailed soil maps. As a data-
driven method, digital soil mapping requires soil profile measurements and
environmental covariates (in which the importance of soil maps is low), and by
including more of these data in mapping will improve the global predictions (Hengl et
al., 2017). More quality assessed data, analysed according to comparable analytical
methods, are needed to support such efforts. The soil data harmonization is undertaken
by the work of GSP Pillar 5 (Pillar 5 Working Group, 2017) and WoSIS (Batjes et al.,
2017). Data derived from proximal sensing, although with higher uncertainty than
traditional soil measurements, can be used in soil mapping (England and Viscarra
Rossel, 2018). To avoid spatial extrapolation, soil profiles should have good
geographical coverage. The temporal variation in global soil is quite challenging due to
a lack of data. Soil image fusion is also needed to merge the local and global soil maps,
and this fusion considers these maps as soil variation components for ensemble
predictions (Hengl et al., 2017). It may take years before a system for automated soil
image fusion is fully functional in an operational system for global soil data fusion.
Mapping the soil depth and DTB separately at the global level also remains challenging
due to a lack of data and the understanding of relevant processes. Uncertainty
estimation, especially spatial uncertainty estimation should be included in the soil
datasets developed in the future. However, incorporating the spatial uncertainty of the
soil properties in ESMs is still challenging due to the cost, and an alternative may be to
use adaptive surrogate modelling.
The gap is large between the amount of data that has been obtained in surveys and
the amount of data freely available. The soil profiles included in global soil databases
such as WoSIS comprise a very small fraction of the soil pits dug by human beings. For
example, there are more than 100,000 soil profiles from the second national soil survey
of China (Zhang et al., 2010) and no more than 9,000 were used to produce the national
soil property maps that are freely available (Shangguan et al., 2013). In the last century,
national soil surveys have been widely accomplished, primarily for agriculture purpose.

However, most of these legacy data are not digitalized and they are usually not made available to the science community even if digitalized. Obtaining these hidden soil data will require some mechanism such as government mandated regulations and money investments to make these data available (Pillar four Working Group, 2014; Pillar 5 Working Group, 2017). Arrouays et al. (2017) reported that about 800,000 soil profiles have been obtained from the selected countries, although most of these are not yet freely available to the international community. In addition, investments in new soil samplings should be made, especially in the under-represented areas. A good example is the U.S., which has the most abundant soil data freely available (http://ncsslabdatamart.sc.egov.usda.gov/) similar to many other data. Censored information produces censored maps and so on. If the hidden data could be made available in any way, science and the whole human being will be promoted. A true big data era is waiting for us. The data compatibility of different analysis methods and different description protocols including soil classifications is also an important issue and data harmonization is necessary when the data are made available to the public.

**Acknowledgements.** This work was supported by the National Key Research and Development Program of China under grants 2017YFA0604303 and 2016YFB0200801 and the Natural Science Foundation of China (under grants 41575072, 41730962 and U1811464).

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

Table 1. Lists of the soil dataset used by land surface models (LSM) of Earth System Models (ESM) or climate models (CM).

| Dataset | Resolution | ESM or CM | LSM | Input soil data |
|---|---|---|---|---|
| Elguindi et al. (2014) | | RegCM | BATS1e (Dickinson et al., 1993) or CLM3.5 | Soil texture classes and Soil color classes prescribed for BATS vegetation/land cover type |
| FAO (2003 a, b) | 5' | CanESM2 | CTEM (Arora et al., 2009) CLASS3.4 (Verseghy, 2000) | Soil texture |
| FAO (2003 a, b) | 5' | EC-EARTH | HTESSEL (Orth et al., 2016) | Soil texture classes |
| FAO (2003 a, b; outside Conterminous US) STATSGO (Miller and White, 1998) | 5' 30" | WRF CWRF | Noah (Chen and Dudhia, 2001) Noah-MP (Niu et al., 2011) CLM4 Other LSMs | Soil texture |
| GSDE (Shangguan et al., 2014) | 30" | CAS_ESM BNU_ESM GRAPES | CoLM 2014(Dai et al., 2003) | Soil texture, gravel, soil organic carbon, bulk density |
| GSDE (Shangguan et al., 2014) | 30" | WRF CWRF | Noah (Chen and Dudhia, 2001) Noah-MP (Niu et al., 2011) CLM4 Other LSMs | Soil texture |
| GSDE (Shangguan et al., 2014) | 30" | BCC_CSM 1.1 BCC_CSM 1.1(m) | BCC_AVIM 1.1 (Wu et al., 2014) | Soil texture |
| Hagemann (2002) | 0.5° (8km over Africa) | MPI-ESM ICON-ESM | JSBACH4 (Mauritsen et al. (2019) | Soil albedo |

| | | | | |
|---|---|---|---|---|
| Hagemann (2002) | 0.5° | MPI-ESM ICON-ESM | JSBACH4 (Mauritsen et al. (2019) | Field capacity, Plant-available soil water holding capacity and wilting point prescribed for ecosystem type |
| Hagemann et al. (1999) | 0.5° | MPI-ESM ICON-ESM | JSBACH4 (Mauritsen et al. (2019) | Volumetric heat capacity and thermal diffusivity prescribed for 5 soil types of FAO soil map |
| HWSD (FAO/IIASA/ISRIC/ISS -CAS/JRC, 2012) | 30" | GFDL ESM | GFDL LM4 (Zhao et al., 2018b) | Soil texture classes |
| HWSD (FAO/IIASA/ISRIC/ISS -CAS/JRC, 2012) | 30" | HadCM3 HadGEM2 QUEST | JULES/MOSESvn 5.4 (Best et al., 2011;Clark et al., 2011) | Soil texture |
| HWSD (FAO/IIASA/ISRIC/ISS -CAS/JRC, 2012) | 30" | CNRM-CM5 | SURFEX8.1 (Moigne,2018) | Soil texture, soil organic matter |
| IGBP-DIS (Global Soil Data Task, 2000) | 5′ | CESM CCSM CMCC–CESM FIO-ESM FGOALS (s2,gl,g2) NorESM1 | CLM 3.0 or CLM 4.0 or CLM 5.0 (Oleson, 2013) | Soil texture (sand, clay) |
| ISRIC-WISE (Batjes, 2006) combined with NCSD (Hugelius et al., 2013) | 5′; 0.25° | CESM CCSM CMCC–CESM FIO-ESM FGOALS (s2,gl,g2) NorESM1 | CLM 3.0 or CLM 4.0 or CLM 5.0 (Oleson, 2013) | Soil organic matter |

| | | | | |
|---|---|---|---|---|
| Lawrence and Chase (2007) | 0.05° | CESM CCSM CMCC– CESM FIO-ESM FGOALS (s2,gl,g2) NorESM1 | CLM 3.0 or CLM 4.0 or CLM 5.0 (Oleson, 2013) | Soil color class |
| Reynolds et al. (2000) | 5′ | GLDAS | Mosaic (Koster and Suarez, 1992) CLM2 Noah (Chen and Dudhia, 2001) VIC (Liang et al., 1994) | Soil texture classes |
| Webb et al. (1993) and Zöbler (1986) | 1° | GISS-E2 | GISS-LSM (Rosenzweig and Abramopoulos, 1997) | Soil texture |
| Wilson and Henderson-Sellers (1985) | 1° | HadCM3 HadGEM2 QUEST | JULES/MOSESvn 5.4 (Best et al., 2011;Clark et al., 2011) | Soil texture |
| Zöbler (1986) | 1° | ACCESS-ESM | CABLE2.0 (Kowalczyk et al 2013) | Soil texture classes |
| Zöbler (1986) | 1° | | SiB (Sellers et al., 1996; Gurney et al., 2008) | Soil texture classes |
| Zöbler (1986) | 1° | CFSv2 | CFSv2/Noah(Saha et al., 2014) | Soil texture |
| Zöbler (1986) | 1° | CSIRO-Mk3.6.0 | CSIRO-Mk3.6.0 (Rotstayn et al., 2012) | Soil texture classes |
| Zöbler (1986) | 1° | MIROC (4h,5) MIROC-ESM | MATSIRO (Takata et al., 2003) | Soil texture classes |

| | | | | |
|---|---|---|---|---|
| Zöbler (1986); Reynolds et al. (2000) | 1°; 5′ | IPSL-CM6 | ORCHIDEE [rev 3977] (Krinner, 2005) | Soil texture classes |

ACCESS = Australia Community Climate and Earth System Simulator

BATS = Biosphere-Atmosphere Transfer Scheme

BCC_CSM = Beijing Climate Center Climate System Model

BCC_AVIM = Beijing Climate Center Atmosphere and Vegetation Interaction Model

BNU_ESM = Beijing Normal University Earth System Model

CABLE = Community Atmosphere Biosphere Land Exchange

CanESM = Canadian Earth System Model

CAS_ESM = Chinese Academy of Sciences Earth System Model

CCSM = Community Climate System Model.

CESM = Community Earth System Model

CFS = Climate Forecast System

CLASS = Canadian Land Surface Scheme

CLM = Community Land Model

CMCC–CESM = Euro-Mediterranean Centre on Climate Change Community Earth System Model

CNRM-CM = Centre National de Recherches Meteorologiques Climate Model

CoLM = Common Land Model

CSIRO-Mk = Commonwealth Scientific and Industrial Research Organization climate system model

CTEM = Canadian Terrestrial Ecosystem Model

EC-EARTH = European community Earth-System Model

FAO = Food and Agriculture Organization (FAO-UNESCO) digital Soil Map of the World (SMW) at a 1:5 million scale

FGOALS = Flexible Global Ocean-Atmosphere-Land System Model

FIO-ESM = First Institute of Oceanography Earth System Model

GRAPES = Global/Regional Assimilation Prediction System

GFDL = Geophysical Fluid Dynamics Laboratory

GISS = Goddard Institute for Space Studies

GLDAS = Global Land Data Assimilation System

GSDE = Global Soil Dataset for Earth System Model

HadCM = Hadley Centre Coupled Model

HadGEM2-ES = Hadley Global Environment Model 2 - Earth System
HTESSEL = Tiled ECMWF Scheme for Surface Exchanges over Land
HWSD = Harmonized World Soil Database
ICON-ESM = Icosahedral non-hydrostatic Earth System Model
IGBP-IDS = Data and Information System of International Geosphere-Biosphere Program
IPSL-CM = Institute Pierre Simon Laplace Climate Model
ISRIC-WISE = World Inventory of Soil Emission Potentials of International Soil Reference and Information Centre
JSBACH = Jena Scheme of Atmosphere Biosphere Coupling in Hamburg
JULES/MOSES= Joint UK Land Environment Simulator/Met Office Surface Exchange Scheme
MATSIRO = Minimal Advanced Treatments of Surface Interaction and Runoff
MIROC = Model for Interdisciplinary Research on Climate
MPI-ESM = Max Planck Institute for Meteorology Earth System Model
Noah-MP = Noah-multiparameterization
NorESM1 = Norwegian Earth System Model
NCSD = Northern Circumpolar Soil Carbon Database
ORCHIDEE = Organising Carbon and Hydrology In Dynamic Ecosystems
QUEST = Quantifying and Understanding the Earth System
RegCM = Regional Climate Model
SiB = Simple Biosphere Model
STATSGO = State Soil Geographic Database
SURFEX = Surface Externalisée
WRF = Weather Research and Forecasting Model

Table 2 Four new global soil datasets forESM updates.

| Dataset* | Resolution | Number of layers | Number of properties | depth to the bottom of a layer (cm) | Mapping method |
|---|---|---|---|---|---|
| HWSD | 1km | 2 | 22 | 30, 100 | Linkage method |
| GSDE | 1km | 8 | 39 | 4.5, 9.1, 16.6, 28.9, 49.3, 82.9, 138.3, 229.6 | Linkage method |
| WISE30sec | 1km | 7 | 20 | 20,40,60,80,100,150,200 | Linkage method |
| SoilGrids | 250m | 6 | 7 | 5, 15, 30, 60, 100, 200 | Digital soil mapping |

*HWSD, GSDE, WISE30sec and SoilGrids are freely available at
http://www.iiasa.ac.at/web/home/research/researchPrograms/water/HWSD.html,
http://globalchange.bnu.edu.cn/research/data, https://www.isric.org/explore/wise-
databases, and http://www.soilgrids.org/, respectively.

Table 3 Derived soil properties considered in four global soil datasets.

| Soil property* | HWSD | GSDE | WISE30sec | SoilGrids | Soil property* | HWSD | GSDE | WISE30sec | SoilGrids |
|---|---|---|---|---|---|---|---|---|---|
| Drainage class | √ | √ | √ | | Total carbon | | √ | | |
| AWC class | √ | √ | | | Total nitrogen | | √ | √ | |
| Soil phase | √ | √ | | | Total sulfur | | √ | | |
| Impermeable layer | √ | √ | | | pH(KCL) | | √ | | √ |
| Obstacle to roots | √ | √ | | | pH(Cacl$_2$) | | √ | | |
| Additional property | √ | √ | | | Exchangeable Ca | | √ | | |
| Soil water regime | √ | √ | | | Exchangeable Mg | | √ | | |
| Reference soil depth | √ | √ | | | Exchangeable K | | √ | | |
| Depth to bedrock | | | | √ | Exchangeable Na | | √ | | |
| Gravel | √ | √ | √ | √ | Exchangeable Al | | √ | | |
| Sand, Silt, Clay | √ | √ | √ | √ | Exchangeable H | | √ | | |
| Texture class** | √ | | | | VWC at -10 kPa | | √ | | |
| Bulk density | √ | √ | √ | √ | VWC at -33 kPa | | √ | √ | |
| Organic Carbon | √ | √ | √ | √ | VWC at -1500 kPa | | √ | √ | |
| pH(H$_2$O) | √ | √ | √ | √ | Phosphorous by Bray method | | √ | | |
| CEC (clay) | √ | | √ | | Phosphorous by Olsen method | | √ | | |
| CEC (soil) | √ | √ | √ | | Phosphorous by New Zealand method | | √ | | |
| Effective CEC | | | √ | | Water soluble phosphorous | | √ | | |
| Base saturation | √ | √ | √ | | Phosphorous by Mechlich method | | √ | | |

| Property | | | | Property | | | |
|---|---|---|---|---|---|---|---|
| TEB | √ | | √ | Total phosphorous | √ | | |
| Calcium Carbonate | √ | √ | √ | Total Potassium | √ | | |
| Gypsum | √ | √ | √ | Salinity (ECE) | √ | √ | √ |
| Sodicity (ESP) | √ | | √ | Aluminium saturation | | | √ |
| C/N ratio | | | √ | | | | |

*CEC is cation exchange capacity. The base saturation measures the sum of exchangeable cations (nutrients) Na, Ca, Mg and K as a
percentage of the overall exchange capacity of the soil (including the same cations plus H and Al). TEB is the total exchangeable base
including Na, Ca, Mg and K. ESP is the exchangeable sodium percentage, which is calculated as Na*100/CECsoil. ECE is electrical
conductivity. AWC is the available water storage capacity. The first 9 soil properties on the left, including the drainage class and
AWC class are available for each soil type, while the other properties are available for each layer. Notebly, many different analytical
methods have been used to derive a given soil property, which is a major source of uncertainty.
**texture class can be calculated using sand, silt and clay content.
Table 4 Evaluation statistics of soil datasets using soil profiles from World Soil
Information Service (WoSIS).

| Soil property | Dataset | Topsoil (0-30 cm)* | | | | Subsoil (30-100 cm) | | | |
|---|---|---|---|---|---|---|---|---|---|
| | | ME | RMSE | CV | $R^2$ | ME | RMSE | CV | $R^2$ |
| Sand content | SoilGrids | -0.906 | 18.6 | 0.457 | 0.518 | -0.27 | 19.1 | 0.501 | 0.492 |
| (% in weight) | GSDE | -0.443 | 23.2 | 0.571 | 0.247 | -1.31 | 23.8 | 0.625 | 0.211 |
| | HWSD | 6.64 | 27.4 | 0.673 | 0.014 | 2.08 | 27.6 | 0.725 | -0.058 |
| | IGBP | 3.74 | 26.3 | 0.647 | 0.051 | 4.06 | 26.3 | 0.691 | 0.055 |
| Clay content | SoilGrids | 1.34 | 12.5 | 0.554 | 0.339 | 0.39 | 13.6 | 0.485 | 0.382 |
| (% in weight) | GSDE | -0.949 | 14.6 | 0.643 | 0.104 | -0.79 | 16.4 | 0.584 | 0.105 |
| | HWSD | 0.77 | 16.2 | 0.718 | -0.119 | 1.42 | 18.9 | 0.672 | -0.182 |
| | IGBP | 3.27 | 15.4 | 0.678 | 0.044 | 2.44 | 16.8 | 0.597 | 0.084 |
| Bulk density | SoilGrids | -79.7 | 237 | 0.164 | 0.338 | -33.5 | 212 | 0.136 | 0.327 |
| (kg/m3) | GSDE | -68.4 | 279 | 0.193 | 0.030 | -65.5 | 269 | 0.173 | -0.043 |
| | HWSD | -105 | 298 | 0.206 | -0.033 | -168 | 317 | 0.204 | -0.107 |
| | IGBP | -55.6 | 273 | 0.189 | 0.050 | -112 | 294 | 0.189 | -0.130 |
| Coarse | SoilGrids | 1.53 | 10.1 | 1.68 | 0.319 | 1.23 | 12.8 | 1.47 | 0.335 |
| fragment | GSDE | 3.2 | 13.5 | 2.24 | -0.165 | 3.18 | 16.8 | 1.93 | -0.115 |
| (% in volume) | HWSD | 1.8 | 13.2 | 2.2 | -0.164 | -0.40 | 16.2 | 1.87 | -0.081 |
| Organic carbon | SoilGrids | 6.21 | 29.8 | 1.69 | 0.218 | 0.99 | 23.5 | 3.32 | 0.134 |
| (g/kg) | GSDE | -0.354 | 34.5 | 1.95 | -0.095 | 0.45 | 27.4 | 3.87 | -0.174 |
| | HWSD | -3.67 | 36.2 | 2.05 | -0.194 | -1.38 | 27.4 | 3.87 | -0.172 |
| | IGBP | 0.61 | 33.4 | 1.89 | -0.026 | 1.67 | 28.5 | 4.02 | -0.268 |

*Quite a number of WoSIS soil profiles were considered in the compilation of the four products.
ME is the mean error. RMSE is the root mean squared error. CV is the coefficient of variation. $R^2$
is the coefficient of determination.

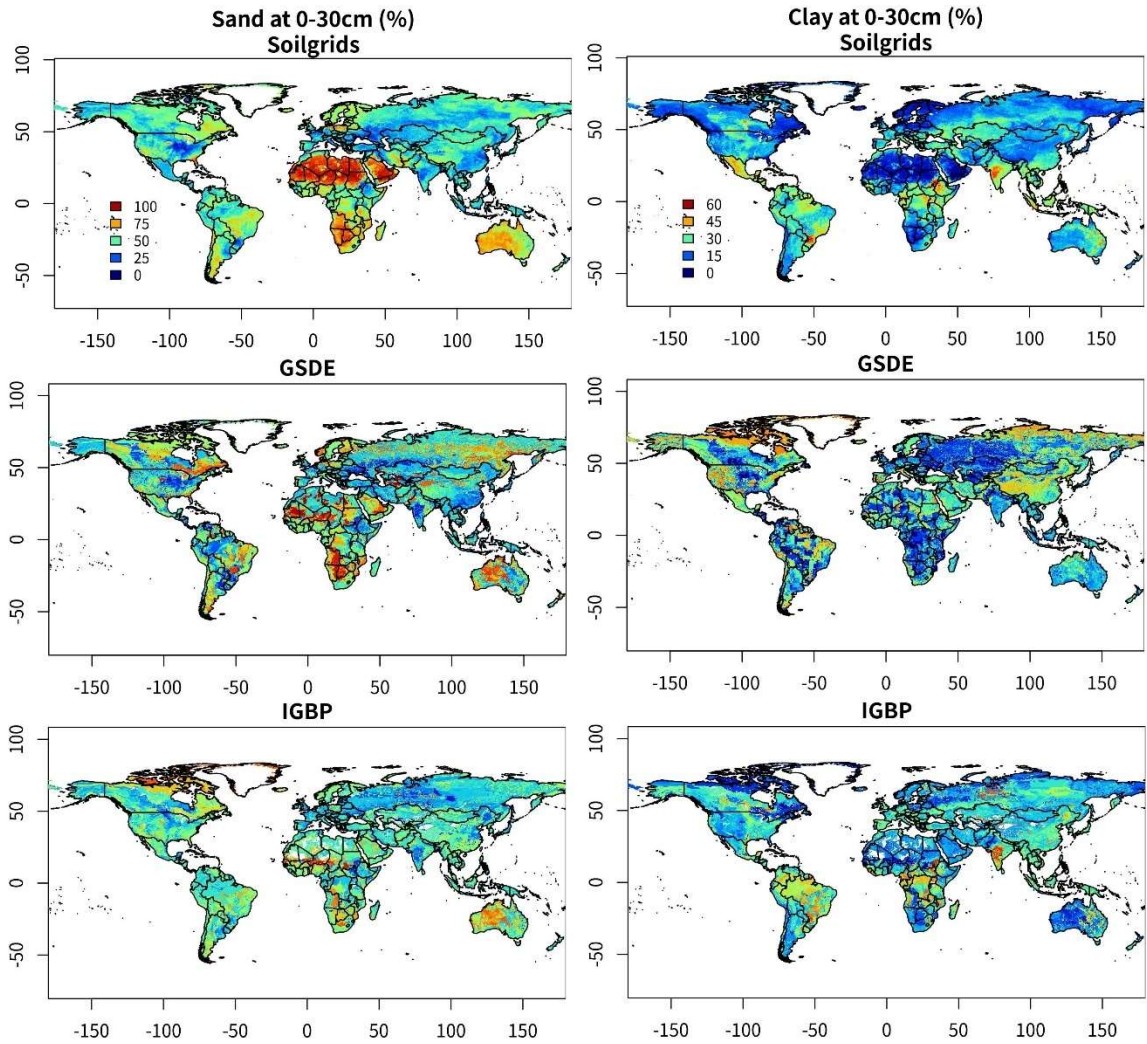

Figure 1 Soil sand and clay fraction at the surface 0-30 cm layer from SoilGrids, IGBP-DIS and GSDE. The difference among them will lead to different modelling results for ESMs. IGBP-DIS is Data and Information System of International Geosphere-Biosphere Program, and GSDE is Global Soil Dataset for Earth System Model.

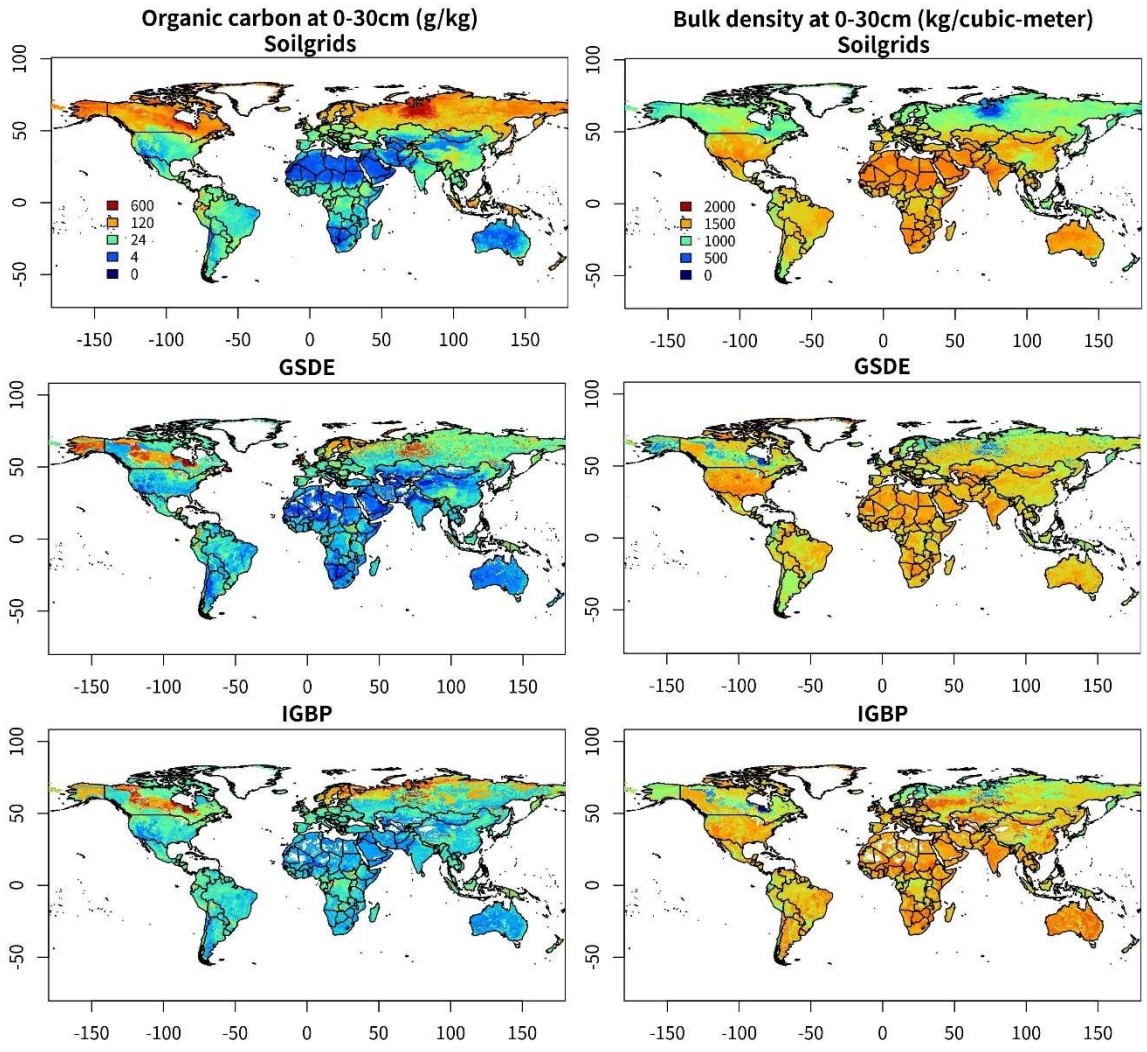

Figure 2 Soil organic carbon and bulk density at the surface 0-30 cm layer from SoilGrids, GSDE and IGBP.