# Peer review of "A review on the global soil property maps for Earth system model"

_SOIL, 2018_

## Referee Comment (RC1) · Anonymous Referee #1 · 31 Oct 2018

**1. General comments**

This is a timely review of global scale soil data sets that are used to underpin Earth System Models, and the still numerous, associated uncertainties. Such soil data sets have evolved greatly since the coarse 1-degree resolution map generalised by Zobler (1986) resulting in a new generation of digital soil maps, and the underpinning soil point data sets and/or covariates layers. That being said, I have a number of queries and comments. For example, rather little attention is given to difficulties associated with the limited comparability of soil analytical data worldwide and uncertainty propagation. Further, several recent global soil databases of possible interest for ESM modelling have not been considered in the review and discussion.

The manuscript would benefit from a thorough English edit by a native speaker.

[Figure]

2. Specific comments

L15-16: Rephrase this as e.g.: Soil is an important regulator of earth system processes, but remains one of the least well-described data layers in such models.

L17: Function as → provide

L22: Abundant soil observations are not 'enough'; these should have been analysed according to comparable analytical methods and quality-assessed (which is seldom the case, see Batjes et al. 2017). What about the geographical distribution, or possible clustering, of the available (i.e. shared) soil profile data?

L24: By their nature, pedotransfer functions generally are not portable from one region to the other. Please add some discussion.

L24-25: Speculative as written, provide some arguments for this.

L27-28: What about uncertainty in the co-variates?

L35-36 / 45: You may consider the following reference here: http://dx.doi.org/10.1002/2015GB005239.

L43: Remove available

L45: How do you define 'better' here? Please clarify.

L47-48: Also other types of soil data, for example soil biology (see ref. line L35-36). See also discussion in https://doi.org/10.1111/gcb.13896.

L56: Useful to say that the range of soil data collected during a soil survey, will vary with scale and projected applications of the data (i.e. type of soil survey, routine versus surveys/studies aimed at answering specific user demands).

L72: How would you define reliable soil data? Remove from this sentence.

L76: Rather refer to measurements here.

L87-88: Should add HWSD (FAO/IIASA/ISRIC/ISS-CAS/JRC, 2012) as reference for this type of 'traditional' approach.

L93: usually not ready for ... → ... not appropriately scaled or formatted for ...

L113-114: ... representing main soil types in a landscape unit characterised by soil profiles considered representative for the main component soils of the respective mapping units.

L124: Rephrase this: ... (FAO, 2003b, Zobler 1986) and these products are known to be outdated. The information on the initial SMW and DSMW has since been updated for large sections of the world in the HWSD product (FAO/IIASA/ISRIC/ISS-CAS/JRC, 2012), which has recently been revised in WISE30sec (http://dx.doi.org/10.1016/j.geoderma.2016.01.034).

L124-125: Start new paragraph for the regional and national level data.

L132: multiply –> multiple

L133: soil properties are observed (e.g. site data) or measured (e.g. pH, sand, silt, clay content)

L138-141: Important to mention here that data served through WoSIS have been standardised, with special attention for the description/comparability of soil analytical methods worldwide. See: http://dx.doi.org/10.17027/isric-wdcsoils.20180001. Also an important element for the discussion is that many countries, although having a large collection of soil profile data, are not yet sharing such data. See for example: https://doi.org/10.1016/j.grj.2017.06.001

L141: The initial list of attributes corresponds with the GlobalSoilMap specifications, with additional properties added/considered later in WoSIS (see http://dx.doi.org/10.17027/isric-wdcsoils.20180001).

L164: The linkage methods assigns a best-estimate for each soil property (and soil interval) under consideration to each component soil unit of a polygon (see e.g. HWSD). [see also 359-360]

L171-173: For a more comprehensive review see also: http://dx.doi.org/10.1016/j.geoderma.2016.01.034 and http://dx.doi.org/10.1002/ldr.2656.

L178: FYI, WISE30sec considers seven layers up to 200 cm depth and 20 soil properties.

L201: Possibly, also mention the GSOC effort of the GSP here, see: https://doi.org/10.5194/soil-4-173-2018

L205: . . . which is currently the most detailed, though not necessarily most accurate, estimation of . . .

L214: See also: Tifafi M, Guenet B and Hatté CCG 2018. Large differences in global and regional total soil carbon stock estimates based on Soil-Grids, HWSD and NCSCD: Intercomparison and evaluation based on field data from USA, England, Wales and France. Global Biogeochemical Cycles, 42-56. http://dx.doi.org/10.1002/2017GB005678

Note: This paper is erroneously referred to as Marwa et al. 2018 in manuscript. This should be: Tifafi et al. 2018.

L214: Check if this is for 0-100 cm; likely these estimates are for 0-200 cm (see also recent sources mentioned above).

L224: Large sections of HWSDv1.2 still draw on the now outdated DSMW.

L295-296: See earlier comments.

L299: WISE30sec presents estimations of uncertainty, unlike the HWSD and GSDE.

L300: Needs some discussion and references to publications on the subject.
L320: Larger number of soil properties for GSDE, but what about the accuracy of the predictions? (not given as indicated earlier).

L303: Rephrase. . . . SoilGrids products currently consider the list of attributes as defined by the GlobalSoilMap consortium.

L323: Most PTFs are not portable (i.e. locally or regionally validated).

L331-332: add database (word is missing in sentence)

L360-361: . . . component soil unit in most cases, and thus a one-to-many relationship exists between the SMU and the profile attributes of the respective soil units. . .

L397-398: Possibly, rephrase this sentence.

L410: remove high from sentence

L441-442: Provide some justification (a sentence or two) for this statement.

L451-452: Speculative as written. Please provide some evidence for this.

L460: and quantifying uncertainty in the predictions

L461: 'need to gain popularity in . . .'. Basically, the "proof of the pudding is in the eating".

L462: What I miss in this paper, is a discussion of the inherent uncertainty attached to using soil profile data coming from various sources. Often, little consideration is given to differences in analytical methods used for analysing e.g. soil organic carbon content worldwide (see Shangguang et al 2014, who consider this as 'a major imitation to their approach'). For a discussion of issues see e.g.: http://dx.doi.org/10.17027/isric-wdcsoils.20180001

L463-464: More soil profiles is not necessarily the solution. More quality-assessed data, analysed according to comparable analytical methods, are needed to support such efforts. Reference should be made to 'new' types of data

as derived from proximal sensing (e.g. http://dx.doi.org/10.5194/soil-2017-36), and associated limitations. Reference, in this respect, could also be made to the GLOSOLAN effort, initiated by the GSP (http://www.fao.org/global-soil-partnership/resources/events/detail/en/c/1037455/) and work of GSP Pillar 5 towards harmonisation (http://www.fao.org/3/a-bs756e.pdf). Also, importantly, the geographical distribution and possible clustering of the shared soil profiles.

L471-475: True, but how many of these profiles are actually being shared for the greater benefit of the international community? See paper by Arrouays et al. 2017 for a discussion.

L479: Some reference to the ongoing work of the Global Soil Partnership, Pillars 4 and 5, is needed here.

L948: Table 2 is not complete; 'recent' datasets not yet considered in the review should be added here ( http://dx.doi.org/10.1002/ldr.2656 ; http://dx.doi.org/10.1016/j.geoderma.2016.01.034 ). Idem for Table 3.

L952: Table 3. Change title to "Derived soil properties considered in three global soil datasets". Essentially, this is a simple enumeration of derived soil properties. However, the fact that many different analytical methods have been used to derive a given soil property (e.g. soil organic carbon Walkley & Black method or LECO total analyses) or which CEC (e.g. measured at 'field pH' or in a buffer-solution at 'pH7' or 'pH8') has been considered is not mentioned here (in a footer perhaps). In their study, Shangguan et al. (2014) rightly indicate that this has not been the case and indicate that they see this an important limitation. However, there are still no straightforward mechanisms for harmonising the data (cf. GSP Pillar 5 and GLOSOLAN activities, as mentioned above).

Potaasium –> Potassium

---

## Author Comment (AC1) · 16 Nov 2018

**1. General comments**

This is a timely review of global scale soil data sets that are used to underpin Earth System Models, and the still numerous, associated uncertainties. Such soil data sets have evolved greatly since the coarse 1-degree resolution map generalised by Zobler (1986) resulting in a new generation of digital soil maps, and the underpinning soil point data sets and/or covariates layers. That being said, I have a number of queries and comments. For example, rather little attention is given to difficulties associated with the limited comparability of soil analytical data worldwide and uncertainty propagation. Further, several recent global soil databases of possible interest for ESM modelling

have not been considered in the review and discussion.

Reply: Thanks for your valuable and detailed comments which help us a lot in improving our manuscript. The reviewer's comments have been addressed one by one in the following replies. This review was done from the perspective of ESMs and its users rather than that of soil data development. So we omitted some details about data development and associate uncertainty as pointed out by the reviewer. We are aware that many uncertainty sources exist in the derived soil dataset, which need attentions to be paid by ESM applications. After considering the comments of the reviewer, we added a paragraph concentrating on the uncertainty sources, uncertainty estimation and accuracy of soil data. And the comparability of soil analytical data, the covariates uncertainty and others are discussed in this paragraph. Some contents about the uncertainty in the original manuscript were also moved to this paragraph. As we see in the literature, ESMs usually do not consider much about uncertainty or even data quality of soil properties, which is not a good situation. ESM users should be more concerned about the uncertainty estimation rather than the uncertainty sources, while data developers need to know both aspects well. Further, we added more global soil databases as suggested by the reviewer (see the reply to table 2 and 3).

Here is the uncertainty paragraph we added:

Because soil property maps are derived products based on soil measurements of soil profiles (point observations) and spatial continuous covariates (including soil maps), it is necessary to discuss the uncertainty sources, uncertainty estimation and accuracy assessment of these derived data. More attention should be paid to this issue in ESM applications instead of taking soil property maps as observations without error. There are various uncertainty sources in deriving soil property maps, including uncertainty from soil maps, soil measurements, soil-related covariates and the linkage method itself (Shangguan et al., 2012; Batjes, 2016; Stoorvogel et al., 2016). The following may not be the complete list of uncertainty but the major ones. The uncertainty of soil maps is a major source of global dataset derived by the linkage methods. For these

dataset, large sections of the world are drawn on the coarse FAO SMW map and the purity of soil maps is likely to be around 50 to 65% (Landon, 1991). Another important source of uncertainty is the limited comparability of different analytical methods of a given soil property in using soil profiles coming from various sources. A week correlation or even negative correlation were found between different analytical methods, though strong positive correlation are revealed in most cases (McLellan et al. 2013). Both dataset by the linkage method and those by digital soil mapping suffers this uncertainty. Though there are no straightforward mechanisms to harmonize the data, efforts are undertaken to address this issue and provide quality assess (Batjes, 2017; Pillar 5 Working Group, 2017). Another source of uncertainty comes from the geographic and taxonomic distribution of soil profiles, especially for the under-represented areas and soils (Batjes, 2016). The fourth source of uncertainty is from the linkage method itself. It does not represent the intra-polygon spatial variation and usually do not consider soil forming factors explicitly like digital soil mapping, though Stoorvogel et al. (2017) proposed a methodology to incorporate landscape properties in the linkage method. Finally, uncertainty from the covariates is minor because spatial perdition models such as machining learning in digital soil mapping can reduce its influences (Hengl et al., 2014). Uncertainty are estimated by different methods for the linkage method and digital soil mapping methods. For the linkage method, statistics such as standard derivation and percentiles can be used as uncertainty estimation, which are calculated for the population of soil profiles linked to a soil type (Batjes, 2016). This estimation has some limitations because soil profiles are not taken probabilistically but based on their availability, especially for the global soil datasets. Uncertainty will be underestimated when the sample size is not big enough to represent a soil type. For digital soil mapping, uncertainty could be estimated by methods such as geostatistical methods and quantile regression forest (Vaysse and Lagacherie, 2017), which make sense of statistic. The accuracy of soil dataset derived by digital soil mapping are estimated by cross-validation, but it is not trivial for those derived by the linkage method due to the global scale, the support of the data and independent data (Stoorvogel et al.,

2017). Instead, some datasets, including WISE and GSDE, use some indictors such as linkage level of soil class and sample size to offer quality control information (Shangguan et al. 2014; Batjes, 2016). A simple way to compare the accuracy of datasets by both methods may be to use a global soil profile database as a validation dataset, though some of these profiles were used in deriving these datasets and questions will be raised.

The manuscript would benefit from a thorough English edit by a native speaker.

Reply: We will take a language service for the revised manuscript.

2. Specific comments L15-16: Rephrase this as e.g.: Soil is an important regulator of earth system processes, but remains one of the least well-described data layers in such models.

Reply: Modified as: Soil is an important regulator of earth system processes, but remains one of the least well-described data layers in Earth System Models (ESMs).

L17: Function as->provide

Reply: Modified.

L22: Abundant soil observations are not 'enough'; these should have been analysed according to comparable analytical methods and quality-assessed (which is seldom the case, see Batjes et al. 2017). What about the geographical distribution, or possible clustering, of the available (i.e. shared) soil profile data?

Reply: We changed the expression as 'with abundant, harmonized and quality controlled soil observations'. Corresponding contents are added accordingly. See the replies to related comments of the reviewer.

L24: By their nature, pedotransfer functions generally are not portable from one region to the other. Please add some discussion.

Reply: We add a sentence to the comments on Line 323.

L24-25: Speculative as written, provide some arguments for this.

Reply: See reply to comments on L451-452.

L27-28: What about uncertainty in the co-variates?

Reply: We put this as a part of the paragraph discussing uncertainty sources of the derive soil dataset

L35-36 / 45: You may consider the following reference here: http://dx.doi.org/10.1002/2015GB005239.

Reply: The reference was added. It is helpful to understand the role of soil information in ESMs.

L43: Remove available

Reply: Removed.

L45: How do you define 'better' here? Please clarify.

Reply: We changed the word to 'more realistic'. This is in the following citations, Brunke et al. (2016); Luo et al. (2016); Oleson et al. (2010). We added an example here: 'For example, Brunke et al., (2016) incorporated the depth to bedrock data in a land surface model using variable soil layers and instead of the previous constant depth.'

L47-48: Also other types of soil data, for example soil biology (see ref. line L35-36). See also discussion in https://doi.org/10.1111/gcb.13896.

Reply: We changed the sentence into 'ESMs require detailed information on the soil physical and, chemical and biological properties'.

L56: Useful to say that the range of soil data collected during a soil survey, will vary with scale and projected applications of the data (i.e. type of soil survey, routine versus surveys/studies aimed at answering specific user demands).

Reply: We added a sentence to say this: The range of soil data collected during a

soil survey, varies with scale, specifications of a country or a region, and projected applications of the data (i.e. type of soil surveys, routine versus specifically designed surveys). As a result, the availability of soil properties differs in different soil databases.

L72: How would you define reliable soil data? Remove from this sentence.

Reply: Removed

L76: Rather refer to measurements here.

Reply: Modified.

L87-88: Should add HWSD (FAO/IIASA/ISRIC/ISS-CAS/JRC, 2012) as reference for this type of 'traditional' approach.

Reply: Added

L93: usually not ready for ...! ...not appropriately scaled or formatted for ...

Reply: modified

L113-114...: representing main soil types in a landscape unit characterised by soil profiles considered representative for the main component soils of the respective mapping units.

Reply: Here we are describing two kinds of data from soil survey, i.e., soil map and soil profiles. So We modified the sentence as: soil polygon maps representing distribution ofmain soil types in a landscape unit and soil profiles with observations of soil properties which are considered representative for the main component soils of the respective mapping units.

L124: Rephrase this...: (FAO, 2003b, Zobler 1986) and these products are known to be outdated. The information on the initial SMW and DSMW has since been updated for large sections of the world in the HWSD product (FAO/IIASA/ISRIC/ISS-CAS/JRC, 2012), which has recently been revised in

WISE30sec (http://dx.doi.org/10.1016/j.geoderma.2016.01.034).

Reply: Added.

L124-125: Start new paragraph for the regional and national level data.

Reply: Modified.

L132: multiply –> multiple

Reply: Modified.

L133: soil properties are observed (e.g. site data) or measured (e.g. pH, sand, silt, clay content)

Reply: Modified.

L138-141: Important to mention here that data served through WoSIS have been standardised, with special attention for the description/comparability of soil analytical methods worldwide. See: http://dx.doi.org/10.17027/isric-wdcsoils.20180001. Also an important element for the discussion is that many countries, although having a large collection of soil profile data, are not yet sharing such data. See for example: https://doi.org/10.1016/j.grj.2017.06.001

Reply: Modified.

L141: The initial list of attributes corresponds with the GlobalSoilMap spec-ifications, with additional properties added/considered later in WoSIS (see http://dx.doi.org/10.17027/isric-wdcsoils.20180001).

Reply: Modified by adding the number of soil properties as follows: The soil pro-files database of World Soil Information Service (WoSIS) contains the most abun-dant profiles (about 118,400) from national and global databases including most of the databases mentioned below (Batjes, 2017), though only a selection of important soil properties (12) are included (Ribeiro et al., 2018).

L164: The linkage methods assigns a best-estimate for each soil property (and soil interval) under consideration to each component soil unit of a polygon (see e.g. HWSD). [see also 359-360].

Reply: modified as: Because the linkage method assigned only one value or a statistical distribution to a soil type in soil polygons (usually a polygon contains multiple soil types), the intra-polygonal spatial variation is not taken into account.

L171-173: For a more comprehensive review see also: http://dx.doi.org/10.1016/j.geoderma.2016.01.034 and http://dx.doi.org/10.1002/ldr.2656.

Reply: we added these two ref.

L178: FYI, WISE30sec considers seven layers up to 200 cm depth and 20 soil properties.

Reply: We added description of WISE30sec as one of the recent global datasets: WISE30sec is another improvement of HWSD incorporated more soil profiles with seven layers up to 200 cm depth and with uncertainty estimated by mean $\pm$ standard deviation. WISE30sec used the soil map from HWSD with minor corrections and climate zone maps as categorical covariate.

L201: Possibly, also mention the GSOC effort of the GSP here, see: https://doi.org/10.5194/soil-4-173-2018.

Reply: Added as: A third global soil mapping project is the Global SOC Map of the Global Soil Partnership, which focuses on country-specific soil organic carbon estimates (Guevara et al., 2018).

L205: . . . which is currently the most detailed, though not necessarily most accurate estimation of . . .

Reply: Modified.

L214: See also: Tifafi M, Guenet B and Hatté CCG 2018. Large differences in global and regional total soil carbon stock estimates based on SoilGrids, HWSD and NCSCD: Intercomparison and evaluation based on field data from USA, England, Wales and France. Global Biogeochemical Cycles, 42-56. http://dx.doi.org/10.1002/2017GB005678. Note: This paper is erroneously referred to as Marwa et al. 2018 in manuscript. This should be: Tifafi et al. 2018.

Reply: Corrected.

L214: Check if this is for 0-100 cm; likely these estimates are for 0-200 cm (see also recent sources mentioned above).

Reply: It is reported as 0-100 cm in the ref.

L224: Large sections of HWSDv1.2 still draw on the now outdated DSMW.

Reply: Modified as: Except GSDE, HWSD and STATSGO (Miller and White, 1998) for USA in Table 1, these datasets were derived from the Soil Map of the World (note that large sections of GSDE and HWSD still used this map as a base map because there are no available regional or national maps)

L295-296: See earlier comments.

Reply: We added WISE30sec in Table 2 and 3.

L299: WISE30sec presents estimations of uncertainty, unlike the HWSD and GSDE.

Reply: Modified as: Except WISE30sec, all these databases do not contain uncertainty estimation.

L300: Needs some discussion and references to publications on the subject.

Reply: we deleted this statement: Soilgrids is considered to be the most accurate one. Because there is not any evaluation of these four datasets, we added a discussion as: The accuracy of these datasets will need to be evaluated and compared for each

soil properties, especially for those frequently used in ESM, including sand, silt and clay content, coarse fragments, bulk density and organic carbon. Special attention should be paid to the data quality of each soil properties in the above datasets as their uncertainty and accuracy vary as discussed above. See also the following discussion.

L302: Larger number of soil properties for GSDE, but what about the accuracy of the predictions? (not given as indicated earlier).

Reply: Not only GSDE but also HWSD, WISE30sec do not provide a quantitative accuracy assessment. WISE30sec provides uncertainty estimation, and HWSD and GSDE could take similar way to estimate the uncertainty. But uncertainty estimation is different from accuracy assessment. As we discussed above, we may need further studies to evaluate them. Maybe use cross validation or independent soil profiles datasets to validate. But it seems like no one has done a cross validation for the datasets estimated by the linkage method like the digital soil mapping yet. GSDE did some quality assessment using some indicators like WISE, including linkage level of soil class, sample size, texture consideration, search radius and map unit level (see figure 6 of Shangguan et al., 2014). But it is only a reference of the accuracy and not straight forward for users, and most users may not even take a look at it. We add some discussions in the paragraph of uncertainty.

L303: Rephrase. ...SoilGrids products currently consider the list of attributes as defined by the GlobalSoilMap consortium.

Reply: modified: while Soilgrids currently contains ten primary soil properties defined by the GlobalSoilMap consortium.

L323: Most PTFs are not portable (i.e. locally or regionally validated).

Reply: we added: PTFs generally are not portable from one region to the other (i.e. locally or regionally validated).

L331-332: add database (word is missing in sentence)

Reply: modified. L360-361: ...component soil unit in most cases, and thus a one-to-many relationship exists between the SMU and the profile attributes of the respective soil units...

Reply: modified.

L397-398: Possibly, rephrase this sentence.

Reply: modified: However, some researches used the "aggregating after" method producing misleading results (Hiederer and Köchy, 2012).

L410: remove high from sentence

Reply: removed.

L441-442: Provide some justification (a sentence or two) for this statement.

Reply: added: because they provide spatial continuous estimations of soil properties using spatial prediction models with various soil-related covariates.

L451-452: Speculative as written. Please provide some evidence for this.

Reply: This issue is discussed extensively by Looy et al. 2017 at the end of section 7. For briefty, we added a sentence here instead of long discussions: because ensemble modeling carries a number of benefits and potential over the use of a single model (Looy et al., 2017).

For you reference, I copied the content from Looy et al. (2017) here: Another recent technique that has merits in this respect is ensemble modeling – i.e. the use of a number of models in combination. This technique is a natural part of weather and climate modeling today, yet it is less used in the prediction of soil properties [Baker and Ellison, 2008b]. Ensemble modeling carries a number of benefits and potential over the use of a single model. Models can differ in their theory and structure, but also in the information that they require. As a result, their sensitivity and scale of support may also differ. The use of ensemble modeling is easy to justify if it is difficult to determine which,

if any, single model may be superior to others. In ensemble modeling, the main aim is not to make the single model perfect, but to capture the trend that multiple models agree on. The ensemble will amplify trends that are common among models, while by-chance predictions will be softened. The outputs, therefore, can be interpreted – qualitatively or quantitatively - as a measure of uncertainty. In the context of integrated Earth system models, the represented complex processes – integrating physical and biochemical processes typically – can be covered by a number of models with strongly varying concept and structure. Here lies an opportunity to construct ensemble models entering different PTF-based parameterizations.

L460: and quantifying uncertainty in the predictions

Reply: Added

L461: 'need to gain popularity in ...'. Basically, the "proof of the pudding is in the eating".

Reply: We provided some examples at regional scales, which shows products by digital soil mapping improved climate modelling results (Kearney and Maino, 2018; Trinh et al., 2018). But no global studies have been taken to compare digital soil mapping products and linkage method products in ESMs yet, which we are doing now. So we changed this sentence to a more conservative one: the new generation soil datasets derived by digital soil mapping need to be tested in ESMs, and some regional studies have shown that these datasets provided better modelling results than products by the linkage method (Kearney and Maino, 2018; Trinh et al., 2018). Moreover, many studies from digital soil mapping have identified that soil maps are not very important to predict soil properties and are usually not used as a covariate in most studies (eg. Hengl et al., 2014; Viscarra Rossel et al., 2015; Arrouays et al., 2018). However, the linkage method usually takes soil map as the major covariate, which essentially affect the accuracy of the derived soil property maps, especially for areas without detailed soil maps.

L462: What I miss in this paper, is a discussion of the inherent uncertainty attached to using soil profile data coming from various sources. Often, little consideration is given to differences in analytical methods used for analysing e.g. soil organic carbon content worldwide (see Shangguang et al 2014, who consider this as 'a major imitation to their approach'). For a discussion of issues see e.g.: http://dx.doi.org/10.17027/isricwdcsoils.20180001

Reply: This was mentioned in L483-L484: Data compatibility of different analysis methods and different description protocols including soil classifications is also an important issue and data harmonization is necessary when the data are made available to public. Also, see the paragraph discussing uncertainty we added.

L463-464: More soil profiles is not necessarily the solution. More quality assessed data, analysed according to comparable analytical methods, are needed to support such efforts. Reference should be made to 'new' types of data as derived from proximal sensing (e.g. http://dx.doi.org/10.5194/soil-2017-36), and associated limitations. Reference, in this respect, could also be made to the GLOSOLAN effort, initiated by the GSP (http://www.fao.org/global-soilpartnership/resources/events/detail/en/c/1037455/) and work of GSP Pillar 5 towards harmonisation (http://www.fao.org/3/a-bs756e.pdf). Also, importantly, the geographical distribution and possible clustering of the shared soil profiles.

Reply: these are added: More quality assessed data, analysed according to comparable analytical methods, are needed to support such efforts. The harmonization of soil data is undertaking by the work of GSP Pillar 5 (Pillar 5 Working Group, 2017) and WoSIS (Batjes et al., 2017). Data derived from proximal sensing, although with higher uncertainty than traditional soil measurements, can be used in soil mapping (England and Viscarra Rossel, 2018). To avoid spatial extrapolation, soil profiles should have a good geographical coverage.

L471-475: True, but how many of these profiles are actually being shared for the greater benefit of the international community? See paper by Arrouays et al. 2017 for a discussion.

Reply: We added: Arrouays et al. (2017) reported that about 800,000 soil profiles have been rescued in the selected countries.

L479: Some reference to the ongoing work of the Global Soil Partnership, Pillars 4 and 5, is needed here.

Reply: Added: (Pillar four Working Group, 2014; Pillar 5 Working Group, 2017)

L948: Table 2 is not complete; 'recent' datasets not yet considered in the review should be added here ( http://dx.doi.org/10.1002/ldr.2656; http://dx.doi.org/10.1016/j.geoderma.2016.01.034 ). Idem for Table 3.

Reply: WISE30sec is added. The other one (Stoorvogel et al., 2017, which we cited in our paper) is more about proposing a new method which can improve HWSD results. 'The RMSD for S‐World was considerably smaller (2.1% SOC) than the RMSD for HWSDweighted (2.9% SOC) .'But this method has some limitation for soil properties with limited samples and for those having week relationship with covariates. We don't find the dataset available online. And in the paper, they only tested 6 soil properties. i) topsoil thickness (cm), ii) soil depth (cm), iii) soil organic carbon (SOC) content in the top 30 cm (%), iv) SOC content in the subsoil (30 to 120 cm) (%), v) clay content in the soil profile (%), and vi) sand content in the soil profile (%) . So we did not add this citation as a dataset for now. Meanwhile, I have written email to the author to check the availability

L952: Table 3. Change title to "Derived soil properties considered in three global soil datasets". Essentially, this is a simple enumeration of derived soil properties. However, the fact that many different analytical methods have been used to derive a given soil property (e.g. soil organic carbon Walkley & Black method or LECO total analyses) or which CEC (e.g. measured at 'field pH' or in a buffer-solution at 'pH7' or 'pH8') has been considered is not mentioned here (in a footer perhaps). In their study, Shangguan et al. (2014) rightly indicate that this has not been the case and indicate that they see this an important limitation. However, there are still no straightforward mechanisms

for harmonising the data (cf. GSP Pillar 5 and GLOSOLAN activities, as mentioned above).

Reply: title changed. We add a sentence in the footnote: It should be noted that many different analytical methods have been used to derive a given soil property, which is a major source of the dataset.

Potaasium –> Potassium

Reply: corrected

---

## Referee Comment (RC2) · Anonymous Referee #2 · 27 Dec 2018

A review of soil datasets available for Earth system modeling is extremely useful, given the wide application of ESMs in important projects such as the coupled model inter-comparison projects (CMIP) serving the IPCC reports, and in view of the challenges of observing soil properties covering the globe. However, the manuscript does in fact not fulfill what it promises in the title. It does not review datasets and compares them quantitatively (apart from selected maps in Fig. 1-2, but a systematic comparison is missing). Instead it discusses in length linkage and digital mapping methods, then how soil observational data in general can be incorporated in ESMs and what challenges arise. This is valuable *ancillary* information, and the manuscript summarizes a lot of important information on these topics. But the main purpose of the paper is missed. A careful review of available datasets needs to be added, which is of course a major

revision: there should be more than the 3 datasets in Tab. 3, unless justified that these 3 are special (for example it would be very illustrative to include all the currently used old datasets as well to know what a difference the new datasets might make). There should be a review also of other data than global maps, as needed e.g. for parameters. Most importantly, however, a quantitative comparison of at least key variables should be included, with useful statistical measure (maps, global mean and variability, latitudinal means, comparison against selected observational high-quality sites, ...). Ideally, model sensitivity simulations would be run, but this latter point is not essential.

A method for the review is missing, which leaves the reader in doubt whether he/she has been reading an opinion piece or a comprehensive review. Currently both the selection of mentioned datasets and the selection of ESMs is incomprehensive and not justified in its selection. For the models one could imagine to do a review of all TRENDY LSMs or of all CMIP5 (or even better CMIP6) ESMs and the datasets they are using. For the available datasets some objective criteria should be given as well, e.g. a list of criteria that datasets need to fulfill to be included in Tab. 3 (global, soil type and property x, y, z need to be included, ...).

The organization of the sections does not appear logical: Datasets and their usage in ESMs (Section 2) is very good, presenting PTFs as Section 3 promises in l. 105 is also very useful — but these PTFs are in fact never presented and compared, just discussed. Section 4 deals with data from the linkage method — why? Why not data from digital mapping as well? Section 5 deals with upscaling to the coarse ESM resolution. This is an important point, but there are many other challenges related to application of soil datasets in ESMs: One obstacle is that observations are not covering the soil depth as deeply as the ESMs and in other layer distributions. Another that soil observations are derived from present-day, which has confounding effects of both environmental changes (climate, CO2, nutrient deposition, ...) and historical land use changes. Would this affect soil thermal and other properties needed as input to ESMs? How should one evaluate ESMs — only for present-day then?

How should ESMs deal with observational uncertainty (see comment below)? I think what this paper needs to cover is (0) specifying what ESMs need, i.e. which spatial and temporal coverage, which variables (extending the list of parameters, initial state, evaluation/benchmarking in the introduction) (1) general methodology of deriving this soil information (mostly Sec. 2, PTFs would go in this section as well.) (2) comprehensive, quantitative comparison of available global soil datasets (largely missing) (3) discussion of existing challenges of data usage in ESMs, where one should come back to the list of usages in the introduction: evaluation data for example does not have to have global coverage. The upscaling would be one of several points here.

The paper is not very well written. First, the use of English language is incorrect or uncommon. Second, many expressions are not accurate. Just taking the first sentence as example: "Soil or pedosphere is a key component of Earth system, and plays an important role in the water, energy and carbon balances and biogeochemical processes." First, it should read "The soil or pedosphere is a key component of the Earth system, ..." (where "Earth" is correctly written in capitals, while it is not in the title...). Second, the carbon cycle is one example of biogeochemical processes, so it should read "... and *other* biogeochemical processes". I am not correcting any of these language and accuracy errors in the following because they are too numerous.

More detailed comments:

p. 1

* "Soil datasets function as model parameters": do the authors mean that model parameters can be derived from soil carbon maps? What parameters are they thinking of?

* "are preferred to those by the linkage method for ESMs": not understandable at this point in the manuscript - what is the "linkage method"?

* "to provide secondary soil parameters to ESMs": what are secondary soil parameters?

\* Generally, the abstract does not read like a review of datasets, but like a commentary on challenges of integrating soil carbon datasets in ESMs. As a reader I would have expected an abstract here of types of data, see general comment above.

p. 2

\* "However, soil dataset used in ESMs is not well updated nor well utilized yet.": This needs citation of which datasets are used and felt by the authors to be outdated.

\* l. 45-48: Kearney & Maino are one specific study for Australia for soil moisture using one new soil dataset. Using this as reference for the entire "Earth system" and for "will improve" in the future is a stretch. Better look for a couple of references and spell them out explicitly.

\* "could avoid the possibility of the non-linear singularity evolution of the modeling": this needs to be explained in one more sentence. Do the authors mean that models may have multiple equilibria?

p. 3

\* "for multiply layers rather than a global constant": This mixes up vertical resolution (-> layers) and horizontal resolution (-> global constant). Be more explicit in your description.

\* Is "linkage method" really the proper technical term here? It seems to me it is used in the literature rather for remapping than for linking soil observations to environmental variables. The paper would benefit from a clearer overview of technical terms and methods, if it is meant to serve as a review.

\* paragraph starting l. 93: Vector to raster conversion and remapping to a different resolution are certainly not the biggest or at least not the only obstacles to including soil datasets in models. That models need different variables than those directly observable

or that observational datasets cover only a certain depth, which most often is different from the one ESMs cover, are examples of other important challenges. Overall, I feel the sections internally should be a bit better structured, with one topic being covered comprehensively by one paragraph.

* "Two kinds of soil data are generated from soil surveys: soil polygon maps representing distribution of soil types and soil profiles with observations of soil properties. ESMs usually require the spatial distribution of soil properties, or soil property maps rather than soil classification information.": It is unclear how the information of the two sentences relates. Would this be correct: "Two kinds of soil data are generated from soil surveys: a classification of soil type (usually in the form of polygon maps) and soil profiles with observations of soil properties. ESMs usually require the spatial distribution of soil properties (soil property maps) rather than a classification of soil type." If so please always use the same term for the same information.

p. 4

* "Soil maps show the geographical distribution of soil types,": I think this is too general, the term "soil map" is not a technical, well specified term. Rather speak explicitly of "soil type maps" to distinguish it from maps of soil properties.

* l. 153 ff linkage method: this is a useful description, but hard to read for non-experts. Please improve the clarity of the text. For example:

* my understanding is that pedotransfer functions map well-observable to less-well-observable properties, but here it sounds as if the PTFs are needed to link site-level (profile) observations of soil properties to soil type maps.

* "The criteria used in the linkage could be one or many factors as following [. . .] and so on": this is very vague. Which type of criteria is this: soil physical and chemical properties?

* "Each soil type is represented by one or a group of soil profiles that meet the criteria,

and usually the median or mean value of a soil property is assigned to the soil type.":
Criteria and properties are mixed up here. Isn't it choosing one (or several) property
as criteria, then mapping the rest?

* l. 165-172: how do these references relate to the examples of "major soil maps" in
the introduction?

* l. 188 ff: Again, please add clarity. The difference between linkage method and digital
soil mapping is not just that the first has the same values across a polygon, but also
in what information is used as criteria for mapping: the digital soil mapping uses envi-
ronmental information, not just physical and chemical properties — if I understood it
correctly.

p. 5

* "purity of soil map units is likely to be around 50 to 65%": which statistical measure is
meant by "purity"?

p. 6

* Why is IGBP-DIS mentioned here the first time? It should have been mentioned
under the linkage or the digital mapping methods (depending on what method is used)
before.

* "soil organic carbon stocks at 1m depth": is it meant "carbon stocks down to a depth
of 1m"?

Fig. 1: remove superfluous information that costs the reader time to read and hides
the differences between the panels (the datasets): since the legend is the same for all
sand (clay) panels it does not have to be repeated; same for "sand (clay) at 0-30cm
(%)", which is even stated in the caption. "Longitude" (typo!) and "latitude" are also
superfluous information.

Fig. 2: Same comment as for Fig. 1. "s" missing in soilgrids.Why is IGBP not included

here as well? A more useful information for modelers would be the total carbon content down to a certain depth rather than units of g/kg.

p. 6 cont'd

* "several most popular ESMs": give objective criteria for "popular"

* l. 227-229: Again, it should be stated in how far the new datasets are superior over previous datasets.

* l. 231: "This was started..." sounds a bit like advertisement and subjective, certainly other groups have been working on this to some extent for a long time as well. Reformulate more neutrally?

* l. 245-253: What is the purpose of these references? Only if they prove model results have improved by the usage of the new soil map is it useful to cite them here.

Tab. 1: please fix typos (inconsistent punctuation and capitalization). Add version numbers to LSMs, as usage of soil information may change between versions. Not sure the references are always correct, e.g. LeQuere et al., ESDD 2018 a and b ("Global carbon budget 2017" and "2018", resp) state Reick or Mauritsen as JSBACH references, not Giorgetta.

p. 7

* l. 299 ff: if there are no uncertainty estimates, how can you judge soilgrids to be the most accurate one?

* l. 305: not all models apply PTFs, some directly require these less observable variables as input, as you show in Tab. 1

p. 9

* l. 359: The methods have been introduced before, so technical terms like "SMU" should have been introduced in these earlier chapters.

* l. 365: A problem of using subgrid soil information is that ES modelers do not know how to map them with land use information, which is also subgrid level. This may be the more fundamental obstacle than the computational issues that are mentioned.

p. 11

* "The temporal variation of global soil is quite challenging due to lack of data.": the aspect of temporal changes has not been addressed before and seems out of place in the summary.

* "Soil image fusion is also needed to merge the local and global soil maps.": What is soil image fusion? Don't bring new methods in the summary section...

* " Uncertainty estimation should be included in the soil datasets developed in the future.": of course uncertainty estimates build trust in an observational dataset. But how do the authors recommend should ESMs use such uncertainty estimates other than as criterion for which dataset to choose in the first place? Running multiple simulations combining upper and lower bounds in all possible combinations is too expensive...

* "The gap between soil data existence and data availability is huge": Reads awkward. Better "The gap between the amount of data that has been taken in surveys and the amount of data freely available is large." p. 12

* l. 482 "like many other data": Too general a statement, remove. l. 481 "... which has the most...": how do you know? Add reference or justify in other ways

* Arbitrary last sentence. l. 465 already mentions the subgrid issue in ESMs. Is there no more general conclusion that can be given? Otherwise just delete the last paragraph and end with the more "outlook"-like previous paragraph.

---

## Author Comment (AC2) · 24 Jan 2019

1. General comments Comment: A review of soil datasets available for Earth system modeling is extremely useful, given the wide application of ESMs in important projects such as the coupled model intercomparison projects (CMIP) serving the IPCC reports, and in view of the challenges of observing soil properties covering the globe. However, the manuscript does in fact not fulfill what it promises in the title. It does not review datasets and compares them quantitatively (apart from selected maps in Fig. 1-2, but a systematic comparison is missing). Instead it discusses in length linkage and digital mapping methods, then how soil observational data in general can be incorporated in ESMs and what challenges arise. This is valuable *ancillary* information, and the manuscript summarizes a lot of important information on these topics. But the main

purpose of the paper is missed. A careful review of available datasets needs to be added, which is of course a major revision: there should be more than the 3 datasets in Tab. 3, unless justified that these 3 are special (for example it would be very illustrative to include all the currently used old datasets as well to know what a difference the new datasets might make). There should be a review also of other data than global maps, as needed e.g. for parameters. Most importantly, however, a quantitative comparison of at least key variables should be included, with useful statistical measure (maps, global mean and variability, latitudinal means, comparison against selected observational high-quality sites, ...). Ideally, model sensitivity simulations would be run, but this latter point is not essential.

Reply: The purpose of the view is to offer insights to both soil data developer and ESM users. So we discussed contents may be interesting to both sides. We agree that a systematic quantitative comparison is a very important aspect this review should cover. So we will compare a selection of global soil data sets with a focus on the most recent developed ones. We will use one or two 'old' data sets which are used in ESMs in the comparison. It is not necessary to compare all the old data sets because they are based limited and outdated source data. All the old soil data are based on the FAO soil map and no more than 5,800 soil profiles. We are sure that they have poor quality. That is why we highly recommend the most recent developed data sets even without a quantitative comparison. We can see that the newly developed soil data in fig.1-2 have some major differences. It is valuable to compare them even though they may not be so comparable, because the datasets developed by the linkage method give the mean value of a soil class while the datasets developed by digital soil mapping simulated the soil as a continuum (though in nature soil may change abruptly). Nevertheless, we plan to use site observations of WOSISS to evaluate the soil data sets, though these observations are used or partly used in the development of global distribution of soil data. We will compare the key variables used in ESM with useful statistical measure as the review recommended. However, model sensitivity simulations will not be done in this reviews and need to be done in other studies. This review focuses on the global

soil property maps in ESMs. We will not extend the content to other data including parameters which is a different topic but valuable. As we mentioned in the manuscript, variables such as soil temperature and To avoid misunderstanding, we will change the title to 'a review on the global soil property maps for earth system model' and modify the corresponding expression in the manuscript. However, we will use the term soil datasets for brevity.

Comment: A method for the review is missing, which leaves the reader in doubt whether he/she has been reading an opinion piece or a comprehensive review. Currently both the selection of mentioned datasets and the selection of ESMs is incomprehensive and not justified in its selection. For the models one could imagine to do a review of all TRENDY LSMs or of all CMIP5 (or even better CMIP6) ESMs and the datasets they are using. For the available datasets some objective criteria should be given as well, e.g. a list of criteria that datasets need to fulfill to be included in Tab. 3 (global, soil type and property x, y, z need to be included,...)

Reply: We will describe the selection of mentioned datasets and the selection of ESMs. As we mentioned in the above reply, datasets are chosen according to their data quality and developing time. In addition, the datasets should be freely available. Our currently list of ESMs covers the major LSMs but not all of them because a complete list will be too lengthy. We will check the list of TRENDY and CMIP5 to see if they used different soil datasets as we are only interested in the soil data rather than itself.

Comment: The organization of the sections does not appear logical: Datasets and their usage in ESMs (Section 2) is very good, presenting PTFs as Section 3 promises in l. 105 is also very useful but these PTFs are in fact never presented and compared, just discussed. Section 4 deals with data from the linkage method. why? Why not data from digital mapping as well? Section 5 deals with upscaling to the coarse ESM resolution. This is an important point, but there are many other challenges related to application of soil datasets in ESMs: One obstacle is that observations are not covering the soil depth as deeply as the ESMs and in other layer distributions. Another
that soil observations are derived from present-day, which has confounding effects of both environmental changes (climate, CO2, nutrient deposition, . . .) and historical land use changes. Would this affect soil thermal and other properties needed as input to ESMs? How should one evaluate ESMs only for present-day then?

Reply: PTFs is not the major focus of this review while there is a very good review on PTF in ESMs which we cited as Looy et al., (2017). Section 4 do not discuss data from digital mapping because it does not have the aggregating problem like the data by the linkage method. Data by the linkage method are derived for each soil map unit and data by the linkage method are derived for each grid. We will add a sentence to clarify this at the beginning of this section. We agree that we should also discuss the lack of deep soil data and the changing of soil properties. For the lack of deep soil data, we do not have very good solution other than extrapolate the values based on the observations of shallower layers. There is not any global soil property map in time-series because we do not have available data. All the available soil observations in the last decades are used in the development of soil property maps without considering the changing environment. So these datasets should be considered as an average state. The effect of environmental change is the topic of quantitative modeling of soil forming processes (soil landscape and pedogenic models). Soil properties are changing but we are now taken it as static in ESMs. As some ESMs already simulate the soil carbon, this may be considered in PTF used to estimate soil hydraulic and thermal parameters. Other soil properties affecting soil hydraulic and thermal parameters include soil texture, bulk density, soil structure and so on, but the change is relatively slow. If we need to simulate the change of soil properties, a coupling of ESMs and soil landscape and pedogenic models will be needed. Otherwise, we need to predict the soil properties in the future using soil landscape and pedogenic models which are small scale models and has high uncertainty. This prediction may also be done by digital soil mapping taken the changing (especially for the future) climate and land use as covariates, which I think is the most feasible one to do.

Comment: How should ESMs deal with observational uncertainty (see comment below)?

Reply: See the reply to the specific comment.

Comment: I think what this paper needs to cover is (0) specifying what ESMs need, i.e. which spatial and temporal coverage, which variables (extending the list of parameters, initial state, evaluation/benchmarking in the introduction) (1) general methodology of deriving this soil information (mostly Sec. 2, PTFs would go in this section as well.) (2) comprehensive, quantitative comparison of available global soil datasets (largely missing) (3) discussion of existing challenges of data usage in ESMs, where one should come back to the list of usages in the introduction: evaluation data for example does not have to have global coverage. The upscaling would be one of several points here.

Reply: We will reorganize this manuscript as the reviewer recommended.

Comment: The paper is not very well written. First, the use of English language is incorrect or uncommon. Second, many expressions are not accurate. Just taking the first sentence as example: "Soil or pedosphere is a key component of Earth system, and plays an important role in the water, energy and carbon balances and biogeochemical processes."First, it should read "The soil or pedosphere is a key component of the Earth system, . . ." (where "Earth" is correctly written in capitals, while it is not in the title: : :). Second, the carbon cycle is one example of biogeochemical processes, so it should read " and *other* biogeochemical processes". I am not correcting any of these language and accuracy errors in the following because they are too numerous. Reply: Thanks for pointing out these errors. We will revise this manuscript and take a language service.

More detailed comments: p. 1 Comment: * "Soil datasets function as model parameters": do the authors mean that model parameters can be derived from soil carbon maps? What parameters are they thinking of?

Reply: we corrected the expression to model inputs. This is the major usage of soil property maps in ESMs (table 1).

Comment: * "are preferred to those by the linkage method for ESMs": not understandable at this point in the manuscript - what is the "linkage method"?

Reply: we modified as "taxotransfer rule-based method', which may be a more understandable terminology.

Comment: * "to provide secondary soil parameters to ESMs": what are secondary soil parameters?

Reply: we modified it to "derived soil properties", which includes soil hydraulic, thermal and biogeochemical parameters. And we explained this when "secondary soil parameter" first appear in the manuscript.

Comment: Generally, the abstract does not read like a review of datasets, but like a commentary on challenges of integrating soil carbon datasets in ESMs. As a reader I would have expected an abstract here of types of data, see general comment above.

Reply: we will revise the abstract adding the related contents.

p. 2 Comment: * "However, soil dataset used in ESMs is not well updated nor well utilized yet.": This needs citation of which datasets are used and felt by the authors to be outdated.

Reply: To make this more objective, we added some citation from FAO and globalsoilmap (a community joint effort project). We have explained this in section 2.2: Except GSDE, HWSD and STATSGO (Miller and White, 1998) for USA in Table 1, these datasets were derived from the Soil Map of the World (note that large sections of GSDE and HWSD still used this map as a base map because there are no available regional or national maps) (FAO, 1971-1981) and limited soil profile data (no more than 5,800 profiles), which gained popularity because its simplicity and ease of use. But these are outdated and should no longer be used because much better soil information as introduced in Section 2.1 can be incorporated (Sanchez et al., 2009; FAO/IIASA/ISRIC/ISS-CAS/JRC, 2012).

Comment: * l. 45-48: Kearney & Maino are one specific study for Australia for soil moisture using one new soil dataset. Using this as reference for the entire "Earth system" and for "will improve" in the future is a stretch. Better look for a couple of references and spell them out explicitly.

Reply: This is only an example. We added more citations here: (eg. Livneh et al., 2015; Dy and Fung, 2016; Kearney and Maino, 2018). More examples are given with brief description in section 2.2.

Comment: * "could avoid the possibility of the non-linear singularity evolution of the modeling": this needs to be explained in one more sentence. Do the authors mean that models may have multiple equilibria?

Reply: Yes, it means models may have multiple equilibria. And we also added a sentence: The setting of initial nutrient stocks is a major factor leading to model-to-model variation in the simulation (Todd-Brown et al., 2014).

p. 3 Comment: * "for multiply layers rather than a global constant": This mixes up vertical resolution (-> layers) and horizontal resolution (-> global constant). Be more explicit in your description.

Reply: we modified it as: As a result, ESMs usually incorporate soil property maps (i.e., horizontal spatial distribution) for multiply layers rather than a global constant or a single layer.

Comment: * Is "linkage method" really the proper technical term here? It seems to me it is used in the literature rather for remapping than for linking soil observations to environmental variables. The paper would benefit from a clearer overview of technical terms and methods, if it is meant to serve as a review.

Reply: We used this term for brevity. But it may be misleading if readers are not
familiar with soil mapping. So we also used the other term taxotransfer rule-based method. We added this term when it first appears in the manuscript: The traditional way (i.e., the linkage method, also called taxotransfer rule-based method) is to link soil profiles and soil mapping units on soil type maps, sometimes with ancillary maps such as topography and land use (Batjes, 2003; FAO/IIASA/ISRIC/ISS-CAS/JRC, 2012).

Comment: * paragraph starting l. 93: Vector to raster conversion and remapping to a different resolution are certainly not the biggest or at least not the only obstacles to including soil datasets in models. That models need different variables than those directly observable or that observational datasets cover only a certain depth, which most often is different from the one ESMs cover, are examples of other important challenges. Overall, I feel the sections internally should be a bit better structured, with one topic being covered comprehensively by one paragraph.

Reply: This paragraph served as a brief introduction of the obstacles to including soil datasets in models and detailed description will be given in the later sections. That models need different variables than those directly observable is related to the PTF development. So we did not put here. We added the challenge of incorporating uncertainty in ESMs here. We modified the paragraph as follows: There are many challenges related to application of soil datasets in ESMs. (1) First, soil datasets are usually not appropriated scaled or formatted for the use of ESMs and some upscaling issues, which is the most frequently encountered, need to be addressed. The soil datasets produced by the linkage methods are polygon-based and need to be converted to fit the grid-based ESMs. This conversion can be done by either subgrid method or spatial aggregation. The up-to-date soil data are provided at a resolution of 1km or finer, while the LSMs are mostly ran at a coarser resolution. So upscaling of soil data is necessary before it can be used by ESMs. Proper upscaling methods need to be chosen carefully to minimize uncertainty in the modeling results introduced by them (Hoffmann and Christian Biernath, 2016; Kuhnert et al., 2017). (2) Second, all the current global soil datasets represent the average state of last decades, but soil properties change over

time. Soil landscape and pedogenic models are developed to simulate soil forming processes and soil property changes, which can be incorporated into ESMs. The prediction of changing soil properties can be also done by digital soil mapping taken the changing climate and land use as covariates. (3) . (4) Last but not least, the depth of soil observations in soil survey are usually less than 2 m, but the ESMs usually covers much deeper. A pragmatic solution to this issue is to extrapolate the values based on the observations of shallower layers.

Comment: * "Two kinds of soil data are generated from soil surveys: soil polygon maps representing distribution of soil types and soil profiles with observations of soil properties. ESMs usually require the spatial distribution of soil properties, or soil property maps rather than soil classification information.": It is unclear how the information of the two sentences relates. Would this be correct: "Two kinds of soil data are generated from soil surveys: a classification of soil type (usually in the form of polygon maps) and soil profiles with observations of soil properties. ESMs usually require the spatial distribution of soil properties (soil property maps) rather than a classification of soil type." If so please always use the same term for the same information.

Reply: you are right. we modified as follows: Two kinds of soil data are generated from soil surveys: a map (usually in the form of polygon maps) representing main soil types in a landscape unit and soil profiles with observations of soil properties which are considered representative for the main component soils of the respective mapping units. ESMs usually require the spatial distribution of soil properties (i.e., soil property maps) rather than information about soil types. Two kinds of methods, i.e. the linkage method and the digital soil mapping method, are used to derive soil property maps.

p. 4 Comment: * "Soil maps show the geographical distribution of soil types,": I think this is too general, the term "soil map" is not a technical, well specified term. Rather speak explicitly of "soil type maps" to distinguish it from maps of soil properties.

Reply: In soil science, if it is not clarified, soil map refers to soil type map. To clarify

this, we modified to: Soil maps (the term soil map refers to soil type map in this paper) show the geographical distribution of soil types

Comment: * l. 153 ff linkage method: this is a useful description, but hard to read for non-experts. Please improve the clarity of the text. For example: * my understanding is that pedotransfer functions map well-observable to less-wellobservable properties, but here it sounds as if the PTFs are needed to link site-level (profile) observations of soil properties to soil type maps.

Reply: Sorry for the description leading to the misunderstanding. Pedotransfer here has nothing to do with Pedotransfer functions discussed in the late section. We added some notification here: The linkage method (called the taxotransfer rule-based method) is to link soil mapping units or soil polygons and soil profiles according to taxonomy-based pedotransfer (taxotransfer in short, note that pedotransfer here does mean pedotransfer functions which is a different thing) rules (Batjes, 2003).

Comment: * "The criteria used in the linkage could be one or many factors as following [. . .] and so on": this is very vague. Which type of criteria is this: soil physical and chemical properties?

Reply: this is related to the above comment. This is the criteria for linking soil map and soil profiles with all soil properties together.

Comment: * "Each soil type is represented by one or a group of soil profiles that meet the criteria, and usually the median or mean value of a soil property is assigned to the soil type.": Criteria and properties are mixed up here. Isn't it choosing one (or several) property as criteria, then mapping the rest?

Reply: this is related to the above comment. We hope the readers will be clear with the above clarification.

Comment: * l. 165-172: how do these references relate to the examples of "major soil maps" in the introduction?

Reply: these references include both soil type maps and soil property maps, while "major soil maps" in section 2.1 (not the introduction) refers to soil type map only.

Comment: * l. 188 ff: Again, please add clarity. The difference between linkage method and digital soil mapping is not just that the first has the same values across a polygon, but also in what information is used as criteria for mapping: the digital soil mapping uses environmental information, not just physical and chemical properties if I understood it correctly.

Reply: This is also related to the misunderstanding of the term pedotranfer.

p. 5 Comment: * "purity of soil map units is likely to be around 50 to 65%": which statistical measure is meant by "purity"?

Reply: This is a term used in soil science, which means the percentage of the dominant soil type in a soil map unit. Details can be found in the following website: https://esdac.jrc.ec.europa.eu/ESDB_Archive/ESDBv2/esdb/sgdbe/metadata/purity_maps/purity.htm.

p. 6 Comment: * Why is IGBP-DIS mentioned here the first time? It should have been mentioned under the linkage or the digital mapping methods (depending on what method is used) before.

Reply: IGBP-DIS is listed in Table 1. It is produced by the linkage method. We added IGBP-DIS under the linkage method.

Comment: * "soil organic carbon stocks at 1m depth": is it meant "carbon stocks down to a depth of 1m"?

Reply: Yes, we corrected it.

Comment: Fig. 1: remove superfluous information that costs the reader time to read and hides the differences between the panels (the datasets): since the legend is the same for all sand (clay) panels it does not have to be repeated; same for "sand (clay) at 0-30cm (%)", which is even stated in the caption. "Longitude" (typo!) and "latitude"

are also superfluous information.

Reply we will remove superfluous information.

Comment: Fig. 2: Same comment as for Fig. 1. "s" missing in soilgrids. Why is IGBP not included here as well? A more useful information for modelers would be the total carbon content down to a certain depth rather than units of g/kg.

Reply: we will remove superfluous information and add IGBP and total carbon content.

p. 6 cont'd Comment: * "several most popular ESMs": give objective criteria for "popular"

Reply: Here we do not have objective criteria. The list will be extended according CMIP5 and TRENDY, but it is necessary to show all of them, only those with different soil datasets.

Comment: * l. 227-229: Again, it should be stated in how far the new datasets are superior over previous datasets.

Reply: This will need quantitatively assessment in the revision.

Comment: * l. 231: "This was started: : :" sounds a bit like advertisement and subjective, certainly other groups have been working on this to some extent for a long time as well. Reformulate more neutrally?

Reply: we modified it to: The Land-Atmosphere Interaction Research Group at Beijing Normal University (BNU, now at Sun Yat-sen University) has put much efforts on this topic.

Comment: * l. 245-253: What is the purpose of these references? Only if they prove model results have improved by the usage of the new soil map is it useful to cite them here.

Reply: These citations are showing the application of the new soil datasets in ESMs,

which is stated in the first sentence of the paragraph: In recent years, efforts were taken to improve the soil data condition in ESMs.

Comment: Tab. 1: please fix typos (inconsistent punctuation and capitalization). Add version numbers to LSMs, as usage of soil information may change between versions. Not sure the references are always correct, e.g. LeQuere et al., ESDD 2018 a and b ("Global carbon budget 2017" and "2018", resp) state Reick or Mauritsen as JSBACH references, not Giorgetta.

Reply: we will check this table and make corrections.

p. 7 Comment: * l. 299 ff: if there are no uncertainty estimates, how can you judge soilgrids to be the most accurate one?

Reply: this is depended on the source data and method they used, and will need quantitative comparison in the revision.

Comment: * l. 305: not all models apply PTFs, some directly require these less observable variables as input, as you show in Tab. 1

Reply: It is true. But these variables are also derived by PTFS. To be precise, we modified it to: Earth system modellers have employed different pedotransfer functions (PTFs) to estimate soil hydraulic parameters (SHP), soil thermal parameters (STP), and biogeochemical parameters (Looy et al., 2017;Dai et al., 2013) or used these parameters as model inputs.

p. 9 Comment: * l. 359: The methods have been introduced before, so technical terms like "SMU"should have been introduced in these earlier chapters.

Reply: We introduced it in describing soil type maps: There are many soil mapping units (SMU) in a soil map and a SMU is composed of more than one component (i.e. soil type) in most cases.

Comment: * l. 365: A problem of using subgrid soil information is that ES modelers

do not know how to map them with land use information, which is also subgrid level. This may be the more fundamental obstacle than the computational issues that are mentioned.

Reply: Yes, this will increase the model complexity, too. A possible solution of mapping them with PFT or land cover is here: classify soil according soil properties and get a number of defined soil classes (SFT, n classes) like PFT (m classes); overlay the defined soil classes with PFT and get n by m combinations assuming PFT and SFT are independent.

p. 11 Comment: * "The temporal variation of global soil is quite challenging due to lack of data.": the aspect of temporal changes has not been addressed before and seems out of place in the summary.

Reply: We will add a section about this.

Comment: * "Soil image fusion is also needed to merge the local and global soil maps.": What is soil image fusion? Don't bring new methods in the summary section...

Reply: This is about outlook instead of summary. Soil image fusion is proposed by Hengl et al. (2017), which consider local and global soil maps as components of soil variation for ensemble predictions. We modified this to: Soil image fusion is also needed to merge the local and global soil maps, which consider them as components of soil variation for ensemble predictions (Hengl et al., 2017). A system for automated soil image fusion might take years before an operational system for global soil data fusion is fully functional.

Comment: * " Uncertainty estimation should be included in the soil datasets developed in the future.": of course uncertainty estimates build trust in an observational dataset. But how do the authors recommend should ESMs use such uncertainty estimates other than as criterion for which dataset to choose in the first place? Running multiple simulations combining upper and lower bounds in all possible combinations is

too expensive...

Reply: We agree that running ESMs with all possible combinations is too expensive. An alternative to quantify effects of the uncertainty of soil properties on ESMs may be to use adaptive surrogate modeling based on statistical regression and machine learning which costs much lower computing time (Gong et al., 2015; Li et al. 2018). We will discuss this using a section.

Comment: * "The gap between soil data existence and data availability is huge": Reads awkward. Better "The gap between the amount of data that has been taken in surveys and the amount of data freely available is large."

Reply: we modified this to: The gap between the amount of data that has been taken in soil surveys and the amount of data freely available is large.

p. 12 Comment: * l. 482 "like many other data": Too general a statement, remove.

Reply: removed

Comment: l. 481 ": : : which has the most: : :": how do you know? Add reference or justify in other ways

Reply: We added a citation here (Hengl et al. 2017).

Comment: * Arbitrary last sentence. l. 465 already mentions the subgrid issue in ESMs. Is there no more general conclusion that can be given? Otherwise just delete the last paragraph and end with the more "outlook"-like previous paragraph.

Reply: the last paragraph is deleted.

---

## Author Response (AR1)

| 1  | Replay to editor's comments                                                                        |
|----|----------------------------------------------------------------------------------------------------|
| 2  | Your manuscript has been thoroughly reviewed by two peer experts in the field. Both consider       |
| 3  | your manuscript as useful and valuable at least in parts. Since it is a review, a certain degree   |
| 4  | of comprehensiveness is expected and has not been achieved yet with several global soil            |
| 5  | datasets not being included as pointed out by both reviewers.                                      |
| 6  | Reply: Thanks for pointing out the value and the weakness of the paper. We added some              |
| 7  | global soil datasets for discussion (See table 1 and table 2).                                     |
| 8  |                                                                                                    |
| 9  | A systematic approach is required with criteria to be defined for the selection or exclusion of    |
| 10 | maps in this review. If the focus of the paper is on global soil maps also Table 1 maybe revised   |
| 11 | to start with the soil maps and not with the ESMs.                                                 |
| 12 | Reply: The list of ESMs includes all the LSMs used in CMIP5 except two models without              |
| 13 | information of soil data used. For the new soil datasets developed in recent years, datasets       |
| 14 | are selected because they have developed with relatively good data sources and are freely          |
| 15 | available. Old versions of these datasets are not shown here. We have revised Table 1 to start     |
| 16 | with the soil maps.                                                                                |
| 17 |                                                                                                    |
| 18 | I also suggest to include more aspects of the quality of the maps as proposed by the first         |
| 19 | reviewer in order to evaluate (e.g. with global or latitudinal means) and where possible to        |
| 20 | compare the maps.                                                                                  |
| 21 | Reply: We have evaluated four global soil datasets using soil profiles of WoSIS. Details are in    |
| 22 | section 3.                                                                                         |
| 23 |                                                                                                    |
| 24 |                                                                                                    |
| 25 | Reply to reviewer 1                                                                                |
| 26 | 1. General comments                                                                                |
| 27 |                                                                                                    |
| 28 |                                                                                                    |
| 29 | This is a timely review of global scale soil data sets that are used to underpin Earth System      |
| 30 | Models, and the still numerous, associated uncertainties. Such soil data sets have evolved         |
| 31 | greatly since the coarse 1-degree resolution map generalised by Zobler (1986) resulting in a       |
| 32 | new generation of digital soil maps, and the underpinning soil point data sets and/or              |
| 33 | covariates layers. That being said, I have a number of queries and comments. For example,          |
| 34 | rather little attention is given to difficulties associated with the limited comparability of soil |
| 35 | analytical data worldwide and uncertainty propagation. Further, several recent global soil         |
| 36 | databases of possible interest for ESM modelling have not been considered in the review and        |
| 37 | discussion.                                                                                        |
| 38 |                                                                                                    |
| 39 |                                                                                                    |
| 40 | Reply: Thanks for your valuable and detailed comments which help us a lot in improving our         |
| 41 | manuscript. The reviewer's comments have been addressed one by one in the following                |
| 42 | replies. This review was done from the perspective of ESMs and its users rather than that of       |
| 43 | soil data development. So we omitted some details about data development and associate             |
|    | uncertainty as pointed out by the reviewer. But it is useful to discuss these details for data     |

45 development. We are aware that many uncertainty sources exist in the derived soil dataset, 46 which need attentions to be paid by ESM applications. After considering the comments of 47 the reviewer (including the comparability of soil analytical data), we added a paragraph 48 concentrating on the uncertainty sources and uncertainty estimation (including spatial 49 uncertainty estimation and accuracy assessment) of soil data, which is a more comprehensive 50 summary on the uncertainty of soil data. And the comparability of soil analytical data, the 51 covariates uncertainty and others are discussed in this paragraph. Some contents about the 52 uncertainty in the original manuscript were also moved to this paragraph. As we see in the 53 literature, ESMs usually do not consider much about uncertainty or even data quality of soil 54 properties, which is not a good situation. ESM users should be more concerned about the 55 uncertainty estimation rather than the uncertainty sources, while data developers need to 56 know both aspects well. Further, we added more global soil databases as suggested by the 57 reviewer (see the reply to table 2 and 3).

58

59 Here is the uncertainty paragraph we added:

60

61 Because soil property maps are derived products based on soil measurements of soil profiles 62 (point observations) and spatial continuous covariates (including soil maps), it is necessary to 63 discuss the uncertainty sources, spatial uncertainty estimation and accuracy assessment of 64 these derived data (the last two are different aspects of uncertainty estimation). More 65 attention should be paid to this issue in ESM applications instead of taking soil property maps 66 as observations without error. There are various uncertainty sources in deriving soil property 67 maps, including uncertainty from soil maps, soil measurements, soil-related covariates and 68 the linkage method itself (Shangguan et al., 2012; Batjes, 2016; Stoorvogel et al., 2017). The 69 following may not be the complete list of uncertainty but the major ones. The uncertainty of 70 soil maps is a major source of global dataset derived by the linkage methods. For these 71 dataset, large sections of the world are drawn on the coarse FAO SMW map and the purity 72 website for the of soil maps (referring to the following definition: 73 https://esdac.jrc.ec.europa.eu/ESDB Archive/ESDBv2/esdb/sqdbe/metadata/purity maps/pu 74 rity.htm) is likely to be around 50 to 65% (Landon, 1991). Another important source of 75 uncertainty is the limited comparability of different analytical methods of a given soil property 76 in using soil profiles coming from various sources. A weak correlation or even a negative 77 correlation was found between different analytical methods, though strong positive 78 correlation revealed in most cases (McLellan et al. 2013). Both datasets by the linkage method 79 and those by digital soil mapping suffer this uncertainty. Though there are no straightforward 80 mechanisms to harmonize the data, efforts are undertaken to address this issue and provide 81 quality assess (Batjes, 2017; Pillar 5 Working Group, 2017). Another source of uncertainty 82 comes from the geographic and taxonomic distribution of soil profiles, especially for the 83 under-represented areas and soils (Batjes, 2016). The fourth source of uncertainty is from the 84 linkage method itself. It does not represent the intra-polygon spatial variation and usually do 85 not consider soil related covariates explicitly like digital soil mapping, though there are cases 86 where climate and topography are considered and Stoorvogel et al. (2017) proposed a 87 methodology to incorporate landscape properties in the linkage method. Finally, uncertainty 88 from the covariates is minor because spatial prediction models such as machining learning in

89 digital soil mapping can reduce its influences (Hengl et al., 2014), though a more 90 comprehensive list of covariates with higher resolution and accuracy will improve the 91 predicted soil property maps. Spatial uncertainty is estimated by different methods for the 92 linkage method and digital soil mapping methods. For the linkage method, statistics such as 93 standard derivation and percentiles can be used as spatial uncertainty estimation, which are 94 calculated for the population of soil profiles linked to a soil type or a land unit (Batjes, 2016). 95 This estimation has some limitations because soil profiles are not taken probabilistically but 96 based on their availability, especially for the global soil datasets. Uncertainty will be 97 underestimated when the sample size is not big enough to represent a soil type. For digital 98 soil mapping, spatial uncertainty could be estimated by methods such as geostatistical 99 methods and quantile regression forest (Vaysse and Lagacherie, 2017), which make sense of 100 statistic. The accuracy of soil dataset derived by digital soil mapping are estimated by cross-101 validation. But it is not trivial for those derived by the linkage method due to the global scale, 102 the support of the data and independent data (Stoorvogel et al., 2017) and most of these 103 maps are validated by statistics such as mean error and coefficient of determination. Instead, 104 some datasets, including WISE and GSDE, use some indictors such as linkage level of soil class 105 and sample size to offer quality control information (Shangguan et al. 2014; Batjes, 2016). A 106 simple way to compare the accuracy of datasets by both methods may be to use a global soil 107 profile database as a validation dataset, though some of these profiles were used in deriving 108 these datasets and questions will be raised. We evaluated several global soil property maps 109 in section 3. 110 111 112 The manuscript would benefit from a thorough English edit by a native speaker. 113 114 Reply: We will take an English editing service for the revised manuscript. 115 116 2. Specific comments 117 L15-16: Rephrase this as e.g.: Soil is an important regulator of earth system processes, but 118 remains one of the least well-described data layers in such models. 119 120 Reply: Modified as: Soil is an important regulator of earth system processes, but remains one 121 of the least well-described data layers in Earth System Models (ESMs). 122 123 L17: Function as->provide 124 125 Reply: Modified. 126 127 L22: Abundant soil observations are not 'enough'; these should have been analysed according 128 to comparable analytical methods and quality-assessed (which is seldom the case, see Batjes 129 et al. 2017). What about the geographical distribution, or possible clustering, of the available 130 (i.e. shared) soil profile data? 131 132 Reply: We changed the expression as 'with abundant, harmonized and guality controlled soil

| 133 | observations'. Corresponding contents are added accordingly. See the replies to related           |
|-----|---------------------------------------------------------------------------------------------------|
| 134 | comments of the reviewer.                                                                         |
| 135 |                                                                                                   |
| 136 | L24: By their nature, pedotransfer functions generally are not portable from one region to the    |
| 137 | other. Please add some discussion.                                                                |
| 138 |                                                                                                   |
| 139 | Reply: We add a sentence to the comments on L323. we added: PTFs generally are not                |
| 140 | portable from one region to the other (i.e. locally or regionally validated).                     |
| 141 |                                                                                                   |
| 142 | L24-25: Speculative as written, provide some arguments for this.                                  |
| 143 |                                                                                                   |
| 144 | Reply: See reply to comments on L451-452. This issue is discussed extensively by Looy et al.      |
| 145 | 2017 at the end of section 3. For briefty, we added a sentence here instead of long discussions:  |
| 146 | because ensemble modeling carries a number of benefits and potential over the use of a            |
| 147 | single model (Looy et al., 2017).                                                                 |
| 148 |                                                                                                   |
| 149 | L27-28: What about uncertainty in the co-variates?                                                |
| 150 |                                                                                                   |
| 151 | Reply: We put this as a part of the paragraph discussing uncertainty sources of the derive soil   |
| 152 | dataset. we added: Finally, uncertainty from the covariates is minor because spatial              |
| 153 | prediction models such as machining learning in digital soil mapping can reduce its influences    |
| 154 | (Hengl et al., 2014), though a more comprehensive list of covariates with higher resolution       |
| 155 | and accuracy will improve the predicted soil property maps.                                       |
| 156 | But there are many sources of uncertainty in addition to covariates. For brevity, we modified     |
| 157 | here: ESMs are often based on limited soil profiles and coarse resolution soil type maps with     |
| 158 | various uncertainty sources.                                                                      |
| 159 |                                                                                                   |
| 160 | L35-36 / 45: You may consider the following reference here:                                       |
| 161 | http://dx.doi.org/10.1002/2015GB005239.                                                           |
| 162 |                                                                                                   |
| 163 | Reply: The reference was added. It is helpful to understand the role of soil information in       |
| 164 | ESMs.                                                                                             |
| 165 |                                                                                                   |
| 166 | L43: Remove available                                                                             |
| 167 |                                                                                                   |
| 168 | Reply: Removed.                                                                                   |
| 169 |                                                                                                   |
| 170 | L45: How do you define 'better' here? Please clarify.                                             |
| 171 |                                                                                                   |
| 172 | Reply: We changed the word to 'more realistic'. This is in the following citations, Brunke et al. |
| 173 | (2016); Luo et al. (2016); Oleson et al. (2010). We added an example here: 'For example,          |
| 174 | Brunke et al., (2016) incorporated the depth to bedrock data in a land surface model using        |
| 175 | variable soil layers and instead of the previous constant depth.'                                 |
| 176 |                                                                                                   |

| 177 | L47-48: Also other types of soil data, for example soil biology (see ref. line L35-36).             |
|-----|-----------------------------------------------------------------------------------------------------|
| 178 | See also discussion in https://doi.org/10.1111/gcb.13896.                                           |
| 179 |                                                                                                     |
| 180 | Reply: We changed the sentence into 'ESMs require detailed information on the soil physical         |
| 181 | and, chemical and biological properties'.                                                           |
| 182 |                                                                                                     |
| 183 | L56: Useful to say that the range of soil data collected during a soil survey, will vary with scale |
| 184 | and projected applications of the data (i.e. type of soil survey, routine versus surveys/studies    |
| 185 | aimed at answering specific user demands).                                                          |
| 186 |                                                                                                     |
| 187 | Reply: We added a sentence to say this: The range of soil data collected during a soil survey,      |
| 188 | varies with scale, specifications of a country or a region, and projected applications of the       |
| 189 | data (i.e. type of soil surveys, routine versus specifically designed surveys). As a result, the    |
| 190 | availability of soil properties differs in different soil databases.                                |
| 191 |                                                                                                     |
| 192 | L72: How would you define reliable soil data? Remove from this sentence.                            |
| 193 |                                                                                                     |
| 194 | Reply: Removed                                                                                      |
| 195 |                                                                                                     |
| 196 | L76: Rather refer to measurements here.                                                             |
| 197 |                                                                                                     |
| 198 | Reply: Modified.                                                                                    |
| 199 |                                                                                                     |
| 200 | L87-88: Should add HWSD (FAO/IIASA/ISRIC/ISS-CAS/JRC, 2012) as reference for                        |
| 201 | this type of 'traditional' approach.                                                                |
| 202 |                                                                                                     |
| 203 | Reply: Added.                                                                                       |
| 204 |                                                                                                     |
| 205 | L93: usually not ready for!not appropriately scaled or formatted for                                |
| 206 |                                                                                                     |
| 207 | Reply: Modified.                                                                                    |
| 208 |                                                                                                     |
| 209 | L113-114: representing main soil types in a landscape unit characterised by soil profiles           |
| 210 | considered representative for the main component soils of the respective mapping units.             |
| 211 |                                                                                                     |
| 212 | Reply: Here we are describing two kinds of data from soil survey, i.e., soil map and soil profiles. |
| 213 | So We modified the sentence as: a map (usually in the form of polygon maps) representing            |
| 214 | main soil types in a landscape unit and soil profiles with observations of soil properties which    |
| 215 | are considered representative for the main component soils of the respective mapping units          |
| 216 |                                                                                                     |
| 217 | L124: Rephrase this: (FAO, 2003b, Zobler 1986) and these products are known to be                   |
| 218 | outdated. The information on the initial SMW and DSMW has since been updated for large              |
| 219 | sections of the world in the HWSD product (FAO/IIASA/ISRIC/ISS-CAS/JRC, 2012), which has            |
| 220 | recently been revised in WISE30sec ( http://dx.doi.org/10.1016/j.geoderma.2016.01.034 ).     |

| 221 |                                                                                                     |
|-----|-----------------------------------------------------------------------------------------------------|
| 222 | Reply: Added.                                                                                       |
| 223 |                                                                                                     |
| 224 | L124-125: Start new paragraph for the regional and national level data.                             |
| 225 |                                                                                                     |
| 226 | Reply: Modified.                                                                                    |
| 227 |                                                                                                     |
| 228 | L132: multiply -> multiple                                                                          |
| 229 |                                                                                                     |
| 230 | Reply: Modified.                                                                                    |
| 231 |                                                                                                     |
| 232 | L133: soil properties are observed (e.g. site data) or measured (e.g. pH, sand, silt, clay content) |
| 233 |                                                                                                     |
| 234 | Reply: Modified.                                                                                    |
| 235 |                                                                                                     |
| 236 | L138-141: Important to mention here that data served through WoSIS have been                        |
| 237 | standardised, with special attention for the description/comparability of soil analytical           |
| 238 | methods worldwide. See: http://dx.doi.org/10.17027/isric-wdcsoils.20180001. Also an                 |
| 239 | important element for the discussion is that many countries, although having a large                |
| 240 | collection of soil profile data, are not yet sharing such data. See for example:                    |
| 241 | https://doi.org/10.1016/j.grj.2017.06.001                                                           |
| 242 |                                                                                                     |
| 243 | Reply: Modified: Data served through WoSIS have been standardized, with special attention           |
| 244 | for the description and comparability of soil analytical methods worldwide. However, many           |
| 245 | countries, although having a large collection of soil profile data, are not yet sharing such data   |
| 246 | (Arrouays et al, 2017).                                                                             |
| 247 |                                                                                                     |
| 248 | L141: The initial list of attributes corresponds with the GlobalSoilMap specifications, with        |
| 249 | additional properties added/considered later in WoSIS (see http://dx.doi.org/10.17027/isric-        |
| 250 | wdcsoils.20180001).                                                                                 |
| 251 |                                                                                                     |
| 252 | Reply: Modified by adding the number of soil properties as follows: The soil profiles database      |
| 253 | of World Soil Information Service (WoSIS) contains the most abundant profiles (about 118,400)       |
| 254 | from national and global databases including most of the databases mentioned below (Batjes,         |
| 255 | 2017), though only a selection of important soil properties (12) are included (Ribeiro et al.,      |
| 256 | 2018).                                                                                              |
| 257 |                                                                                                     |
| 258 | L164: The linkage methods assigns a best-estimate for each soil property (and soil interval)        |
| 259 | under consideration to each component soil unit of a polygon (see e.g. HWSD). [see also 359-        |
| 260 | 360].                                                                                               |
| 261 |                                                                                                     |
| 262 | Reply:Modified as: Because the linkage method assigned only one value or a statistical              |
| 263 | distribution to a soil type in soil polygons (usually a polygon contains multiple soil types with   |
| 264 | their fractions), the intra-polygonal spatial variation is not taken into account.                  |

| 265        |                                                                                                   |
|------------|---------------------------------------------------------------------------------------------------|
| 266        | L171-173: For a more comprehensive review see also:                                               |
| 267        | http://dx.doi.org/10.1016/j.geoderma.2016.01.034 and http://dx.doi.org/10.1002/ldr.2656.          |
| 268        |                                                                                                   |
| 269        | Reply: We added the first one. But we did not add the second one because we did not find          |
| 270        | any available dataset online. We also sent an email to the corresponding author but no reply.     |
| 271        |                                                                                                   |
| 272        | 178: EVI WISE30sec considers seven layers up to 200 cm depth and 20 soil properties               |
| 272        |                                                                                                   |
| 273        | Renly: We added description of WISE30sec as one of the recent global datasets: WISE30sec          |
| 274        | is another improvement of HWSD incorporated more soil profiles with seven layers up to 200        |
| 275        | sm donth and with uncertainty estimated by mean + standard doviation. W/ISE20ses used the         |
| 270        | cill deptit and with direct antity estimated by mean $\pm$ standard deviation. Wisesosed used the |
| 277        | soli map from HWSD with minor corrections and climate zone maps as categorical covariate.         |
| 278        | 1991 Describle also securities the CCOC effects of the CCD have as                                |
| 279        | L201: Possibly, also mention the GSOC effort of the GSP here, see:                                |
| 280        | https://doi.org/10.5194/soil-4-1/3-2018.                                                          |
| 281        |                                                                                                   |
| 282        | Reply: Added as: A third global soil mapping project is the Global SOC Map of the Global Soil     |
| 283        | Partnership, which focuses on country-specific soil organic carbon estimates (Guevara et al.,     |
| 284        | 2018).                                                                                            |
| 285        |                                                                                                   |
| 286        | L205: which is currently the most detailed, though not necessarily most accurate estimation       |
| 287        | of ···                                                                                            |
| 288        |                                                                                                   |
| 289        | Reply: Due to our evaluation in section 3, it is also the most accurate estimation. So we did     |
| 290        | not describe the accuracy here but in section 3.                                                  |
| 291        |                                                                                                   |
| 292        | L214: See also: Tifafi M, Guenet B and Hatté CCG 2018. Large differences in global and            |
| 293        | regional total soil carbon stock estimates based on Soil-Grids, HWSD and NCSCD:                   |
| 294        | Intercomparison and evaluation based on field data from USA, England, Wales and France.           |
| 295        | Global Biogeochemical Cycles, 42-56. http://dx.doi.org/10.1002/2017GB005678. Note: This           |
| 296        | paper is erroneously referred to as Marwa et al. 2018 in manuscript. This should be: Tifafi et    |
| 297        | al. 2018.                                                                                         |
| 298        |                                                                                                   |
| 299        | Reply: Corrected                                                                                  |
| 300        |                                                                                                   |
| 300        | 1214: Check if this is for 0-100 cm; likely these estimates are for 0-200 cm (see also recent     |
| 303        | sources mentioned above)                                                                          |
| 202
202 |                                                                                                   |
| 3U3
204 | Paply It is reported as 0, 100 am in the ref                                                      |
| 304
205 | Reply: It is reported as 0-100 cm in the ref.                                                     |
| 305        |                                                                                                   |
| 306
007 | LZZ4: Large sections of HWSDV1.2 still draw on the now outdated DSMW.                             |
| 307        |                                                                                                   |
| 308        | Reply: Modified as: Except GSDE, HWSD and STATSGO (Miller and White, 1998) for USA in             |

| 309        | Table 1, these datasets were derived from the Soil Map of the World (note that large             |
|------------|--------------------------------------------------------------------------------------------------|
| 310        | sections of GSDE and HWSD still used this map as a base map because there are no available       |
| 311        | regional or national maps)                                                                       |
| 312        |                                                                                                  |
| 313        | L295-296: See earlier comments.                                                                  |
| 314        |                                                                                                  |
| 315        | Reply: We added WISE30sec in Table 2 and 3.                                                      |
| 316        |                                                                                                  |
| 317        | L299: WISE30sec presents estimations of uncertainty, unlike the HWSD and GSDE.                   |
| 318        |                                                                                                  |
| 319        | Reply: Modified as: Except WISE30sec, all these databases do not contain uncertainty             |
| 320        | estimation.                                                                                      |
| 321        |                                                                                                  |
| 322        | L300: Needs some discussion and references to publications on the subject.                       |
| 323        |                                                                                                  |
| 324        | Reply: We evaluated several global soil dataset using WoSIS in section 3:                        |
| 325        | 3 Comparison of available global soil datasets                                                   |
| 326        | For the convenience of ESMs' application, we compared several available soil datasets and        |
| 327        | evaluated them with soil profiles from WoSIS for some key variables (Sand, clay, content         |
| 328        | organic carbon coarse fragment and bulk density) used in FSMs. In addition to the most           |
| 329        | recent developed soil datasets, we also included one old data set (i.e. IGBP) used in ESMs for   |
| 330        | the evaluation It is not necessary to compare all the old data sets because they are based on    |
| 331        | similar limited and outdated source data as described in section 2.2. They have coarser          |
| 332        | resolution (Table 1) than the newly developed soil datasets (Table 2)                            |
| 333        | We present basic descriptions about the new soil datasets in Table 2 and 3. As described in      |
| 331        | section 2.1 four available global soil datasets i.e. HWSD_GSDE_WISE30sec and Soilgrids           |
| 225        | have been developed in the last several years (Table 2). These soil datasets are selected to be  |
| 338        | shown here because they have a global coverage with key variables used by ESMs and               |
| 330        | developed with relatively good data sources in recent years, and are freely available. Old       |
| 228        | versions of these datasets are not shown here. Table 3 shows the available soil properties of    |
| 330        | these soil datasets. Except WISE30sec, all these databases do not contain spatial uncertainty    |
| 310        | estimation. The explained variance of soil properties in Soilgrids is between 56% and 82% while  |
| 2/1        | the other datasets do not offer quantitative accuracy assessment. CSDE has the largest           |
| 241        | number of soil properties, while Soilgride currently contains to primary soil properties         |
| 04Z        | defined by the Clobal cill Man concertium                                                        |
| 343        | The ecourse of the power developed and detects (Solidride, CCDE and UN(CD) and an old            |
| 344
245 | deteret (ICDD) are evoluted for five key veriables using 04.441 coil profiles from MoSIS (Table  |
| 343        | (IGBP) are evaluated for the evaluation including mean error (ME) react mean error               |
| 340        | 4). We used four statistics in the evaluation, including mean error (ME), root mean squared      |
| 347        | error (RMSE), coefficient of variation (CV) and coefficient of determination (R2). All soli-     |
| 348
240 | datasets are evaluated for topsoli (U-bucm) and subsoli (BU-LUUCM). The layer schemes of soli    |
| 349        | datasets are different (Table 1) and they were converted to the two layers. Soil datasets are in |
| 350        | nign resolution and were converted to the resolution of 10 km by averaging. All datasets have    |
| 351        | relative small ME. In general. Soligrids has much better accuracy than the other three due to    |
| 352        | RMSE, CV and R2, and GSDE ranks the second, followed by IGBP and HWSD. However, IGBP             |

353 is slightly better than GSDE for bulk density and organic carbon of topsoil. Note that the IGBP 354 does contain coarse fragment, which is needed in calculating soil carbon stocks. We did not 355 evaluate the WISE30sec here to save some time in data processing, because previous 356 evaluation using WoSIS showed that WISE30sec had slightly better accuracy than HWSD 357 (https://github.com/thengl/SoilGrids250m/tree/master/grids/HWSD). This evaluation has 358 some limitations. First, because the datasets developed by the linkage method give the mean 359 value of a SMU resulted in abrupt change between the boundaries of soil polygons while the 360 datasets developed by digital soil mapping simulated the soil as a continuum with a spatial 361 continuous change of soil properties, they may not be so comparable. Second, the original 362 resolution of soil datasets are different, which means that maps with higher resolution 363 provides more spatial details and we should judge the map quality due to not only the 364 accuracy assessment but also the resolution. As a result, datasets with higher resolution (i.e. 365 HWSD and GSDE) are preferred than that with lower resolution (i.e., IGBP) as they have similar 366 accuracy, especially when the LSMs are run at a high resolution such as 1km. Third, the vertical 367 variation are better represented by Soilgrids, GSDE and WISE30sec with more than 2 layers 368 and to a depth over 2m (Table 2). This will provide more useful information for ESMs, 369 especially when they model deeper soils with multiply layers.

370 The new generation soil dataset produced by digital soil mapping method gave a quite 371 different distribution of soil properties from those produced by the linkage method. Figure 1 372 shows soil sand and clay fraction at the surface 0-30 cm layer from Soilgrids, IGBP and GSDE. 373 Figure 2 shows soil organic carbon and bulk density at the surface 0-30 cm layer from 374 Soilgrids, IGBP and GSDE. Significant differences are visible in these datasets. This will lead to 375 different modelling results in ESMs. Tifafi et al. (2018) found that the global soil organic carbon 376 stocks down to a depth of 1m is 3,400 Pg estimated by Soilgrids while it is 2500 Pg by HWSD, 377 and the estimates by Soilgrids are closer to the observations, though they all underestimated 378 the soil carbon stocks. Figure 1 of Tifafi et al. (2018) showed the global distribution of soil 379 carbon stocks by Soilgrids and HWSD.

In general, Soilgrids is preferred for ESMs' application as it has the highest accuracy and resolution at the time. When soil properties are not available in Soilgrids, WISE30sec and GSDE offers the alternative options. However, model sensitivity simulations need to be done to investigate the effects of different soil datasets on ESMs in future studies.

L302: Larger number of soil properties for GSDE, but what about the accuracy of the

385 predictions? (not given as indicated earlier).

386

387 Reply: Not only GSDE but also HWSD, WISE30sec do not provide a quantitative accuracy 388 assessment. WISE30sec provides uncertainty estimation of each soil unit, and HWSD and 389 GSDE could take similar way to estimate the uncertainty. But (spatial) uncertainty estimation 390 is different from accuracy assessment. As we discussed above, we do the evaluation in section 391 3. GSDE did some quality assessment using some indicators like WISE, including linkage level 392 of soil class, sample size, texture consideration, search radius and map unit level (see figure 6 393 of Shangguan et al., 2014). But it is only a reference of the accuracy and not straight forward 394 for users, and most users may not even take a look at it. We add some discussions in the 395 paragraph of uncertainty: Spatial uncertainty is estimated by different methods for the linkage 396 method and digital soil mapping methods. For the linkage method, statistics such as standard 397 derivation and percentiles can be used as spatial uncertainty estimation, which are calculated 398 for the population of soil profiles linked to a soil type or a land unit (Batjes, 2016). This 399 estimation has some limitations because soil profiles are not taken probabilistically but based 400 on their availability, especially for the global soil datasets. Uncertainty will be underestimated 401 when the sample size is not big enough to represent a soil type. For digital soil mapping, 402 spatial uncertainty could be estimated by methods such as geostatistical methods and 403 quantile regression forest (Vaysse and Lagacherie, 2017), which make sense of statistic. The 404 accuracy of soil dataset derived by digital soil mapping are estimated by cross-validation. But 405 it is not trivial for those derived by the linkage method due to the global scale, the support of 406 the data and independent data (Stoorvogel et al., 2017) and most of these maps are validated 407 by statistics such as mean error and coefficient of determination. Instead, some datasets, 408 including WISE and GSDE, use some indictors such as linkage level of soil class and sample 409 size to offer quality control information (Shangguan et al. 2014; Batjes, 2016). A simple way 410 to compare the accuracy of datasets by both methods may be to use a global soil profile 411 database as a validation dataset, though some of these profiles were used in deriving these 412 datasets and questions will be raised. We evaluated several global soil property maps in 413 section 3. 414 415 L303: Rephrase. ... SoilGrids products currently consider the list of attributes as defined 416 by the GlobalSoilMap consortium. 417 418 Reply: Modified: while Soilgrids currently contains ten primary soil properties defined by the 419 GlobalSoilMap consortium. 420 421 L323: Most PTFs are not portable (i.e. locally or regionally validated). 422 423 Reply: We added: PTFs generally are not portable from one region to the other (i.e. locally or 424 regionally validated). 425 426 L331-332: add database (word is missing in sentence) 427 428 Reply: Modified. 429 430 L360-361: ...component soil unit in most cases, and thus a one-to-many relationship 431 exists between the SMU and the profile attributes of the respective soil units... 432 433 Reply: Modified. 434 435 L397-398: Possibly, rephrase this sentence. 436 437 Reply: Modified: However, some researches used the "aggregating after" method producing 438 misleading results (Hiederer and Köchy, 2012). 439 440 L410: remove high from sentence

442 Reply: removed.

443

444 L441-442: Provide some justification (a sentence or two) for this statement.

445

446 Reply: added: because they provide spatial continuous estimations of soil properties using447 spatial prediction models with various soil-related covariates.

448

449 L451-452: Speculative as written. Please provide some evidence for this.

450

Reply: This issue is discussed extensively by Looy et al. 2017 at the end of section 3. For briefty,
we added a sentence here instead of long discussions: because ensemble modeling carries a
number of benefits and potential over the use of a single model (Looy et al., 2017).

454

455 For you reference, I copied the content from Looy et al. (2017) here: Another recent 456 technique that has merits in this respect is ensemble modeling – i.e. the use of a number of 457 models in combination. This technique is a natural part of weather and climate modeling 458 today, yet it is less used in the prediction of soil properties [Baker and Ellison, 2008b]. 459 Ensemble modeling carries a number of benefits and potential over the use of a single model. 460 Models can differ in their theory and structure, but also in the information that they require. 461 As a result, their sensitivity and scale of support may also differ. The use of ensemble modeling 462 is easy to justify if it is difficult to determine which, if any, single model may be superior to 463 others. In ensemble modeling, the main aim is not to make the single model perfect, but to 464 capture the trend that multiple models agree on. The ensemble will amplify trends that are 465 common among models, while by-chance predictions will be softened. The outputs, therefore, 466 can be interpreted - qualitatively or quantitatively - as a measure of uncertainty. In the context 467 of integrated Earth system models, the represented complex processes – integrating physical 468 and biochemical processes typically – can be covered by a number of models with strongly 469 varying concept and structure. Here lies an opportunity to construct ensemble models 470 entering different PTF-based parameterizations.

471

472 L460: and quantifying uncertainty in the predictions

473

474 Reply: Added

475

476 L461: 'need to gain popularity in ...'. Basically, the "proof of the pudding is in the

477 eating".

478

Reply: We provided some examples at regional scales, which shows products by digital soil mapping improved climate modelling results (Kearney and Maino, 2018; Trinh et al., 2018). But no global studies have been taken to compare digital soil mapping products and linkage method products in ESMs yet, which we are doing now. So we changed this sentence to a more conservative one: the new generation soil datasets derived by digital soil mapping need to be tested in ESMs, and some regional studies have shown that these datasets provided

better modelling results than products by the linkage method (Kearney and Maino, 2018;
Trinh et al., 2018). Moreover, many studies from digital soil mapping have identified that soil
maps are not very important to predict soil properties and are usually not used as a covariate
in most studies (eg. Hengl et al., 2014; Viscarra Rossel et al., 2015; Arrouays et al., 2018).
However, the linkage method usually takes soil map as the major covariate, which essentially
affect the accuracy of the derived soil property maps, especially for areas without detailed soil
maps.

492

L462: What I miss in this paper, is a discussion of the inherent uncertainty attached to using
soil profile data coming from various sources. Often, little consideration is given to differences
in analytical methods used for analysing e.g. soil organic carbon content worldwide (see
Shangguang et al 2014, who consider this as 'a major imitation to their approach'). For a
discussion of issues see e.g.: http://dx.doi.org/10.17027/isricwdcsoils.20180001

498

Reply: This was mentioned in L483-L484: Data compatibility of different analysis methods and
different description protocols including soil classifications is also an important issue and data
harmonization is necessary when the data are made available to public. Also, see the
paragraph discussing uncertainty we added.

503

L463-464: More soil profiles is not necessarily the solution. More quality assessed data, analysed according to comparable analytical methods, are needed to support such efforts.
Reference should be made to 'new' types of data as derived from proximal sensing (e.g. <a href="http://dx.doi.org/10.5194/soil-2017-36">http://dx.doi.org/10.5194/soil-2017-36</a>), and associated limitations. Reference, in this respect, could also be made to the GLOSOLAN effort, initiated by the GSP (http://www.fao.org/global-soilpartnership/resources/events/detail/en/c/1037455/) and work of GSP Pillar 5 towards harmonisation (http://www.fao.org/3/a-bs756e.pdf). Also, importantly, the geographical

- 511 distribution and possible clustering of the shared soil profiles.
- 512

Final Series Series

L471-475: True, but how many of these profiles are actually being shared for the greaterbenefit of the international community? See paper by Arrouays et al. 2017 for a discussion.

522

523 Reply: We added: Arrouays et al. (2017) reported that about 800,000 soil profiles have been524 rescued in the selected countries.

525

526 L479: Some reference to the ongoing work of the Global Soil Partnership, Pillars 4 and

527 5, is needed here.

529 Reply: Added: (Pillar four Working Group, 2014; Pillar 5 Working Group, 2017)

L948: Table 2 is not complete; 'recent' datasets not yet considered in the review should be
added here (<a href="http://dx.doi.org/10.1002/ldr.2656">http://dx.doi.org/10.1002/ldr.2656</a>;
http://dx.doi.org/10.1016/j.geoderma.2016.01.034
ldem for Table 3.

534

530

535 Reply: WISE30sec is added. The other one (Stoorvogel et al., 2017, which we cited in our paper) 536 is more about proposing a new method which can improve HWSD results. 'The RMSD for S-537 World was considerably smaller (2.1% SOC) than the RMSD for HWSDweighted (2.9% 538 SOC). 'But this method has some limitation for soil properties with limited samples and for 539 those having week relationship with covariates. We don't find the dataset available online. 540 And in the paper, they only tested 6 soil properties. i) topsoil thickness (cm), ii) soil depth 541 (cm), iii) soil organic carbon (SOC) content in the top 30 cm (%), iv) SOC content in the subsoil 542 (30 to 120 cm) (%), v) clay content in the soil profile (%), and vi) sand content in the soil profile 543 (%). So we did not add this citation as a dataset for now. I have written email to the 544 corresponding author to check the availability but not reply.

545

546 L952: Table 3. Change title to "Derived soil properties considered in three global soil datasets". 547 Essentially, this is a simple enumeration of derived soil properties. However, the fact that many 548 different analytical methods have been used to derive a given soil property (e.g. soil organic 549 carbon Walkley & Black method or LECO total analyses) or which CEC (e.g. measured at 'field 550 pH' or in a buffer-solution at 'pH7' or 'pH8') has been considered is not mentioned here (in 551 a footer perhaps). In their study, Shangguan et al. (2014) rightly indicate that this has not been 552 the case and indicate that they see this an important limitation. However, there are still no 553 straightforward mechanisms for harmonising the data (cf. GSP Pillar 5 and GLOSOLAN 554 activities, as mentioned above).

555

Reply: title changed. We add a sentence in the footnote: It should be noted that many
different analytical methods have been used to derive a given soil property, which is a major
source of the dataset.

- 559
- 560 Potaasium -> Potassium
  561
  562 Reply: corrected
- 563

| 564 | Reply to reviewer 2                                                                              |
|-----|--------------------------------------------------------------------------------------------------|
| 565 |                                                                                                  |
| 566 | 2. General comments                                                                              |
| 567 | Comment: A review of soil datasets available for Earth system modeling is extremely useful,      |
| 568 | given the wide application of ESMs in important projects such as the coupled model               |
| 569 | intercomparison projects (CMIP) serving the IPCC reports, and in view of the challenges of       |
| 570 | observing soil properties covering the globe. However, the manuscript does in fact not fulfill   |
| 571 | what it promises in the title. It does not review datasets and compares them quantitatively      |
| 572 | (apart from selected maps in Fig. 1-2, but a systematic comparison is missing). Instead it       |
| 573 | discusses in length linkage and digital mapping methods, then how soil observational data in     |
| 574 | general can be incorporated in ESMs and what challenges arise. This is valuable *ancillary*      |
| 575 | information, and the manuscript summarizes a lot of important information on these topics.       |
| 576 | But the main purpose of the paper is missed. A careful review of available datasets needs to     |
| 577 | be added, which is of course a major revision: there should be more than the 3 datasets in       |
| 578 | Tab. 3, unless justified that these 3 are special (for example it would be very illustrative to  |
| 579 | include all the currently used old datasets as well to know what a difference the new datasets   |
| 580 | might make). There should be a review also of other data than global maps, as needed e.g.        |
| 581 | for parameters. Most importantly, however, a quantitative comparison of at least key variables   |
| 582 | should be included, with useful statistical measure (maps, global mean and variability,          |
| 583 | latitudinal means, comparison against selected observational high-quality sites,). Ideally,      |
| 584 | model sensitivity simulations would be run, but this latter point is not essential.              |
| 585 |                                                                                                  |
| 586 | Reply: The purpose of the review is to offer insights to both soil data developer and ESM users. |
| 587 | So we discussed contents may be interesting to both sides. That is why we discussed in           |
| 588 | length linkage and digital mapping methods, then how soil observational data in general can      |
| 589 | be incorporated in ESMs and what challenges arise.                                               |
| 590 | We agree that a systematic quantitative comparison is a very important aspect this review        |
| 591 | should cover. So we compared a selection of global soil data sets with a focus on the most       |
| 592 | recent developed ones (i.e., HWSD, GSDE, WISE30sec and Soilgrids). We also included one          |
| 593 | 'old' data set (i.e. IGBP) used in ESMs for the comparison. It is not necessary to compare all   |
| 594 | the old data sets because they are based on similar, limited and outdated source data. All the   |
| 595 | old soil data are based on the FAO soil map and no more than 5,800 soil profiles (described      |
| 596 | in section 2.2). We can see that the newly developed soil data in fig.1-2 have some major        |
| 597 | differences. It is valuable to compare them even though they may not be so comparable,           |
| 598 | because the datasets developed by the linkage method give the mean value of a SMU                |
| 599 | resulted in abrupt change between the boundaries of soil polygons while the datasets             |
| 600 | developed by digital soil mapping simulated the soil as a continuum with a spatial continuous    |
| 601 | change of soil properties. Nevertheless, we used site observations of WOSIS to evaluate the      |
| 602 | soil data sets, though these observations are used or partly used in the development of global   |
| 603 | distribution of soil data. We compared the key variables (Sand, clay content, organic carbon,    |
| 604 | coarse tragment and bulk density) used in ESM with useful statistical measures. However,         |
| 605 | model sensitivity simulations will not be done in this review and need to be done in other       |
| 606 | studies. This review focuses on the global soil property maps in ESMs. We did not extend the     |
| 607 | content to other data including model parameters which is a different topic but valuable. As     |

608 we mentioned in the manuscript, variables such as soil temperature and soil moisture are 609 beyond this paper's scope. To avoid misunderstanding, we changed the title to 'a review on 610 the global soil property maps for earth system model' and modify the corresponding 611 expression in the manuscript. However, we will use the term soil datasets for brevity. We 612 added a new section (section 3) to show the comparison of the soil datasets:

613 3 Comparison of available global soil datasets

[revised manuscript text omitted]

673 Comment: A method for the review is missing, which leaves the reader in doubt whether 674 he/she has been reading an opinion piece or a comprehensive review. Currently both the 675 selection of mentioned datasets and the selection of ESMs is incomprehensive and not 676 justified in its selection. For the models one could imagine to do a review of all TRENDY LSMs 677 or of all CMIP5 (or even better CMIP6) ESMs and the datasets they are using. For the available 678 datasets some objective criteria should be given as well, e.g. a list of criteria that datasets need to fulfill to be included in Tab. 3 (global, soil type and property x, y, z need to be 679 680 included,...)

681

682 Reply: We described the selection of mentioned datasets and the selection of ESMs. As we 683 mentioned in the above reply, datasets are chosen according to their source data quality and 684 developing time. In addition, the datasets should be freely available. We do not require 685 minimum number of soil properties as long as the soil dataset are global maps (but they all 686 have the key variables in ESMs evaluated in section 3). The selection of mentioned datasets 687 in Table 2 and 3 are described as: These soil datasets are selected to be shown here because 688 they have with key variables required by ESMs and developed with relatively good data 689 sources in recent years, and are freely available. Old versions of these datasets are not shown 690 here.

691

692 Our currently list of ESMs covers the major LSMs but not all of them because a complete list 693 will be too lengthy. We also checked the list of CMIP5 and added these models if they have 694 documented the soil dataset used. Only two ESMs (i.e., INMCM and MRI-ESM) are not include 695 in Table 1 as there is no information about the soil dataset. According to the editors' comment,

we start the table 1 with the soil dataset instead of the models as our focus is on soil data
rather than ESMs or LSMs. We modified the description of Table 1: Table 1 shows several
most popular ESMs (specifically, their land surface models) and their input soil datasets. The
ESMs in Table 1 cover the list of CMIP5 (Coupled Model Intercomparison Project) except those
without information about the input soil datasets.

- 701
- 702

703 Comment: The organization of the sections does not appear logical: Datasets and their usage 704 in ESMs (Section 2) is very good, presenting PTFs as Section 3 promises in I. 105 is also very 705 useful but these PTFs are in fact never presented and compared, just discussed. Section 4 706 deals with data from the linkage method. why? Why not data from digital mapping as well? 707 Section 5 deals with upscaling to the coarse ESM resolution. This is an important point, but 708 there are many other challenges related to application of soil datasets in ESMs: One obstacle 709 is that observations are not covering the soil depth as deeply as the ESMs and in other layer 710 distributions. Another that soil observations are derived from present-day, which has 711 confounding effects of both environmental changes (climate, CO2, nutrient deposition, ...) 712 and historical land use changes. Would this affect soil thermal and other properties needed 713 as input to ESMs? How should one evaluate ESMs only for present-day then?

714

715 Reply: PTFs is not the major focus of this review while there is a very good review on PTF in 716 ESMs which we cited as Looy et al., (2017). Section 4 does not discuss data from digital 717 mapping because it does not have the aggregating problem like the data by the linkage 718 method. Data by the linkage method are derived for each soil map unit and data by digital 719 mapping are derived for each grid. We added sentences to clarify this: Soil data by the linkage 720 method are derived for each soil mapping unit or land unit and thus is polygon-based, while 721 ESMs are usually grid-based. However, soil data derived by digital soil mapping are grid-722 based. So, the compatibility between soil data derived by the linkage method and ESMs needs 723 to be addressed.

724

We agree that issues about the changing soil properties should be discussed. We added asection : section 4.3 The changing soil properties:

727 There is not any global soil property map in time-series because we do not have enough 728 available data. In all the global soil property maps, all the available soil observations in the 729 last decades are used in the development of soil property maps without considering the 730 changing environment. So these datasets should be considered as an average state. The 731 critical issue for mapping global soil properties in time-series is to establish a soil profile 732 database with time stamps and then divide them into two or more groups of different periods 733 such as 1950s-1970s. This is still quite challenging at the global scale because the spatial 734 coverage of soil profiles is quite uneven for different periods and the sample size may not be 735 big enough to derive maps with satisfied accuracy.

736

Soil properties are changing but we are now taking it as static in ESMs. As some ESMs already
simulate the soil carbon, this may be considered in PTFs used to estimate soil hydraulic and
thermal parameters. Other soil properties affecting soil hydraulic and thermal parameters

740 include soil texture, bulk density, soil structure and so on, but the change is relatively slow. 741 The effect of environmental change on soil properties is the topic of quantitative modeling of 742 soil forming processes, i.e. soil landscape and pedogenic models (Gessler et al., 1995; Minasny 743 et al., 2008). If we need to simulate the change of soil properties, a coupling of ESMs and soil 744 landscape and pedogenic models will be needed. Otherwise, we need to predict the soil 745 properties in the future using soil landscape and pedogenic models which are small scale 746 models and has high uncertainty. The prediction of changing soil properties may also be done 747 by digital soil mapping taken the changing (especially for the future) climate and land use as 748 covariates, which may be the more feasible than dynamic models.

749 750

751 We agree that we should also discuss the lack of deep soil data and the different layer 752 schemes of soil data and ESMs. We added:

4.5 Layer schemes and lack of deep layer soil data

The layer scheme of a soil data set needs to be coveted to that of ESMs for model use. A simple way for this conversion is the depth weighting method. When a more accurate conversion is needed, the equal-area quadratic smoothing spline functions can be used, which is proved to be advantageous in predicting the depth function of soil properties (Bishop et al., 1999). Mass conservation for a soil property of a layer is guaranteed by this method under the assumption of continuous vertical variation of soil properties. This method may produce some negative values which should be set to zero.

761 The depth of soil observations in soil survey are usually less than 2 m and thus resulted in 762 missing values for the deep layers of ESMs. For the lack of deep soil data, there is not any 763 good solution other than extrapolate the values based on the observations of shallower layers, 764 which will lead to higher uncertainty of soil properties for deep layers. The extrapolation can 765 be done by the above-mentioned spline method or simply by assigning soil properties of the 766 last layer to the rest of deeper soil layers. Depth to bedrock map (Shangguan et al., 2017) can 767 be utilized in defining the low boundary of soil layers, and a default set of thermal and 768 hydraulic characteristic can be assigned for bedrocks.

- 769
- 770

771 Comment: How should ESMs deal with observational uncertainty (see comment below)?

772

773 Reply: See the reply to the specific comment below.

774

775 Comment: I think what this paper needs to cover is (0) specifying what ESMs need, i.e. which 776 spatial and temporal coverage, which variables (extending the list of parameters, initial state, 777 evaluation/benchmarking in the introduction) (1) general methodology of deriving this soil 778 information (mostly Sec. 2, PTFs would go in this section as well.) (2) comprehensive, 779 quantitative comparison of available global soil datasets (largely missing) (3) discussion of 780 existing challenges of data usage in ESMs, where one should come back to the list of usages 781 in the introduction: evaluation data for example does not have to have global coverage. The 782 upscaling would be one of several points here.

784 Reply: We reorganized this manuscript as the reviewer recommended. However, there are 785 some issues to be clarified. As we mentioned above, this review focuses on the global soil 786 property maps in ESMs. We did not extend the content to other data including model 787 parameters and data without a global coverage which is a different topic but valuable. As we 788 mentioned in the manuscript, variables such as soil temperature and soil moisture are beyond 789 this paper's scope. For the temporal change of soil properties, we addressed it as a challenge 790 as there is no time series of global soil property map yet. 791 792 793 Comment: The paper is not very well written. First, the use of English language is incorrect or 794 uncommon. Second, many expressions are not accurate. Just taking the first sentence as 795 example: "Soil or pedosphere is a key component of Earth system, and plays an important 796 role in the water, energy and carbon balances and biogeochemical processes."First, it should 797 read "The soil or pedosphere is a key component of the Earth system, ..." (where "Earth" is 798 correctly written in capitals, while it is not in the title: : :). Second, the carbon cycle is one 799 example of biogeochemical processes, so it should read " and \*other\* biogeochemical 800 processes". I am not correcting any of these language and accuracy errors in the following 801 because they are too numerous. 802 803 Reply: Thanks for pointing out these errors. We revised this manuscript and will take a 804 language service after the final revision. 805 806 807 808 More detailed comments: 809 p. 1 810 Comment: \* "Soil datasets function as model parameters": do the authors mean that model 811 parameters can be derived from soil carbon maps? What parameters are they thinking of? 812 813 Reply: we corrected the expression to model inputs. This is the major usage of soil property 814 maps in ESMs (table 1). 815 816 Comment: \* "are preferred to those by the linkage method for ESMs": not understandable 817 at this point in the manuscript - what is the "linkage method"? 818 819 Reply: we also added the other name of it: "taxotransfer rule-based method', which may be 820 a more understandable terminology. But this terminology is not possible to explain in the 821 abstract. 822 Comment: \* "to provide secondary soil parameters to ESMs": what are secondary soil 823 824 parameters? 825 826 Reply: we modified it to "derived soil properties", which includes soil hydraulic, thermal and

827 biogeochemical parameters. And we explained this when "secondary soil parameter" first

828 appear in the manuscript.

829

830

Comment: Generally, the abstract does not read like a review of datasets, but like a
commentary on challenges of integrating soil carbon datasets in ESMs. As a reader I would
have expected an abstract here of types of data, see general comment above.

834

835 Reply: we revised the abstract adding the related contents. Note that comparison and 836 evaluation of datasets is only one aspect of this review: Soil is an important regulator of Earth 837 system processes, but remains one of the least well-described data layers in Earth System 838 Models (ESMs). We reviewed global soil property maps from the perspective of ESMs, 839 including soil physical and, chemical and biological properties, which can also offer insights 840 to soil data developers. These soil datasets provide model inputs, initial variables and 841 benchmark datasets. For modeling use, the dataset should be geographically continuous, 842 scalable and with uncertainty estimates. The popular soil datasets used in ESMs are often 843 based on limited soil profiles and coarse resolution soil type maps with various uncertainty 844 sources. Updated and comprehensive soil information needs to be incorporated in ESMs. New 845 generation soil datasets derived by digital soil mapping with abundant, harmonized and 846 quality controlled soil observations and environmental covariates are preferred to those by 847 the linkage method (i.e. taxotransfer rule-based method) for ESMs. Soilgrids has the highest 848 accuracy and resolution among the global soil datasets at the time, while other recently 849 developed datasets are useful compliments. Because there is no universal pedotransfer 850 function, an ensemble of them may be more suitable to provide derived soil properties to 851 ESMs. Aggregation and upscaling of soil data are needed for model use but can be avoid by 852 taking a subgrid method in ESMs at the cost of increases in model complexity. Producing soil 853 property maps in time series is still challenging. Uncertainty of soil data needs to be 854 estimated and incorporated in ESMs.

855

856 p. 2

857 Comment: \* "However, soil dataset used in ESMs is not well updated nor well utilized yet.":
858 This needs citation of which datasets are used and felt by the authors to be outdated.

859

860 Reply: To make this more objective, we added some citation from FAO and globalsoilmap (a 861 community joint effort project), not felt by us. We have explained this in section 2.2: Except 862 GSDE, HWSD and STATSGO (Miller and White, 1998) for USA in Table 1, these datasets were 863 derived from the Soil Map of the World (note that large sections of GSDE and HWSD still used 864 this map as a base map because there are no available regional or national maps) (FAO, 1971-865 1981) and limited soil profile data (no more than 5,800 profiles), which gained popularity 866 because its simplicity and ease of use. But these are outdated and should no longer be used 867 because much better soil information as introduced in Section 2.1 can be incorporated 868 (Sanchez et al., 2009; FAO/IIASA/ISRIC/ISS-CAS/JRC, 2012).

869 870

871 Comment: \* I. 45-48: Kearney & Maino are one specific study for Australia for soil moisture

872 using one new soil dataset. Using this as reference for the entire "Earth system" and for "will 873 improve" in the future is a stretch. Better look for a couple of references and spell them out 874 explicitly. 875 876 Reply: This is only an example. We added more citations here: (eg. Livneh et al., 2015; Dy and 877 Fung, 2016; Kearney and Maino, 2018). More examples are given with brief description in 878 section 2.2. 879 880 Comment: \* "could avoid the possibility of the non-linear singularity evolution of the 881 modeling": this needs to be explained in one more sentence. Do the authors mean that 882 models may have multiple equilibria? 883 884 Reply: Yes, it means models may have multiple equilibria. And we also added a sentence: The 885 setting of initial nutrient stocks is a major factor leading to model-to-model variation in the 886 simulation (Todd-Brown et al., 2014). 887 р. З 888 889 Comment: \* "for multiply layers rather than a global constant": This mixes up vertical 890 resolution (-> layers) and horizontal resolution (-> global constant). Be more explicit in your 891 description. 892 893 Reply: we modified it as: As a result, ESMs usually incorporate soil property maps (i.e., 894 horizontal spatial distribution) for multiply layers rather than a global constant or a single 895 layer. 896 897 Comment: \* Is "linkage method" really the proper technical term here? It seems to me it is 898 used in the literature rather for remapping than for linking soil observations to environmental 899 variables. The paper would benefit from a clearer overview of technical terms and methods, 900 if it is meant to serve as a review. 901 902 Reply: We used this term for brevity. But it may be misleading if readers are not familiar with 903 soil mapping. So we also used the other term taxotransfer rule-based method. We added this 904 term when it first appears in the manuscript: The traditional way (i.e., the linkage method, also 905 called taxotransfer rule-based method) is to link soil profiles and soil mapping units on soil 906 type maps, sometimes with ancillary maps such as topography and land use (Batjes, 2003; 907 FAO/IIASA/ISRIC/ISS-CAS/JRC, 2012). 908 909 Comment: \* paragraph starting I. 93: Vector to raster conversion and remapping to a different 910 resolution are certainly not the biggest or at least not the only obstacles to including soil 911 datasets in models. That models need different variables than those directly observable or 912 that observational datasets cover only a certain depth, which most often is different from the 913 one ESMs cover, are examples of other important challenges. Overall, I feel the sections 914 internally should be a bit better structured, with one topic being covered comprehensively by 915 one paragraph.

917 Reply: This paragraph served as a brief introduction of the obstacles to including soil datasets 918 in models and detailed description were given in the later sections. That models need different 919 variables than those directly observable is related to the PTF development. So we did not put 920 here. We added the challenge of layer schemes here. We modified the paragraph as follows: 921 There are many challenges related to application of soil datasets in ESMs. First, soil datasets 922 are usually not appropriated scaled or formatted for the use of ESMs and some upscaling 923 issues, which is the most frequently encountered, need to be addressed. The soil datasets 924 produced by the linkage methods are polygon-based and need to be converted to fit the 925 grid-based ESMs. This conversion can be done by either subgrid method or spatial 926 aggregation. The up-to-date soil data are provided at a resolution of 1km or finer, while the LSMs are mostly ran at a coarser resolution. So upscaling of soil data is necessary before it 927 928 can be used by ESMs. Proper upscaling methods need to be chosen carefully to minimize 929 uncertainty in the modeling results introduced by them (Hoffmann and Christian Biernath, 930 2016; Kuhnert et al., 2017). Second, all the current global soil datasets represent the average 931 state of last decades, and producing soil property maps in time series is still challenging. Soil 932 landscape and pedogenic models are developed to simulate soil forming processes and soil 933 property changes, which can be incorporated into ESMs. The prediction of changing soil 934 properties can be also done by digital soil mapping taken the changing climate and land use 935 as covariates. Third, the uncertainty of soil properties can be estimated, and adaptive 936 surrogate modeling based on statistical regression and machine learning may be used to 937 assess effects of the uncertainty of soil properties on ESMs (Gong et al., 2015; Li et al. 2018). 938 Last but not the least, the layer schemes of soil data sets need to be converted for model use 939 and missing values for deeper soil layers needs to be filled.

940

916

941 Comment: \* "Two kinds of soil data are generated from soil surveys: soil polygon maps 942 representing distribution of soil types and soil profiles with observations of soil properties. 943 ESMs usually require the spatial distribution of soil properties, or soil property maps rather 944 than soil classification information.": It is unclear how the information of the two sentences 945 relates. Would this be correct: "Two kinds of soil data are generated from soil surveys: a 946 classification of soil type (usually in the form of polygon maps) and soil profiles with 947 observations of soil properties. ESMs usually require the spatial distribution of soil properties 948 (soil property maps) rather than a classification of soil type." If so please always use the same 949 term for the same information.

950

951 Reply: you are right. we modified as follows:

952

953 Two kinds of soil data are generated from soil surveys: a map (usually in the form of polygon 954 maps) representing main soil types in a landscape unit and soil profiles with observations of 955 soil properties which are considered representative for the main component soils of the 956 respective mapping units. ESMs usually require the spatial distribution of soil properties (i.e., 957 soil property maps) rather than information about soil types. Two kinds of methods, i.e. the 958 linkage method and the digital soil mapping method, are used to derive soil property maps. 959 960 p.4

961 Comment: \* "Soil maps show the geographical distribution of soil types,": I think this is too
962 general, the term "soil map" is not a technical, well specified term. Rather speak explicitly of
963 "soil type maps" to distinguish it from maps of soil properties.

964

Reply: In soil science, if it is not clarified, soil map refers to soil type map. To clarify this, we
modified to: Soil maps (the term soil map refers to soil type map in this paper) show the
geographical distribution of soil types.

968

969 Comment: \* I. 153 ff linkage method: this is a useful description, but hard to read for non-970 experts. Please improve the clarity of the text. For example: \* my understanding is that 971 pedotransfer functions map well-observable to less-wellobservable properties, but here it 972 sounds as if the PTFs are needed to link site-level (profile) observations of soil properties to 973 soil type maps.

974

975 Reply: Sorry for the description leading to the misunderstanding. Pedotransfer here has 976 nothing to do with Pedotransfer functions discussed in the late section. We added some 977 notification here: The linkage method (called the taxotransfer rule-based method) is to link 978 soil maps (with soil mapping units or soil polygons) and soil profiles (with soil properties) 979 according to taxonomy-based pedotransfer (taxotransfer in short, note that pedotransfer 980 here does mean pedotransfer functions which is a different thing) rules (Batjes, 2003). 981

982 Comment: \* "The criteria used in the linkage could be one or many factors as following […]
983 and so on": this is very vague. Which type of criteria is this: soil physical and chemical
984 properties?

985

Reply: this is related to the above comment. These are the criteria for linking soil map and soilprofiles with all soil properties together. Soil properties are not creteria.

988 989 Comme

989 Comment: \* "Each soil type is represented by one or a group of soil profiles that meet the
990 criteria, and usually the median or mean value of a soil property is assigned to the soil type.":
991 Criteria and properties are mixed up here. Isn't it choosing one (or several) property

as criteria, then mapping the rest?

993

994 Reply: this is related to the above comment. Soil properties are not creteria.

995

296 Comment: \* I. 165-172: how do these references relate to the examples of "major soil maps"297 in the introduction?

998

Reply: these references include both soil type maps and soil property maps, while "major soilmaps" in section 2.1 (not the introduction) refers to soil type map only.

1001

1002 Comment: \* I. 188 ff: Again, please add clarity. The difference between linkage method and1003 digital soil mapping is not just that the first has the same values across a polygon, but also in

| 1004 | what information is used as criteria for mapping: the digital soil mapping uses environmental     |
|------|---------------------------------------------------------------------------------------------------|
| 1005 | information, not just physical and chemical properties if I understood it correctly.              |
| 1006 |                                                                                                   |
| 1007 | Reply: This is also related to the misunderstanding of the term pedotranfer.                      |
| 1008 |                                                                                                   |
| 1009 | р. 5                                                                                              |
| 1010 | Comment: * "purity of soil map units is likely to be around 50 to 65%": which statistical measure |
| 1011 | is meant by "purity"?                                                                             |
| 1012 |                                                                                                   |
| 1013 | Reply: This is a term used in soil science, which means the percentage of the dominant soil       |
| 1014 | type in a soil map unit. Modified as: the purity of soil maps (referring to the following         |
| 1015 | website for the definition:                                                                       |
| 1016 | https://esdac.irc.ec.europa.eu/ESDB Archive/ESDBv2/esdb/sgdbe/metadata/purity m                   |
| 1017 | aps/purity.htm) is likely to be around 50 to 65% (Landon, 1991)                                   |
| 1018 |                                                                                                   |
| 1019 | р. 6                                                                                              |
| 1020 | Comment: * Why is IGBP-DIS mentioned here the first time? It should have been mentioned           |
| 1021 | under the linkage or the digital mapping methods (depending on what method is used)               |
| 1022 | before.                                                                                           |
| 1023 |                                                                                                   |
| 1024 | Reply: IGBP-DIS is listed in Table 1. It is produced by the linkage method. We added IGBP-        |
| 1025 | DIS under the linkage method: At the global level, many databases were derived by the             |
| 1026 | linkage method: the FAO Soil Map of the World with derived soil properties (FAO, 2003a), the      |
| 1027 | Data and Information System of International Geosphere-Biosphere Programme (IGBP-DIS)             |
| 1028 | database (Global Soil DataTask, 2000),                                                            |
| 1029 |                                                                                                   |
| 1030 |                                                                                                   |
| 1031 | Comment: * "soil organic carbon stocks at 1m depth": is it meant "carbon stocks down to a         |
| 1032 | depth of 1m"?                                                                                     |
| 1033 |                                                                                                   |
| 1034 | Reply: Yes, we corrected it.                                                                      |
| 1035 |                                                                                                   |
| 1036 | Comment: Fig. 1: remove superfluous information that costs the reader time to read and hides      |
| 1037 | the differences between the panels (the datasets): since the legend is the same for all sand      |
| 1038 | (clay) panels it does not have to be repeated; same for "sand (clay) at 0-30cm (%)", which is     |
| 1039 | even stated in the caption. "Longitude" (typo!) and "latitude" are also superfluous information.  |
| 1040 |                                                                                                   |
| 1041 | Reply: we removed superfluous information.                                                        |
| 1042 |                                                                                                   |
| 1043 |                                                                                                   |
| 1044 | Comment: Fig. 2: Same comment as for Fig. 1. "s" missing in soilarids. Why is IGBP not            |
| 1045 | included here as well? A more useful information for modelers would be the total carbon           |
| 1046 | content down to a certain depth rather than units of a/ka.                                        |
| 1047 |                                                                                                   |
|      |                                                                                                   |

1048 Reply: we removed superfluous information and add IGBP. Soil carbon stock maps can be 1049 calculated based on the soil organic carbon, coarse fragment and bulk density. Due to the 1050 evaluation of this study and a former study, the most accurate one is Soilgrids. Figure 1 of 1051 Tifafi et al. (2018) showed this map. We added: Tifafi et al. (2018) found that the global soil 1052 organic carbon stocks down to a depth of 1m is 3,400 Pg estimated by Soilgrids while it is 1053 2500 Pg by HWSD, and the estimates by Soilgrids are closer to the observations, though they 1054 all underestimated the soil carbon stocks. Figure 1 of Tifafi et al. (2018) showed the global 1055 distribution of soil carbon stocks by Soilgrids and HWSD. 1056 1057 p. 6 cont'd 1058 Comment: \* "several most popular ESMs": give objective criteria for "popular" 1059 1060 Reply: Here we do not have objective criteria. So, we delete this word. Instead, we just 1061 extended the list of ESMs according CMIP5. Our focus is on the soil datasets rather than ESMs. 1062 So, we did not assess or indicate the popularity in the list of Table 1. 1063 1064 Comment: \* I. 227-229: Again, it should be stated in how far the new datasets are superior 1065 over previous datasets. 1066 1067 Reply: This is quantitatively assessed in section 3. 1068 1069 Comment: \* I. 231: "This was started: :: " sounds a bit like advertisement and subjective, 1070 certainly other groups have been working on this to some extent for a long time as well. 1071 Reformulate more neutrally? 1072 1073 Reply: we modified it to: The Land-Atmosphere Interaction Research Group at Beijing Normal 1074 University (BNU, now at Sun Yat-sen University) has put much efforts on this topic. 1075 1076 Comment: \* I. 245-253: What is the purpose of these references? Only if they prove model 1077 results have improved by the usage of the new soil map is it useful to cite them here. 1078 1079 Reply: These citations are showing the application of the new soil datasets in ESMs, which is 1080 stated in the first sentence of the paragraph: In recent years, efforts were taken to improve 1081 the soil data condition in ESMs. Note that not all the citation has a comparison with the old 1082 datasets. 1083 1084 Comment: Tab. 1: please fix typos (inconsistent punctuation and capitalization). Add version 1085 numbers to LSMs, as usage of soil information may change between versions. Not sure the 1086 references are always correct, e.g. LeQuere et al., ESDD 2018 a and b ("Global carbon budget 1087 2017" and "2018", resp) state Reick or Mauritsen as JSBACH references, not Giorgetta. 1088 1089 Reply: we checked this table and make corrections. We added version numbers to LSMs if 1090 possible. We changed Mauritsen et al. (2019) as JSBACH references. 1091

| 1092         | p. 7                                                                                                   |
|--------------|--------------------------------------------------------------------------------------------------------|
| 1093         | Comment: * I. 299 ff: if there are no uncertainty estimates, how can you judge soilgrids to be         |
| 1094         | the most accurate one?                                                                                 |
| 1095         |                                                                                                        |
| 1096         | Reply: According the evaluation in section 3, Soilgrids is the most accurate one.                      |
| 1097         |                                                                                                        |
| 1098         | Comment: * I. 305: not all models apply PTFs, some directly require these less observable              |
| 1099         | variables as input, as you show in Tab. 1                                                              |
| 1100         |                                                                                                        |
| 1101         | Reply: It is true. But these variables are also derived by PTFS. To be precise, we modified it to:     |
| 1102         | Earth system modellers have employed different pedotransfer functions (PTFs) to estimate               |
| 1103         | soil hydraulic parameters (SHP), soil thermal parameters (STP), and biogeochemical                     |
| 1104         | parameters (Loov et al., 2017:Dai et al., 2013) or used these parameters as model inputs.              |
| 1105         | [                                                                                                      |
| 1106         | p. 9                                                                                                   |
| 1107         | Comment: * 1, 359: The methods have been introduced before so technical terms like                     |
| 1108         | "SMU" should have been introduced in these earlier chapters                                            |
| 1109         |                                                                                                        |
| 1110         | Reply: We introduced it in describing soil type maps: There are many soil mapping units                |
| 1111         | (SMU) in a soil map and a SMU is composed of more than one component (i.e. soil type) in               |
| 1112         | most cases                                                                                             |
| 1113         |                                                                                                        |
| 1114         | Comment: * 1 365: A problem of using subgrid soil information is that FS modelers do not               |
| 1115         | know how to map them with land use information, which is also subgrid level. This may be               |
| 1116         | the more fundamental obstacle than the computational issues that are mentioned                         |
| 1117         |                                                                                                        |
| 1118         | Reply: Yes, this will increase the model complexity, too. We added: This will bring the problem        |
| 1119         | of how to map the soil subgrids with land cover (or plant function type) subgrids. A possible          |
| 1120         | solution is to: classify soil according soil properties and get a number of defined soil classes       |
| 1121         | (SC n classes) like land cover types (I CT m classes); overlay the defined soil classes with land      |
| 1122         | cover types and get n by m combinations assuming soil classes and land cover types are                 |
| 1122         | independent. However, this will increase the computing time and the complexity of FSMs'                |
| 1120         | structure which needs to implement the soil processes over each suborid soil column within             |
| 1124         | a grid instead of the entire model grid                                                                |
| 1126         | a gha instead of the entire model gha.                                                                 |
| 1120         | n 11                                                                                                   |
| 1127         | p. 11
Comment: * "The temporal variation of global soil is quite challenging due to lack of data ": |
| 1120         | the aspect of temporal changes has not been addressed before and scores out of place in the            |
| 1120         | summany                                                                                                |
| 1121         | Summary.                                                                                               |
| 1120         | Poply We added a section (section 4.3) about this                                                      |
| 1100         |                                                                                                        |
| 1124         | Comment: * "Soil image fusion is also needed to marge the local and clobal soil marge": What           |
| 1105
1105 | is soil image fusion? Don't bring new methods in the summer eaction                                    |
| TTOD         | is some mage rusion? Don't bring new methods in the summary section                                    |

1137 Reply: This is about outlook instead of summary. Soil image fusion is proposed by Hengl et 1138 al. (2017), which consider local and global soil maps as components of soil variation for 1139 ensemble predictions. We modified this to: Soil image fusion is also needed to merge the 1140 local and global soil maps, which consider them as components of soil variation for ensemble 1141 predictions (Hengl et al., 2017). A system for automated soil image fusion might take years 1142 before an operational system for global soil data fusion is fully functional.

1143

1144 Comment: \* " Uncertainty estimation should be included in the soil datasets developed in the 1145 future.": of course uncertainty estimates build trust in an observational dataset. But how do 1146 the authors recommend should ESMs use such uncertainty estimates other than as criterion 1147 for which dataset to choose in the first place? Running multiple simulations combining upper 1148 and lower bounds in all possible combinations is too expensive...

1149

Reply: We agree that running ESMs with all possible combinations is too expensive. An alternative to quantify effects of the uncertainty of soil properties on ESMs may be to use adaptive surrogate modeling based on statistical regression and machine learning which costs much lower computing time (Gong et al., 2015; Li et al. 2018). We discussed this using a section:

1155 4.4 Incorporating the uncertainty of soil data in ESMs

1156 Incorporating the uncertainty of soil data in ESMs is a rising challenge. Except WISE30sec, all 1157 the current global soil data sets do not have a corresponding uncertainty map for a soil 1158 property. But the spatial uncertainty can be estimated by the methods mentioned in section 1159 2.1 and soil data sets with uncertainty map will be made available sooner or later. It is too 1160 expensive to run multiply ESM simulations combining upper and lower bounds in all possible 1161 combinations to quantify the effect of soil data uncertainty on ESMs. Instead, adaptive 1162 surrogate modeling based on statistical regression and machine learning can be used, which 1163 costs much lower computing time and proves to be effective and efficient (Gong et al., 2015; 1164 Li et al. 2018). Surrogate models are used to emulate the responses of ESMs to the variation 1165 of soil properties at each location.

1166

1167 Comment: \* "The gap between soil data existence and data availability is huge": Reads
1168 awkward. Better "The gap between the amount of data that has been taken in surveys and
1169 the amount of data freely available is large."

1170

1171 Reply: Modified: The gap between the amount of data that has been taken in surveys and1172 the amount of data freely available is large.

1173

1174 p. 12

1175 Comment: \* I. 482 "like many other data": Too general a statement, remove.

1176

1177 Reply: Thanks for mentioning this point. Data sharing is a very important issue for the whole

1178 science community. So I would like to keep it here.

| 1180 | Comment: I. 481 ": : : which has the most: : : ": how do you know? Add reference or justify in  |
|------|-------------------------------------------------------------------------------------------------|
| 1181 | other ways                                                                                      |
| 1182 |                                                                                                 |
| 1183 | Reply: We added a citation here (Batjes et al., 2017).                                          |
| 1184 |                                                                                                 |
| 1185 | Comment: * Arbitrary last sentence. I. 465 already mentions the subgrid issue in ESMs. Is there |
| 1186 | no more general conclusion that can be given? Otherwise just delete the last paragraph and      |
| 1187 | end with the more "outlook"-like previous paragraph.                                            |
| 1188 |                                                                                                 |
| 1189 | Reply: the last paragraph is deleted.                                                           |
| 1190 |                                                                                                 |
| 1191 |                                                                                                 |
| 1192 |                                                                                                 |
| 1193 |                                                                                                 |

[revised manuscript text omitted]
 manning technology and then the soil detects that have already been                    |
| incomposed in ESMs. Spation 2 presents DTEs that are used in ESMs to estimate soil          |
| hudroulia and thermal nerometers. Section 4 describes how to deal with as it data           |
| derived by the linkage methods. Section 5 introduces the uncertained of high and by         |
| active by the linkage methods. Section 5 introduces the upscaling of high-resolution        |
| soli data to the coarse resolution of ESMs. Section 6 gives the summary and an              |
| outlook of further improvements.                                                            |
| Compared worth a delayer of desiring and detay to fee EQM 0. (1.1.4).                       |
| 2 General methodology of deriving soil datasets for ESMsSoil datasets used in               |
| ESMIS                                                                                       |
| 2.1 Global and national soil datasets                                                       |

[revised manuscript text omitted]

---

## Author Response (AR2)

The authors have duly considered and addressed the various queries from the editor and reviewers and provided adequate replies, in so far as possible. For example, evaluation statistics in Table 4 are with respect to profiles from WoSIS, quite a number of which were considered in the compilation of the IGBP, HWSD and GSDE products. Although this is duly mentioned in the text, this aspect should also be indicated in a footnote to Table 4, as tables are often considered without reference to the text.

Reply:

Thanks for the editor and reviewers for providing queries and comment for this manuscript. This helped a lot in improving it.   We hope this review will provide some value to the soil data development and Earth system modeling. We added a footnote to Table 4 as suggested.

Modification:

We added a sentence in the text: though quite a number of WoSIS soil profiles were considered in the complication of these datasets which means that this evaluation is not independent validation.

A footnote to Table 4 was added:   Quite a number of WoSIS soil profiles were considered in the compilation of the four products.

Overall, this is a useful and timely review. Unfortunately, the manuscript is still not very well written. There are still various flaws in the English used; these should be corrected by a professional editor.

Reply: We have taken a language service for English correction by two professional editors. Modifications can be seen in the marked manuscript.

Minor remarks (line numbers according to annotated manuscript)

line 69: usually single point observations at a given moment in time, instead of 'usually time-independent'.

Reply: modified.

line 135-142: Check numbering/naming of sections against text.

Reply: we checked this and modified the text.

Modification: In section 2, we first introduce soil datasets produced by the linkage method and digital soil mapping technology at global and national scales, and then, we introduce the soil datasets that have already been incorporated into ESMs, and we also present PTFs that are used in ESMs to estimate soil hydraulic and thermal parameters. In section 3, several global soil datasets are compared and evaluated with a global soil profile database. In section 4, two issues regarding the model use of soil data are described and existing challenges related to the application of soil datasets in ESMs are discussed.   In Section 5, a summary and the outlook of further improvements are provided.

line 158, 321-322: remove bold case

Reply: modified.

line 484: the proportion of coarse fragments is also considered in HWSD and WISE30sec.

Reply: Here we made a small mistake. Only IGBP does not contain coarse fragments.

Modification: Notably, only the IGBP does not contain coarse fragments.

line 1628-1630, 1714: rephrase for clarity. PTFs are empirical, predictive functions of certain soil properties (e.g. hydraulic conductivity) from more easily obtained soil properties (e.g. sand, silt, clay and organic carbon content).
Reply: modified.
Modification: PTFs are the empirical, predictive functions that account for the relationships between certain soil properties (e.g., hydraulic conductivity) and more easily obtainable soil properties (e.g. sand, silt, clay and organic carbon content).

line 1931-1933: the soil map is the base map here, not a co-variate.
Reply: modified. However, we argue that from the perspective of 'scorpan' framework soil map can be considered as the major covariate (sometimes the only one) for the linkage method.
Modification: the linkage method usually considers the soil map to be a base map line 1967: ⋯ countries, most of these are not yet freely available to the international community.
Reply: modified.
Modification: countries, although most of these are not yet freely available to the international community.

line 1969: Should refer to NCSS as the data provider here: https://ncsslabdatamart.sc.egov.usda.gov/
Reply: modified.

line 1337: Fifth column, GSDE. 0 is given as depth of bottom layer. Delete this.
Reply: deleted.

[revised manuscript text omitted]
 (% in weight) | SoilGrids | -0.906 | 18.6 | 0.457 | 0.518 | -0.27 | 19.1 | 0.501 | 0.492 |
| | GSDE | -0.443 | 23.2 | 0.571 | 0.247 | -1.31 | 23.8 | 0.625 | 0.211 |
| | HWSD | 6.64 | 27.4 | 0.673 | 0.014 | 2.08 | 27.6 | 0.725 | -0.058 |
| | IGBP | 3.74 | 26.3 | 0.647 | 0.051 | 4.06 | 26.3 | 0.691 | 0.055 |
| Clay content (% in weight) | SoilGrids | 1.34 | 12.5 | 0.554 | 0.339 | 0.39 | 13.6 | 0.485 | 0.382 |
| | GSDE | -0.949 | 14.6 | 0.643 | 0.104 | -0.79 | 16.4 | 0.584 | 0.105 |
| | HWSD | 0.77 | 16.2 | 0.718 | -0.119 | 1.42 | 18.9 | 0.672 | -0.182 |
| | IGBP | 3.27 | 15.4 | 0.678 | 0.044 | 2.44 | 16.8 | 0.597 | 0.084 |
| Bulk density (kg/m3) | SoilGrids | -79.7 | 237 | 0.164 | 0.338 | -33.5 | 212 | 0.136 | 0.327 |
| | GSDE | -68.4 | 279 | 0.193 | 0.030 | -65.5 | 269 | 0.173 | -0.043 |
| | HWSD | -105 | 298 | 0.206 | -0.033 | -168 | 317 | 0.204 | -0.107 |
| | IGBP | -55.6 | 273 | 0.189 | 0.050 | -112 | 294 | 0.189 | -0.130 |
| Coarse fragment (% in volume) | SoilGrids | 1.53 | 10.1 | 1.68 | 0.319 | 1.23 | 12.8 | 1.47 | 0.335 |
| | GSDE | 3.2 | 13.5 | 2.24 | -0.165 | 3.18 | 16.8 | 1.93 | -0.115 |
| | HWSD | 1.8 | 13.2 | 2.2 | -0.164 | -0.40 | 16.2 | 1.87 | -0.081 |
| Organic carbon (g/kg) | SoilGrids | 6.21 | 29.8 | 1.69 | 0.218 | 0.99 | 23.5 | 3.32 | 0.134 |
| | GSDE | -0.354 | 34.5 | 1.95 | -0.095 | 0.45 | 27.4 | 3.87 | -0.174 |
| | HWSD | -3.67 | 36.2 | 2.05 | -0.194 | -1.38 | 27.4 | 3.87 | -0.172 |
| | IGBP | 0.61 | 33.4 | 1.89 | -0.026 | 1.67 | 28.5 | 4.02 | -0.268 |

*Quite a number of WoSIS soil profiles were considered in the compilation of the four products.
ME is the mean error. RMSE is the root mean squared error. CV is the coefficient of variation. $R^2$ is the coefficient of determination.

[Figure]

Figure 1 Soil sand and clay fraction at the surface 0-30 cm layer from SoilGrids, IGBP-DIS and GSDE. The difference among them will lead to different modelling results for ESMs. IGBP-DIS is Data and Information System of International Geosphere-Biosphere Programme, and GSDE is Global Soil Dataset for Earth System Model.

[Figure]

Figure 2 Soil organic carbon and bulk density at the surface 0-30 cm layer from
SoilGrids, GSDE and IGBP.